# Integral Performance Approximation for Continuous-Time Reinforcement Learning Control

**Brent A. Wallace & Jennie Si**
Department of Electrical Engineering
Arizona State University
Tempe, AZ, USA
{bawalla2,si}@asu.edu

## Abstract

We introduce integral performance approximation (IPA), a new continuous-time reinforcement learning (CT-RL) control method. It leverages an affine nonlinear dynamic model, which partially captures the dynamics of the physical environment, alongside state-action trajectory data to enable optimal control with great data efficiency and robust control performance. Utilizing Kleinman algorithm structures allows IPA to provide theoretical guarantees of learning convergence, solution optimality, and closed-loop stability. Furthermore, we demonstrate the effectiveness of IPA on three CT-RL environments including hypersonic vehicle (HSV) control, which has additional challenges caused by unstable and nonminimum phase dynamics. As a result, we demonstrate that the IPA method leads to new, SOTA control design and performance in CT-RL.

## 1 Introduction

Many important applications such as flight control (Stengel, 2022), robotics (Craig, 2005), process control (Morari & Zafiriou, 1989), and waste water treatment (Yang et al., 2022) are inherently continuous-time in their dynamics and require real-time high performance controls. However, an accurate dynamic model is required in the traditional classical control methods. This poses great challenges, for example, in the HSV context, for which aeroprorolsive/aeroelastic effects are exceptionally difficult to model (Bolender & Doman, 2006b; Dickeson et al., 2009b). This modeling challenge extends to almost all control applications, as modeling complex systems perfectly is not feasible. Reinforcement learning control designs have provided a new way of circumventing this stringent requirement, as RL can learn from data to accommodate unmodeled dynamics.

Currently, there are two major classes of algorithms for continuous-time RL (CT-RL). The SOTA deep RL (DRL) Fitted Value Iteration (FVI) methods (Lutter et al., 2021; 2023b), which have achieved some of the greatest empirical successes in CT-RL to-date, require data from over 5,000,000 simulations to solve the simple inverted pendulum control task (cf. Section 5). For mission-critical applications like HSVs, achieving millions of test flights is impossible – typically only a few test flights can be conducted. Thus, the large data requirements of deep CT-RL methods presents great challenges to many real-world applications where simulation data from hardware is expensive to collect and limited in quantity. The adaptive dynamic programming (ADP) methods represented by seminal works of (Vrabie & Lewis, 2009; Vamvoudakis & Lewis, 2010; Jiang & Jiang, 2014; Yang et al., 2024) (cf. Appendix M) have achieved promising theoretical results and attracted substantial research attention and body of work since about fifteen years ago. Yet, few ADP works have resulted in meaningful controllers, as evaluations of those methods have been limited to simple toy problems with closed-form solutions known *a priori* (Wallace & Si, 2024).

Even though discrete-time (DT) RL algorithms (Sutton & Barto, 2018; Lewis et al., 2012b; Si et al., 2004; Bertsekas, 2017) have demonstrated extensive theoretical guarantees and demonstrations in applications, CT-RL algorithms have only developed a handful of results. One approach to CT-RL discretizes the CT environment and/or value integral in order to approximate a CT problem by a DT

one (Kim et al., 2021; Yildiz et al., 2021). Discretization of CT environments, however, may cause acute numerical issues for real-world systems, especially ones with complex dynamics like the HSV (Chen & Francis, 1995). Even if a discretization can be formed, this casts the problem into a completely different class of DT control problem. The CT HJB equation (a first-order nonlinear PDE) is not equivalent to the DT Bellman equation (a difference equation). For additional discussion, see (Wallace & Si, 2024; Cao & Pan, 2024). Another approach resorts to LQR/linearization to eliminate the nonlinearity challenge, as in seminal works (Bradtke et al., 1994) and subsequent ADP (Jiang & Jiang, 2012; Jha et al., 2019) and $Q$-learning-based methods (Possieri & Sassano, 2022). A third approach addresses CT-RL problems under general nonlinear (non-affine) dynamics (Yildiz et al., 2021; Sandoval et al., 2023). However, fully-nonlinear algorithms are at an early stage. The existing evaluations study simple cart/pendulum variant systems and second-order academic examples. Comprehensive theoretical results and meaningful designs without stringent assumptions are yet to be developed. Perhaps the most studied fourth approach deals with affine nonlinear systems, including the aforementioned ADP works and deep CT-RL continuous fitted value iteration (cFVI) (Lutter et al., 2021) and robust FVI (rFVI) (Lutter et al., 2023b). While both ADP works and FVIs require an affine-nonlinear model, the FVIs have demonstrated learning with low variance and meaningful control performance, thus standing as SOTA in CT-RL.

From an applications standpoint of CT-RL, take HSVs as an example. CT-RL methods have been developed for HSVs, yet these studies are of limited scope as they do not address the model uncertainty of HSVs and consequently their controls. For instance, Zhao et al. (2023) develops a composite RL/observer-based attitude control method for HSVs. However, the HSV model used is a simplified approximation of the standard Wang and Stengel model (Wang & Stengel, 2000; Marrison & Stengel, 1998; Shaughnessy et al., 1990) in which Mach dependencies are neglected, a significant limitation in the high-Mach hypersonic regime. The neural control methods in (Xu et al., 2013; 2015) use this same simplified model. Along a similar vein, the developed stability results of many RL HSV works (Zhao et al., 2023; Xu et al., 2013; 2015; Bu et al., 2019; Bu & Qi, 2022; Qiao et al., 2019) which are essential in flight control applications require that multiple complex stability inequalities hold simultaneously along trajectories, and no constructive method is provided for ensuring that the inequalities are met. Furthermore, these RL-based HSV control works do not provide substantive generalization studies to unmodeled dynamics.

A realistic HSV control problem is challenging, as these high-performance aircraft fly 20 times faster than commercial jets. Their long shape and rearward-set propulsion systems give HSVs a nose-up pitch instability (placing hard lower bounds on how "slow" the HSV can be controlled). Meanwhile, the parasitic coupling from upward deflections of the tail causing a temporary dip in altitude results in nonminimum phase behavior (placing hard upper bounds on how "fast" the HSV can be controlled) (Bolender & Doman, 2006a). The combination of these two constraints makes HSV control exceptionally challenging (Rodriguez et al., 2008).

**Contributions.** We propose a new model-based CT-RL learning method that leads to the following three contributions: 1) Our novel IPA CT-RL design approach takes advantage of an affine nonlinear dynamic model of the environment and a quadratic cost performance structure, thus to exploit the Kleinman's solution framework. By using state-action data-driven learning, we address unmodeled dynamics with great data efficiency and robust control performance. 2) We provide theoretical guarantees of IPA CT-RL learning convergence, solution optimality, and system stability alongside comprehensive evaluations and comparisons to demonstrate IPA-enabled SOTA CT-RL results in optimal control problems. 3) We demonstrate, perhaps for the first time, that a CT-RL method (IPA) has successfully addressed the challenging optimal control of HSVs.

## 2 Method

The IPA method takes advantage of an affine nonlinear model of the environment, which is usually not an accurate representation of the physical environment, and utilizes state-action data with learning to accommodate uncertainties. It is therefore an organic integration of classical control design principles and RL for adaptive optimal control. We assume an affine nonlinear environment of form:

$$\dot{x} = f(x) + g(x)u, \tag{1}$$

where $x \in \mathbb{R}^n$ is the state, $u \in \mathbb{R}^m$ is the control, $f : \mathbb{R}^n \to \mathbb{R}^n$, and $g : \mathbb{R}^n \to \mathbb{R}^{n \times m}$. As standard, we assume $f$ and $g$ are Lipschitz on a compact set $\Omega \subset \mathbb{R}^n$ containing the origin $x = 0$ in

its interior, and that $f(0) = 0$. We consider the infinite-horizon undiscounted cost

$$J(x_0) = \int_0^\infty (x^T Q x + u^T R u) \, d\tau, \tag{2}$$

where $Q \in \mathbb{R}^{n \times n}$, $Q = Q^T \geq 0$ and $R \in \mathbb{R}^{m \times m}$, $R = R^T > 0$ are the state and control penalties.

## 2.1 THE IPA ALGORITHM

**Bellman Optimality for Continuous-Time Systems: The HJB Equation.** Consider the control affine nonlinear dynamics of (1), and the infinite-horizon undiscounted performance index of (2). CT-RL and CT optimal control both aim to solve the common HJB equation for optimal value (a continuous-time analogy of the Bellman equation in discrete time), from which an optimal control is computed. We begin with the system Hamiltonian function (Lewis et al., 2012a).

$$H(x, u, \tfrac{\partial V}{\partial x}) = (f(x) + g(x)u)^T \tfrac{\partial V}{\partial x} + x^T Q x + u^T R u. \tag{3}$$

The goal of CT-RL is to find the optimal control policy $\mu^*$ which minimizes the value in (2), yielding the optimal value $V^*$. Such a solution can be obtained from the necessary condition based on Pontryagin's minimum principle: $\frac{\partial H}{\partial u} = 0$. Plugging in Pontryagin's minimum principle to the Hamiltonian $H$ (3) in the case of affine nonlinear dynamics reduces to:

$$0 = \tfrac{\partial H}{\partial u}\big|_{u=\mu^*(x)} = g^T(x)\tfrac{\partial V^*}{\partial x}(x) + 2R\mu^*(x) \quad \implies \quad \mu^*(x) = -\tfrac{1}{2}R^{-1}g^T(x)\tfrac{\partial V^*}{\partial x}(x). \tag{4}$$

Note that we are able to reach this nice, closed-form control solution for the optimal nonlinear policy $\mu^*$ given the optimal value $V^*$ because of the quadratic cost structure in the performance index (2) in combination with the control affine nonlinearity in the form of (1). Plugging now the closed-form optimal policy $\mu^*$ (4) into the Hamiltonian $H$ (3) and applying Pontryagin's minimum principle $\frac{\partial H}{\partial u} = 0$, we arrive at the HJB equation for affine nonlinear systems (Figure 1, Block #1):

$$0 = H(x, \mu^*(x), \tfrac{\partial V^*}{\partial x}) = f^T(x)\tfrac{\partial V^*}{\partial x} - \tfrac{1}{4}\tfrac{\partial V^{*T}}{\partial x}g(x)R^{-1}g^T(x)\tfrac{\partial V^*}{\partial x} + x^T Q x. \tag{5}$$

Our focus now is to solve for $V^*$ from the HJB (5), in turn yielding the optimal policy $\mu^*$ from (4).

**Setting up Critic Value and Respective Policy Based on Generalized HJB Equation.** We now inspect the value $V$ (critic value) from a perspective of the generalized HJB (GHJB) equation (Beard & McLain, 1998) below. Here, the goal of the critic value $\hat{V}$ is to approximate the cost index $J$ in (2) achieved when the control $u = \mu_i(x)$ is applied to the nonlinear system. A function $V$ approximates the cost $J$ of current policy $\mu_i$ if and only if $V$ satisfies the GHJB:

$$(f + g\mu_i)^T \tfrac{\partial V}{\partial x} + x^T Q x + \mu_i^T R \mu_i = 0. \tag{6}$$

This is a differential equivalent to the value integral in (2); indeed, given any function $V$ and policy $\mu_i$ with cost $J$ (2), $V = J$ if and only if $V$ satisfies the GHJB (6) (Beard & McLain, 1998).

**CT Temporal Difference Equation.** Let $\mu_i$ be the control policy at the $i$-th iteration with respective value $V$. Given the performance index (2), re-written below in the current context for convenience,

$$V(x(t)) = \int_t^\infty (x^T Q x + \mu_i^T(x) R \mu_i(x)) \, d\tau, \tag{7}$$

we can obtain the following CT temporal difference equation (Figure 1, Block #4), analogous to the discrete-time Bellman equation as follows,

$$V(x(t_0)) - V(x(t_1)) = \int_{t_0}^{t_1} x^T Q x + \mu_i^T(x) R \mu_i(x) \, d\tau, \tag{8}$$

where in the above, the left-hand side is the CT temporal value difference, and the right-hand side is called the **integral reinforcement signal – note that it requires requires only state-action data.** Just as in discrete time where the temporal difference points to the stage cost, in CT the temporal difference becomes an integral reinforcement signal. Similarly to DT, Bellman's optimality principle from the HJB (5) dictates that the optimal cost $V(x(t_0))$ at time $t_0$ is the same as minimizing the cost accrued over the interval $[t_0, t_1]$ plus the optimal cost $V(x(t_1))$ at time $t_1$. Analogous to discrete time TD, we use CT TD to learn the value function in (7). Given the time indices $t_0 < t_1$, we examine the CT temporal value difference $V(x(t_0)) - V(x(t_1))$ (Figure 1, Block #2), the integral reinforcement signal $\int_{t_0}^{t_1} x^T Q x + \mu_i^T(x) R \mu_i(x) \, d\tau$, and the quadratic cost (2) (Figure 1, Block #3) to develop a CT learning rule for solving for $V$ from the HJB equation (Figure 1, Block #1).

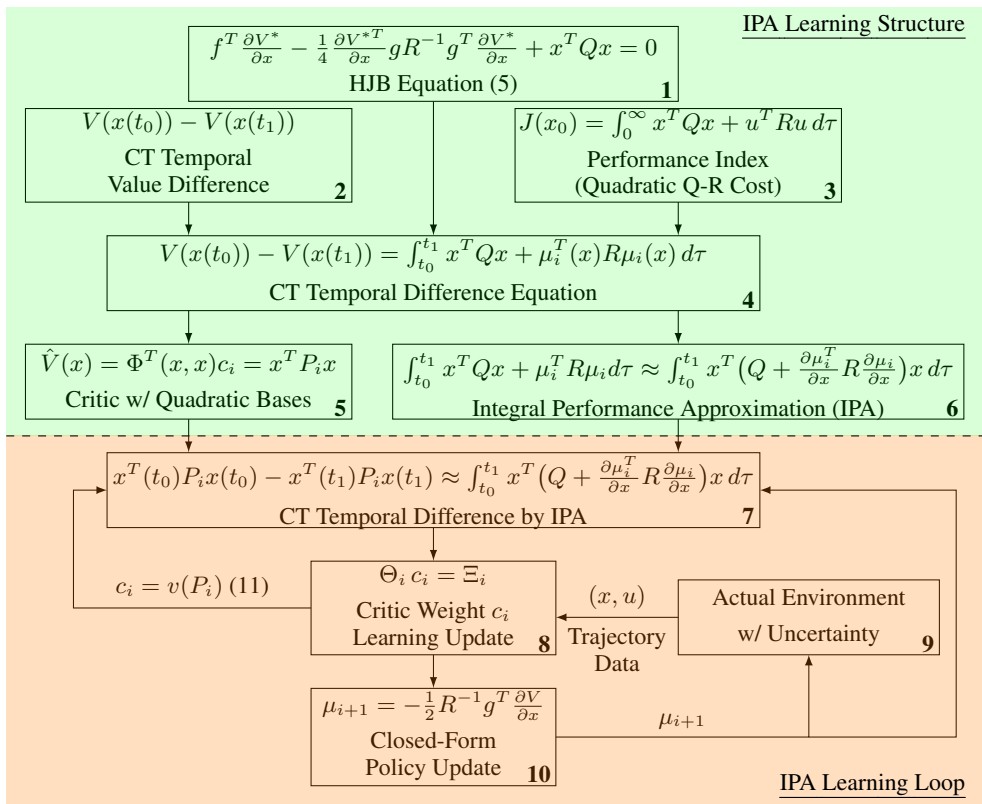

Figure 1: IPA algorithm block diagram. IPA RL begins with the HJB equation (Block #1), which when combined with the CT temporal value difference (Block #2) and IPA's quadratic Q-R cost index (Block #3) yields the CT temporal difference (TD) equation (Block #4). When IPA's critic with quadratic bases (Block #5) is combined with the novel integral performance approximation scheme (Block #6), this yields the IPA continuous-time TD (Block #7), which in turn forms the basis of the IPA critic weight learning update (Block #8). IPA learning occurs in a closed loop with the actual physical process with model uncertainty (Block #9), which supplies state-action data $(x, u)$ for the IPA critic weight update (Block #8). IPA updates its policy via the closed-form HJB-based update (Block #10), from which a new weight update is constructed, and so on.

**The Critic and its Bases.** We construct the critic network as in (9) below,

$$\hat{V}(x) = \Phi^T(x,x)c_i. \tag{9}$$

Here, $c_i \in \mathbb{R}^{\underline{n}}$, $\underline{n} \triangleq \frac{n(n+1)}{2}$, are the critic weights at the $i$-th iteration. We have chosen quadratic bases $\Phi(x,x) \in \mathbb{R}^{\underline{n}}$ as approximation features of the value $V$ to leverage the quadratic Q-R cost structure in (2), and we define these bases as

$$\Phi(x,y) = \tfrac{1}{2}\big[2x_1y_1,\, x_1y_2 + x_2y_1, \ldots,\, x_1y_n + x_ny_1,\, 2x_2y_2, \ldots,\, 2x_ny_n\big]^T. \tag{10}$$

To further see that our choice of bases is quadratic, the critic $\hat{V}$ (9) can be represented equivalently by the following quadratic form $c_i \leftrightarrow P_i$, $P_i = P_i^T \in \mathbb{R}^{n \times n}$ (Figure 1, Block #5):

$$\hat{V}(x) = \Phi^T(x,x)c_i = \Phi^T(x,x)v(P_i) = x^T P_i x, \tag{11}$$

where for a symmetric matrix $P = P^T \in \mathbb{R}^{n \times n}$, define its "vectorization" $v(P) \in \mathbb{R}^{\underline{n}}$ as

$$v(P) = \big[p_{11}, 2p_{12}, \ldots, 2p_{1n}, p_{22}, 2p_{23}, \ldots, 2p_{n-1,n}, p_{nn}\big]^T. \tag{12}$$

Indeed, quick linear algebra shows that for each critic weight vector $c_i \in \mathbb{R}^{\underline{n}}$, there exists a unique symmetric matrix $P_i = P_i^T$ such that $c_i = v(P_i)$. The last equality in (11) is also quick algebra.

Equation (11) renders a quadratic function as an approximant of the integral cost (2), which is of the Q-R form. It is well-known in control theory and engineering that this Q-R cost structure is an appropriate form to be used as feedback control design objective, as it leads to physical insights of system performance (Lewis et al., 2012a; Beard & McLain, 1998).

**Integral Performance Approximation (IPA).** Now, define the following matrices to be used for integral performance approximation only (these are not the IPA policies):

$$K_i \triangleq - \tfrac{\partial}{\partial x}\{\mu_i(x)\}\big|_{x=0}. \tag{13}$$

Then $\mu_i(x) \approx -K_i x$, to first order; more precisely, $\mu_i(x) = -K_i x + o(\|x\|)$, where the residual $o(\|x\|)$ satisfies $\lim_{\|x\|\to 0} \frac{\|\mu_i(x) - K_i x\|}{\|x\|} = 0$. We may then approximate the integral reinforcement signal (8) with what we term **integral performance approximation (IPA)** (Figure 1, Block #6):

$$\int_{t_0}^{t_1} x^T Q x + \mu_i^T(x) R \mu_i(x)\, d\tau \approx \int_{t_0}^{t_1} x^T \big( Q + K_i^T R K_i \big) x\, d\tau, \qquad (\text{to } o(\|x\|^2)). \tag{14}$$

**CT Temporal Difference by IPA.** Now, substituting the quadratic bases (11) as the TD on the left-hand side of the CT TD equation (8), and substituting the IPA scheme (14) as the integral reinforcement signal in the right-hand side of (8), we get the CT TD by IPA (Figure 1, Block #7):

$$x^T(t_0) P_i x(t_0) - x^T(t_1) P_i x(t_1) \approx \int_{t_0}^{t_1} x^T \big( Q + K_i^T R K_i \big) x\, d\tau, \qquad (\text{to } o(\|x\|^2)). \tag{15}$$

With the above equation, and since $P_i$ is related to the critic weights $c_i$ as discussed in the above, we have now reached the basis of the IPA learning algorithm. As will be seen, its quadratic structure enables our suite of theoretical guarantees (Section 3) as well as high data efficiency (Section 5).

**IPA for Critic Learning.** We now use IPA-facilitated CT temporal difference in (15) (which comprises a single trajectory sample) to construct a value function weight update. To solve for the weights $c_i$, we use $l \in \mathbb{N}$ trajectory samples. The update rule proceeds by a designer first selecting a sequence of sample instants $\{t_k\}_{k=0}^{l}$, and reference command input $r(t)$ for exploration (see Appendix F). We apply $r(t)$ to the **actual** environment (Figure 1, Block #9) under an initial stabilizing policy $\mu_0$, to collect the resulting state-action trajectory data $\big\{ \big( x(t), u(t) \big) \big\}_{t \in [t_0, t_l]}$. Applying (15) at the sample instants $\{t_k\}_{k=0}^{l}$, we arrive at the learning update (Figure 1, Block #8):

$$\Theta_i\, c_i = \Xi_i. \tag{16}$$

The exact form of the matrices $\Theta_i \in \mathbb{R}^{l \times \underline{n}}$, $\Xi_i \in \mathbb{R}^l$ and their algebraic derivation are provided in Appendix C. In essence, the matrices capture the IPA CT temporal difference features in (15) via:

$$\Theta_i c_i = \begin{bmatrix} x^T(t_0) P_i x(t_0) - x^T(t_1) P_i x(t_1) \\ \vdots \\ x^T(t_{l-1}) P_i x(t_{l-1}) - x^T(t_l) P_i x(t_l) \end{bmatrix}, \quad \Xi_i = \begin{bmatrix} \int_{t_0}^{t_1} x^T \big( Q + K_i^T R K_i \big) x\, d\tau \\ \vdots \\ \int_{t_{l-1}}^{t_l} x^T \big( Q + K_i^T R K_i \big) x\, d\tau \end{bmatrix} \tag{17}$$

**Policy Update using Closed-Form Formula.** As derived in (4), given the affine nonlinear dynamics $(f, g)$, the optimal policy update can be solved directly from:

$$\mu_{i+1}(x) = -\tfrac{1}{2} R^{-1} g^T(x) \frac{\partial \hat{V}}{\partial x}(x). \tag{18}$$

Having solved for the critic weights $c_i$ (16), we update the nonlinear control policy $\mu_{i+1}$ (Figure 1, Block #10) via (18), and so on, iteratively learning the optimal policy.

## 3 THEORETICAL RESULTS

Our choice of critic network structure $\hat{V}$ (11), nonlinear policy structure $\mu_i$ (18), and use of IPA (14) allows us to take advantage of classical control results in Kleinman's well-tested algorithm (Kleinman, 1968) to be combined with state-action data $(x, u)$ from the actual physical environment in learning. This enables us to develop our key theoretical guarantees. To state the properties of IPA, let $(A, B)$ denote the linearization of the affine nonlinearity $(f, g)$ (1). Then for all $i \geq 0$

$$\tfrac{\partial}{\partial x}\{f(x) + g(x)\mu_i(x)\}\big|_{x=0} = A - BK_i. \tag{19}$$

Similarly, $f(x) + g(x)\mu_i(x) = (A - BK_i)x + o(\|x\|)$. Let $P^* \in \mathbb{R}^{n \times n}$, $P^* = P^{*T} > 0$ be the Riccati equation solution for $(A, B)$ (Rodriguez, 2004), and $K^* \in \mathbb{R}^{n \times m}$ the optimal LQR control.

**Theorem 3.1 (Local Convergence, Optimality, Closed-Loop Stability, and Robustness of IPA)**

Suppose that the initial policy $\mu_0$ stabilizes the system (1), and that the sample instants $\{t_k\}_{k=0}^l$ are chosen such that the IPA matrix $\mathcal{I} \in \mathbb{R}^{l \times \underline{n}}$ given by

$$\mathcal{I} = \left[ \begin{array}{ccc} \int_{t_0}^{t_1} \Phi(x,x)\, d\tau & \cdots & \int_{t_{l-1}}^{t_l} \Phi(x,x)\, d\tau \end{array} \right]^T \tag{20}$$

has full rank $\underline{n}$ ($\underline{n}$ being the number of bases in critic approximation). Then identifying the quadratic form of the bases $c_i = v(P_i)$ from (11), IPA produces identical sequences of matrices $\{P_i\}_{i=0}^\infty$, $\{K_i\}_{i=0}^\infty$ as Kleinman's algorithm (Kleinman, 1968) to first order if the Kleinman control sequence is produced based on a linearized actual nonlinear process that is unknown to IPA. Thus:

- Each of the policies $\{\mu_i\}_{i=0}^\infty$ in (18) stabilizes the nonlinear system $(f, g)$ (1) under feedback.
- $P^* \leq P_{i+1} \leq P_i$ for all $i \geq 0$, and $\lim_{i \to \infty} P_i = P^*$. Thus, in the limit the value function $\hat{V}$ (11) is

optimal to second-order (stronger than first-order); i.e., $\hat{V}(x) = x^T P_i x \to V^*(x) + o(\|x\|^2)$.

- $\lim_{i \to \infty} \frac{\partial}{\partial x} \mu_i = -K^*$. Thus, in the limit the nonlinear policies $\{\mu_i\}_{i=0}^\infty$ (18) are optimal to first-

order; i.e., $\mu_i(x) \to \mu^*(x) + o(\|x\|)$.

As a direct result, IPA inherits the guaranteed stability robustness margins of Kleinman's algorithm: $\|S_u\|_{\mathcal{H}^\infty} \leq 0$ dB, $\|T_u\|_{\mathcal{H}^\infty} \leq 2$ dB, where $\|\cdot\|_{\mathcal{H}^\infty}$ denotes the $\mathcal{H}^\infty$ norm, and where $S_u, T_u$ denote the sensitivity and complementary sensitivity closed-loop maps, respectively, at the control loop breaking point $u$ (Rodriguez, 2004). For definitions and further discussion, see Appendix F.

*Proof:* An induction argument presented in Appendix D. ∎

## 4 EXPERIMENT SETUP FOR EVALUATIONS

In this section, we show how IPA learning and control performance compares to the two most promising classes of CT-RL methods: 1) the foundational ADP CT-RL methods as reviewed in (Wallace & Si, 2024), and 2) the SOTA FVIs (Lutter et al., 2021; 2023b).

We perform in-depth evaluations on three CT-RL environments: 1) a second order system (SOS) (Vamvoudakis & Lewis, 2010) that is what ADP methods were demonstrated on as being solvable, however, only in the case of using exact bases induced from *a priori* knowledge of the solution (but not solvable if the bases slightly differ from the true structure of the value function, refer to Appendix M), 2) a pendulum that was extensively evaluated in (Lutter et al., 2021; 2023b), the results of which stand as SOTA in current CT-RL, and 3) a hypersonic vehicle (HSV) (Wang & Stengel, 2000; Shaughnessy et al., 1990) that, for the first time, is being extensively evaluated in this study, which is considered a SOTA environment that has ever been studied in CT-RL.

Due to ADP's inability to solve even SOS under realistic conditions (see evaluations in Appendix M), this study focuses mainly on IPA and the FVIs for extensive and systematic evaluations. All implementation details and hyperparameter selections for these studies can be found in Appendix H. The dynamics, physical insights, and realistic modeling discrepancy issues of the environments are provided in Appendices J, K, L for the SOS, pendulum, and HSV, respectively.

**Baseline Methods.** These include ADP and FVIs with a strong focus on FVIs, as they have successfully synthesized meaningful controllers for the environments being evaluated:
- "**Continuous FVI (cFVI)**": SOTA deep CT-RL method (Lutter et al., 2021).
- "**Robust FVI (rFVI)**": Robust variant of SOTA FVI (Lutter et al., 2023b).

**Questions Addressed.** Our evaluations aim to quantitatively address the following:
**Q1:** How does IPA CT-RL learning performance (average cost, learning speed and variance, and success rate) compare to baseline FVIs?
**Q2:** How time/data efficient is IPA compared to FVIs?
**Q3:** How does IPA learning performance generalize to environment uncertainty compared to FVIs?
**Q4:** How does IPA cost performance generalize to environment uncertainty compared to FVIs?
**Q5:** How does IPA critic estimation generalize to environment uncertainty compared to FVIs?
**Q6:** How do IPA time responses generalize to environment uncertainty compared to FVIs?

**Remark on Simulating Unmodeled Dynamics.** For Q3–Q6 above, in order to simulate uncertainty present in the actual physical process, we study the performance of the algorithms under systematic variations of a model uncertainty parameter $\nu$ (nominally 1) inserted in the dynamics. For a detailed description of this parameter in relation to each of the three environments studied, please see Appendices J–L. For details on performance measures, refer to Appendix G.

**Remark on Environments.** For details on SOTA environments evaluated in leading CT-RL algorithms, please refer to Appendix A, where as shown that the pendulum and the HSV environments are SOTA. We use the SOS environment because it is the only one that can be successfully solved by all leading methods: ADP-based (if the bases are selected using *a priori* knowledge of the value function structure), FVIs, and the current IPA-based CT-RL.

## 5 COMPARISONS BETWEEN IPA AND DRL FVIS

This evaluation focuses on comparing IPA to SOTA deep RL cFVI and rFVI (Lutter et al., 2021; 2023b). Training and evaluation procedures are described in detail in Section 4 and Appendix H.

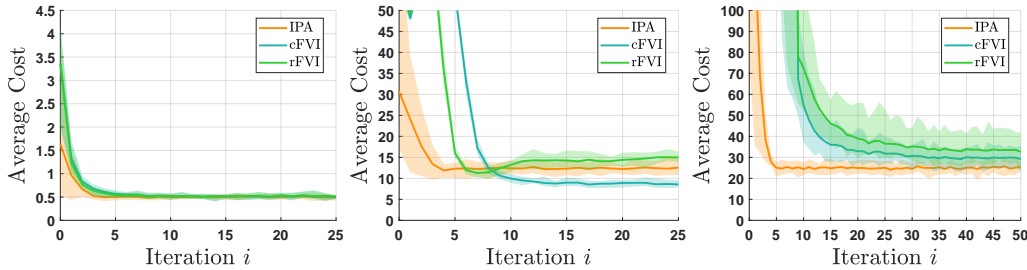

Figure 2: Learning curves of IPA and FVIs obtained over 20 seeds for the nominal models $\nu = 0\%$ of the SOS (left), pendulum (middle) and HSV (right). The shaded area displays the min/max range among seeds, as in the original works (Lutter et al., 2021; 2023b).

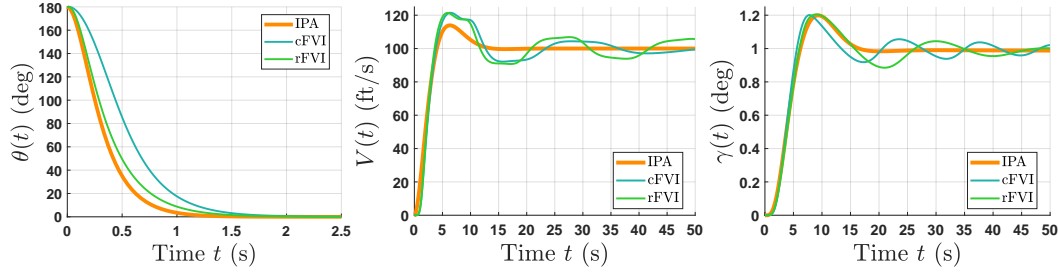

Figure 3: Closed-loop time-domain responses on nominal models $\nu = 0\%$. Left: Swing-up response of pendulum model $\theta(t)$ from hanging pendulum position $\theta_0 = 180°$. Middle: HSV response to 100 ft/s step velocity command. Right: HSV response to $1°$ step FPA command. Overall, IPA exhibits fast, well-behaved responses. FVIs on the HSV have large oscillatory transients.

**Q1: IPA exhibits the fastest learning convergence. Its cost performance and evaluation success rate meet or exceed FVIs.** The learning curves of the tested methods trained to 20 seeds for the three respective nominal models (model uncertainty level $\nu = 0\%$) are plotted in Figure 2, where IPA exhibits the fastest learning convergence of the three methods on all three environments. Except for cFVI on the pendulum benchmark, IPA also delivers the lowest cost and variance. The average cost (closer to 0 is better) of the tested methods is evaluated on the three CT-RL environments in Table 2, each at three levels of modeling error $\nu = 0\%, 10\%, 25\%$. Success rates are summarized in Table 3 for evaluations of the three environments and different tasks under three levels of modeling error. See Table 5 of Appendix G for definitions of task criteria.

Examining the performance in Figure 2 and Table 3, overall the three methods (IPA, cFVI, rFVI) perform quite comparably on the baseline SOS environment. However, examining the closed-loop performance in the pendulum environment swinging up from full free-hanging pendulum displacement ($\theta_0 = 180°$) in Figure 3, one can see that the IPA response is the fastest of the three methods and is well-behaved with no overshoot. This hence points to a trade-off between lower cost (achieved by cFVI) and faster swing-up performance (achieved by IPA). Finally, IPA exhibits a clear learning performance edge over FVIs on the HSV environment, with the fastest convergence, lowest cost, and 100% evaluation success rate on all tasks (Figure 2, Table 3).

**Q2: IPA is 6 orders of magnitude more data efficient and 3 orders of magnitude more time efficient than FVIs.** Due to IPA's low-dimension quadratic bases enabled by our choice of quadratic Q-R cost, it achieves substantial learning efficiency compared to the FVIs training deep networks. Table 1 lists key time/data complexity parameters for IPA and the FVIs. In many cases, IPA exhibits orders of magnitude less time/data complexity than the SOTA FVI works. To illustrate, we examine the ratio of IPA/FVI on the HSV model. For number of data samples required: 1/5,750,000, training episodes required: IPA/FVI = 1/5,250,000, training time: 1/1,500, algorithm iterations: 1/5.

Table 1: Comparison of time/data complexity of IPA and FVIs on 3 environments

| Parameter | SOS | | Pendulum | | HSV | |
|---|---|---|---|---|---|---|
| | IPA | cFVI/rFVI | IPA | cFVI/rFVI | IPA | cFVI/rFVI |
| # Traj. data samples | 10 | 2.88e+7 | 15 | 8.63e+7/2.88e+7 | 30 | 1.73e+8 |
| # Data episodes | 1 | 6.40e+5 | 1 | 2.63e+6/6.40e+5 | 1 | 5.25e+6 |
| Avg train time (s) | 0.20 | 1.42e+3/1.41e+3 | 0.22 | 1.76e+3/1.46e+3 | 2.75 | 4.60e+3/4.79e+3 |
| # Alg. iterations $i$ | 5 | 25 | 5 | 25 | 10 | 50 |

**Q3: IPA learning generalizes the best on all three environments as modeling errors are introduced.** In Table 2, we see that regardless of the environment tested, IPA achieves average cost that meets or exceeds the FVIs when modeling error increases. Examining Table 2 on the HSV at $\nu = 25\%$ modeling error, IPA cost (32.45, a 31% degradation from the nominal model) far surpasses that of cFVI (63.80, a 114% degradation) and rFVI (74.15, a 122% degradation). Turning to the evaluation success rates for the HSV in Table 3, we see that IPA achieves a 100% success rate on all tasks regard-

Table 2: Average cost at varying modeling error levels $\nu$

| $\nu$ | Cost | SOS | Pendulum | HSV |
|---|---|---|---|---|
| 0% | IPA | 0.50±0.65 | 12.12±17.96 | 24.69±36.14 |
| | cFVI | 0.50±0.66 | 8.99±10.04 | 29.79±31.51 |
| | rFVI | 0.50±0.67 | 16.26±17.61 | 33.37±36.47 |
| 10% | IPA | 0.53±0.69 | 8.48±11.85 | 27.19±40.63 |
| | cFVI | 0.53±0.70 | 7.73±8.70 | 36.42±36.49 |
| | rFVI | 0.53±0.70 | 16.85±17.79 | 41.54±42.29 |
| 25% | IPA | 0.58±0.75 | 5.77±6.31 | 32.45±50.38 |
| | cFVI | 0.58±0.78 | 7.01±7.85 | 63.80±44.73 |
| | rFVI | 0.56±0.76 | 23.15±18.07 | 74.15±48.64 |

less of modeling error severity. FVIs exhibit a high success rate on the lax regulation Tasks 1 and 2; however, they struggle with the stringent Tasks 3 and 4 for reasons discussed in Q6 below.

**Q4: FVIs deliver good cost performance results on the nominal model, but IPA generalizability is superior.** Figure 4 shows the relative cost performance $J_{cFVI} - J_{IPA}$ between cFVI and IPA (left two plots) and $J_{rFVI} - J_{IPA}$ between rFVI and IPA (right two plots) on the nominal HSV model $\nu = 0\%$ and at 25% modeling error. Full plots can be found in Figures 9, 10, and 11 of Appendix I on the SOS, pendulum, and HSV, respectively. This data is tabulated in Tables 11, 12, and 13 of Appendix I. On the nominal HSV model (first and third plots of Figure 4), FVIs and IPA perform comparably near the origin, but IPA begins to outperform FVIs by $\approx 10 - 20$ in the majority of the operating region of the states. There are two fringes of the domain (corresponding to large initial flightpath angle (FPA) displacements $\gamma = \pm 1$ deg) in which FVIs slightly outperform IPA; however, in the mean IPA outperforms cFVI by 13.14 and rFVI by 16.45 (cf. Table 13). Cost performance of both FVI algorithms degrades significantly when modeling error is introduced, in particular at the severe 25% modeling error (second and fourth plots of Figure 4). Here, IPA outperforms FVIs pointwise, in the mean outperforming cFVI by 34.10 and rFVI by 35.90.

Table 3: Success rates for different environments and tasks at varying modeling error $\nu = 0\%, 10\%,$ $25\%$ (cf. Table 5 for task criteria)

| $\nu$ | Success [%] | SOS Task 1 | Pendulum Task 1 | HSV Task 1 | HSV Task 2 | HSV Task 3 | HSV Task 4 | HSV Task 5 | HSV Task 6 | HSV Task 7 |
|---|---|---|---|---|---|---|---|---|---|---|
| | IPA | 100 | 100 | 100 | 100 | 100 | 100 | 100 | 100 | 100 |
| 0% | cFVI | 100 | 100 | 100 | 100 | 31.95 | 15.30 | 100 | 3.90 | 3.90 |
| | rFVI | 100 | 100 | 100 | 100 | 19.20 | 8.75 | 100 | 1.25 | 1.25 |
| | IPA | 100 | 100 | 100 | 100 | 100 | 100 | 100 | 100 | 100 |
| 10% | cFVI | 100 | 100 | 100 | 99.50 | 32.70 | 9.95 | 99.50 | 2.35 | 2.35 |
| | rFVI | 100 | 100 | 100 | 100 | 15.50 | 7.70 | 100 | 1.45 | 1.45 |
| | IPA | 100 | 100 | 100 | 100 | 100 | 100 | 100 | 100 | 100 |
| 25% | cFVI | 100 | 100 | 100 | 99.20 | 48.00 | 8.30 | 99.20 | 4.20 | 4.20 |
| | rFVI | 100 | 100 | 100 | 100 | 16.45 | 5.70 | 100 | 1.95 | 1.95 |

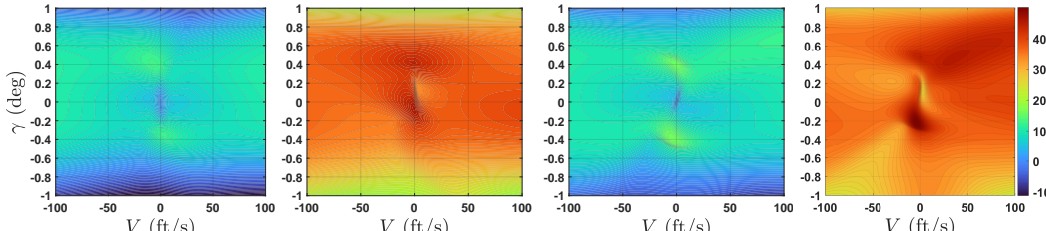

Figure 4: Relative cost performance $J_{\star FVI} - J_{IPA}$ ($\star$ = r or c, $> 0$ means IPA better) on the HSV environment. Left to right: cFVI 0% uncertainty, cFVI 25% uncertainty, rFVI 0% uncertainty, rFVI 25% uncertainty. FVIs exhibit slightly higher cost on average than IPA on the nominal model, and the performance gap increases significantly when environment uncertainty is introduced.

**Q5: IPA exhibits more consistent critic approximation behavior as environment uncertainty is introduced.** Figure 5 shows the critic network error $J - \hat{V}$ for IPA (left column), cFVI (middle column), and rFVI (right column) on the nominal HSV model $\nu = 0\%$ (top row) and at 25% modeling error (bottom row). Full plots can be found in Figures 12, 13, and 14 of Appendix I on the SOS, pendulum, and HSV, respectively. On the nominal HSV model $\nu = 0\%$ (left column of Figure 14), one can see that both cFVI and rFVI struggle with overestimation in the band $\gamma \in [-0.5°, 0.5°]$, in particular for large velocity displacements $V = \pm 100$ ft/s. IPA approximation error remains lower and more consistent in this region. Furthermore, rFVI begins to underestimate policy performance for large FPA displacements $\gamma = \pm 1°$, where cFVI performs comparably better. Overall, approximation performance significantly degrades for FVIs. Here, both FVI algorithms begin to overestimate policy performance to a large degree. Where rFVI begins underestimates policy performance on the nominal model near $\gamma = \pm 1°$, it performs comparably better than cFVI at 25% modeling error. This suggests that rFVI's adversarial input successfully improves generalization properties of this robust variant somewhat. By comparison, IPA's approximation error remains at comparable levels and in a similar radial pattern to its performance on the nominal model.

**Q6: IPA exhibits the fastest closed-loop performance with least overshoot on all environments, performance advantage retained with modeling error.** Figure 3 plots the closed-loop time responses of the tested methods on the nominal model for 1) the pendulum swing-up task (left), 2) HSV 100 ft/s step velocity command issued from trim (middle), and 3) HSV 1° step FPA command issued from trim (right). Full plots for the SOS, pendulum, and HSV can be found in Figures 15, 16, and 17 of Appendix I, respectively. The trends are similar as modeling error is introduced. On the pendulum, we can see that the IPA swing-up is more responsive than either of the FVI methods. For the HSV velocity response (middle plot of Figure 3), we can see that both FVI algorithms have large overshoot compared to IPA. Furthermore, the FVI responses exhibit pronounced and slow-decaying transients about the 100 ft/s reference command setpoint. Similar transients are seen in the HSV FPA response for the two FVIs (right plot of Figure 3). Such oscillations in FPA (i.e., in deflection the vehicle velocity vector) are highly undesirable in flight control settings, as they cause wear on the fuselage and may excite the HSV flexible modes (Bolender & Doman, 2006a). This time-

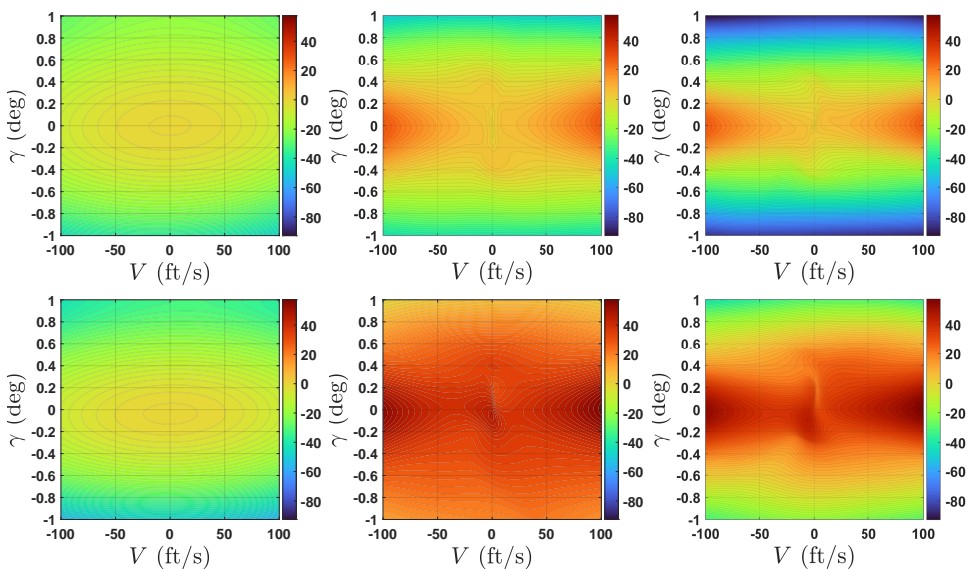

Figure 5: Critic NN approximation error $J(x) - \hat{V}(x)$ of nominal HSV model (top row) and at 25% modeling error (bottom row) for IPA (left), cFVI (middle), and rFVI (right). IPA exhibits consistent approximation performance, and the FVIs struggle to generalize.

domain behavior is also a clear visual indicator of why FVIs struggle with evaluation success rate in the more stringent HSV regulation Tasks 3 and 4 (cf. Table 3). By comparison, IPA exhibits less overshoot, faster settling time, and a well-behaved system response without oscillatory transients.

## 6 CONCLUSION AND DISCUSSION

We introduce IPA, a new, model-based CT-RL control method which innovatively combines a partially-known affine nonlinear dynamic model of the actual nonlinear environment, state-action trajectory data, Kleinman control structures, and reference control input for learning exploration. These give our IPA method significant data efficiency, as we have summarized in comparison to the FVIs in Table 1. When compared to the FVIs, IPA offers substantial theoretical guarantees, and its learning performance at least matches, and often outperforms, the SOTA FVIs in terms of average return, evaluation success rate, critic network approximation accuracy, closed-loop time-domain performance, and generalization to unmodeled dynamics. The IPA performance advantage observed on the HSV control task demonstrates that IPA achieves SOTA results in both the domains of learning and dynamical control. Finally, we would like to point out the need of spending extra time to tune the FVIs for convergence on the HSV environment (for detailed discussions and implementation, see Remark H.1 of Appendix H).

**Limitations of this Study.** The theoretical guarantees and design of IPA are based on an affine nonlinear model, not the general nonlinear systems as in some recent works (Yildiz et al., 2021; Sandoval et al., 2023). However, as discussed in Section 1, fully-nonlinear algorithms are at an early stage. Comprehensive theoretical results and meaningful designs without stringent assumptions are still under development. As a result, few methods have synthesized meaningful controllers. For example, the ADP CT-VI for general nonlinear systems (Appendix M) fails to synthesize for simple second-order systems with known closed-form solutions (Wallace & Si, 2024).

**Reproducibility Statement.** All IPA code and all datasets for this study are available in Supplemental and at (Wallace & Si, 2025). All FVI results (Lutter et al., 2021; 2023b) are generated by the open-source code developed by the authors available at Lutter et al. (2023a). For an in-depth discussion of our setup and a complete list of numerical hyperparameter selections, see Section 4 and Appendix H. All theoretical assumptions can be found in our method setup of Section 2, and all proofs of theoretical results can be found in Appendix D.

ACKNOWLEDGMENTS

This work was supported in part by the NSF under Grants 1808752 and 2211740. The work of Brent A. Wallace was supported in part by the NSF under Graduate Research Fellowship Grant 026257-001.

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

## APPENDIX A  ENVIRONMENTS STUDIED AND KEY THEORETICAL ASSUMPTIONS REQUIRED BY LEADING AND SOTA CT-RL WORKS

We provide an overview of the environments studied in the evaluations of the leading and SOTA CT-RL works in Table 4 below as discussed in Section 4.

Table 4: Environments in SOTA CT-RL evaluations

| Alg | System | Order | # inputs | Source of model parameters |
|---|---|---|---|---|
| IPA | SOS | $\longrightarrow$ | $\longrightarrow$ | Identical to SPI as benchmark |
| | Pendulum | $\longrightarrow$ | $\longrightarrow$ | Identical to FVIs as benchmark |
| | HSV | 5 | 2 | NASA Langley aeropropulsive data (Shaughnessy et al., 1990) Unstable, nonminimum phase, complex |
| FVIs | Pendulum | 2 | 1 | Quanser STEM curriculum |
| | Cart Pendulum | 4 | 1 | resources (Quanser, 2018) |
| | Furatura Pendulum | 4 | 1 | |
| IRL | SOS | 2 | 1 | Non-physical, optimal known *a priori* |
| | SOS | 2 | 1 | Non-physical, optimal known *a priori* |
| SPI | Simple Linear | 3 | 1 | Non-physical LQR example |
| | SOS | 2 | 1 | Non-physical, optimal known *a priori* |
| RADP | Simple Engine | 2 | 1 | Non-physical for illustration |
| | Simple Power Bus | 2 | 1 | Non-physical for illustration |
| CT-VI | SOS | 2 | 1 | Non-physical, optimal known *a priori* |
| | Simple Robot Arm | 4 | 2 | Non-physical for illustration |

**Theoretical Assumptions Required by Leading CT-RL Works.** As shown below, IPA is among the least restrictive in CT-RL in its theoretical assumptions. As a note, all methods require that be Lipschitz near origin to assure well-posedness of solutions to the system differential equations. Also note that none of the ADP designs resulted in meaningful controllers, please refer to (Wallace & Si, 2024).

- IPA (present work):
  - Linearization of nonlinear system $(A, B)$ stabilizable and $(Q^{1/2}, A)$ detectable (for well-posedness and definiteness of regulation problem)
  - Full column rank $\underline{n} \triangleq \frac{n(n+1)}{2}$ of the IPA matrix $\mathcal{I} \in \mathbb{R}^{l \times \underline{n}}$ (20). This assumption is easy to satisfy, as the algorithm can continue to collect $l > \underline{n}$ samples until it is met. It is also virtually instantaneous to verify (matrix rank calculation). We observe no issue satisfying this assumption on the complex HSV in practice
  - Initial stabilizing nonlinear policy $\mu_0$

- FVIs (Lutter et al., 2021; 2023b):
  - $f$ and $g$ are smooth in their partial derivatives in the state $x$ and model uncertainty parameters $\theta$, and these partials are all known *a priori*
  - Undiscounted problem $\gamma = 1$ can be approximated by discounted problem $0 < \gamma < 1$
  - Discrete-time running cost $r(x, u)$ can be approximated by continuous-time counterpart: $r(x, u) = \Delta t \, r_c(x, u)$ with sample time $\Delta t$
  - Strict convexity of action penalty $g_c$
  - Availability of convex conjugate function to action penalty $g_c$
  - Higher-order terms in Taylor series expansion of optimal value $V^*$ are negligible
  - Existence of an *a priori* state grid $x \in \mathcal{D}$ to contain trajectories to for fitting procedure
  - Trajectories leaving the grid $x \in \mathcal{D}$ can be instantaneously re-initialized to the previous position inside the grid

- IRL (Vrabie & Lewis, 2009):

  - There exists a sequence of sampling instants $t_0 < t_1 < \cdots < t_l$ such that the IRL regression matrix has full rank. This assumption is qualitatively similar to IPA, but the method does not lead to meaningful controllers in practice, as there is no constructive method to ensure the full rank condition (Wallace & Si, 2024)
  - Chosen basis functions approximate optimal value and its gradient uniformly on compact sets
  - Basis functions for critic network are linearly-independent
  - Initial stabilizing policy

- SPI (Vamvoudakis & Lewis, 2010):

  - Existence and uniqueness of least-squares solution to approximate HJB equation
  - PE assumption on various learning signals
  - Chosen basis functions approximate optimal value and its gradient uniformly on compact sets
  - Chosen basis functions approximate optimal policy uniformly on compact sets
  - Basis functions for critic network are linearly-independent
  - Basis functions for actor network are linearly-independent
  - Initial stabilizing policy

- RADP (Jiang & Jiang, 2014):

  - Optimal value can be bounded from above and below by *a priori* known class $\mathcal{K}_\infty$ functions
  - Existence of *a priori* known compact set $\Omega_0$ for which the closed-loop system under the initial policy is invariant with respect to the probing noise $d$
  - PE assumption on various learning signals
  - Chosen basis functions approximate optimal value and its gradient uniformly on compact sets
  - Chosen basis functions approximate optimal policy uniformly on compact sets
  - Basis functions for critic network are linearly-independent
  - Basis functions for actor network are linearly-independent
  - Initial stabilizing policy

- CT-VI (Bian & Jiang, 2022):

  - Existence and uniqueness of solutions to an uncountable family of finite-horizon HJB equations
  - Properness of each solution to the finite-horizon HJB equation
  - Convergence of family of solutions of finite-horizon HJB equation to the infinite-horizon HJB solution
  - Invariance of closed-loop state/action trajectory to compact set with respect to the probing noise $d$
  - Initial *globally asymptotically stabilizing* policy
  - PE assumption on various learning signals
  - Chosen basis functions approximate optimal value and its gradient uniformly on compact sets
  - Chosen basis functions approximate optimal policy uniformly on compact sets
  - Chosen basis functions approximate optimal Hamiltonian uniformly on compact sets
  - Basis functions for critic network are linearly-independent
  - Basis functions for actor network are linearly-independent
  - Basis functions for Hamiltonian network are linearly-independent

## APPENDIX B  KLEINMAN'S ALGORITHM FOR LINEAR SYSTEMS (KLEINMAN, 1968)

We adapt some successive approximation concepts from Kleinman's algorithm to the proposed non-linear IPA algorithm for data efficiency. Classical Kleinman's algorithm considers the linear time-invariant system $\dot{x} = Ax + Bu$. We assume that $(A, B)$ is stabilizable and $(Q^{1/2}, A)$ is detectable, for well-posedness (Rodriguez, 2004). Kleinman's algorithm iteratively solves for the optimal LQR control $K^* = R^{-1}B^T P^*$, where $P^* \in \mathbb{R}^{n \times n}$, $P^* = P^{*T} > 0$ is the solution of the Riccati equation (Rodriguez, 2004), as follows. For iteration $i = 0, 1, \dots$, on the current policy $K_i$, let $P_i \in \mathbb{R}^{n \times n}$, $P_i = P_i^T > 0$ be the solution of the algebraic Lyapunov equation (ALE)

$$(A - BK_i)^T P_i + P_i(A - BK_i) + K_i^T R K_i + Q = 0. \tag{21}$$

Then, $P_i$ solved from (21) leads to the new policy $K_{i+1} \in \mathbb{R}^{m \times n}$ as

$$K_{i+1} = R^{-1}B^T P_i. \tag{22}$$

The following theorem is needed to prove the theoretical results of Section 3.

**Theorem B.1 (Convergence, Optimality, and Closed-Loop Stability of Kleinman's Algorithm (Kleinman, 1968))** Suppose the initial policy $K_0$ is such that $A - BK_0$ is Hurwitz. Then we have the following:

(i)  $A - BK_i$ is Hurwitz for all $i \geq 0$.

(ii)  $P^* \leq P_{i+1} \leq P_i$ for all $i \geq 0$, and $\lim_{i \to \infty} P_i = P^*$, $\lim_{i \to \infty} K_i = K^*$.

## APPENDIX C  ADDITIONAL DERIVATIONS OF IPA METHOD (SECTION 2)

The following algebraic manipulation of the IPA CT temporal difference equation (15) is necessary in order to 1) cast it into the form for a learning update (16), which requires that all terms pertaining to the value function weights $c_i$ appear as regression parameters, and 2) accommodate when the control input $u(t)$ includes excitation signals. The integral reinforcement (8) and surrounding equations only hold if the control $u(t) = \mu_i(x(t))$ is applied without excitation signals, yet these excitation signals are necessary for good data/learning quality and thus need to be accommodated by the IPA framework. First, we must formally establish some of the properties of the operators $v$ (12) and $\Phi$ (10):

**Proposition C.1** The operators $v$ (12) and $\Phi$ (10) satisfy the following:

(i)  The restriction of $v$ to the symmetric matrices is a linear isomorphism; thus, for each $c \in \mathbb{R}^n$, there exists a unique $P \in \mathbb{R}^{n \times n}$, $P = P^T$ such that

$$c = v(P). \tag{23}$$

(ii)  Whenever $P \in \mathbb{R}^{n \times n}$, $P = P^T$, the following identity holds

$$\Phi^T(x, y)v(P) = x^T P y, \quad \forall\, x, y \in \mathbb{R}^n. \tag{24}$$

**We are now ready to proceed with the derivation.** Beginning with the $i$-th iteration policy $\mu_i$, we first note that as $x \to 0$, we have

$$f(x) + g(x)\mu_i(x) \;\longrightarrow\; \xi(x) \triangleq Ax + B\mu_i(x) \tag{25}$$

at a rate $o(\|x\|)$, where $(A, B)$ denotes the linearization of $(f, g)$ (1). Combining the quadratic value function approximator structure $V(x) = x^T P_i x$ (11) with the GHJB equation (6) implies to the first order that

$$2\xi^T(x)P_i x + x^T Q x + \mu_i^T R \mu_i = 0. \tag{26}$$

Next, we differentiate the value function integral $V(x(t))$ (7) along the trajectories of the nonlinear system, yielding

$$V(x(t_1)) - V(x(t_0)) = \int_{t_0}^{t_1} \frac{d}{d\tau} \{V(x)\} \, d\tau = \int_{t_0}^{t_1} (f(x) + g(x)u)^T \frac{\partial V}{\partial x}(x) \, d\tau. \quad (27)$$

The crux of the CT temporal difference formulation in (27) is that it accommodates arbitrary control signals $u$. In terms of the quadratic bases chosen (11), we can express (27) as

$$x^T(t_1)P_i x(t_1) - x^T(t_0)P_i x(t_0) = 2 \int_{t_0}^{t_1} (f(x) + g(x)u)^T P_i x \, d\tau. \quad (28)$$

Next, we need to re-introduce the integral reinforcement signal $\int_{t_0}^{t_1} x^T Q x + \mu_i^T(x) R \mu_i(x) d\tau$ as it is found in the original CT TD equation (8) back into the CT TD formulation (28) so that IPA (14) may be applied to (28). To do so, we add an integral reinforcement constant based on $\xi$ (25) to both sides of (28):

$$x^T(t_1)P_i x(t_1) - x^T(t_0)P_i x(t_0) - 2 \int_{t_0}^{t_1} \xi^T(x) P_i x \, d\tau = 2 \int_{t_0}^{t_1} (f(x) + g(x)u - \xi(x))^T P_i x \, d\tau. \quad (29)$$

Plugging the approximate GHJB equation (26) into (29) and rearranging, we have

$$-2 \int_{t_0}^{t_1} (f(x) + g(x)u - \xi(x))^T P_i x \, d\tau + \left[ x^T(t_1)P_i x(t_1) - x^T(t_0)P_i x(t_0) \right]$$

$$\approx -\int_{t_0}^{t_1} x^T Q x + \mu_i^T(x) R \mu_i(x) \, d\tau. \quad (30)$$

Finally, by using the identification $c_i = v(P_i)$ and algebraic identities from Proposition 2.1, in particular with $c_i = v(P_i)$ and the bilinear algebra of $\Phi$:

$$x^T(t_1)P_i x(t_1) - x^T(t_0)P_i x(t_0) = [\Phi^T(x(t_1), x(t_1)) - \Phi^T(x(t_0), x(t_0))]c_i$$

$$= \Phi^T \big( x(t_1) + x(t_0), x(t_1) - x(t_0) \big) c_i, \quad (31)$$

then (30) becomes

$$\left[ -2 \int_{t_0}^{t_1} \Phi \big( f(x) + g(x)u - \xi(x), x \big) \, d\tau + \Phi \big( x(t_1) + x(t_0), x(t_1) - x(t_0) \big) \right]^T c_i$$

$$= -\int_{t_0}^{t_1} x^T Q x + \mu_i^T(x) R \mu_i(x) \, d\tau \approx -\int_{t_0}^{t_1} x^T Q x + x^T K_i^T R K_i x \, d\tau. \quad (32)$$

Here, the last approximation in (32) is precisely IPA (14). Equation (32) is of the final form required for a learning update, as all terms pertaining to the value function $V$ via the weights $c_i$ now appear as a regression to update the critic network weights, and the control $u(t)$ has been kept arbitrary so as to accommodate excitation signals $r(t)$ for training. The learning update matrices in (16) now follow directly from applying (32) at the sample instants $\{t_k\}_{k=0}^l$:

$$\Theta_i = \begin{bmatrix} -2 \int_{t_0}^{t_1} \Phi^T \big( f(x) + g(x)u - \xi(x), x \big) \, d\tau \\ +\Phi^T \big( x(t_1) + x(t_0), x(t_1) - x(t_0) \big) \\ \vdots \\ -2 \int_{t_{l-1}}^{t_l} \Phi^T \big( f(x) + g(x)u - \xi(x), x \big) \, d\tau \\ +\Phi^T \big( x(t_l) + x(t_{l-1}), x(t_l) - x(t_{l-1}) \big) \end{bmatrix}, \quad \mathcal{I} \triangleq \begin{bmatrix} \int_{t_0}^{t_1} \Phi^T(x, x) \, d\tau \\ \vdots \\ \int_{t_{l-1}}^{t_l} \Phi^T(x, x) \, d\tau \end{bmatrix} \in \mathbb{R}^{l \times \underline{n}},$$

$$(33)$$

$$\Xi_i = -\mathcal{I} \, v \big( Q + K_i^T R K_i \big). \quad (34)$$

**Remark C.1 (IPA Increases Data Efficiency)** IPA via (33) allows the current policy information to be pulled out of the integral reinforcement signal, enabling re-use of a single trajectory $\{x(t)\}_{t \in [t_0, t_l]}$ from the real environment, which may include nonlinearity, uncertainty, and unmodeled dynamics. This single trajectory can now generate all of the critic weights $\{c_i\}_{i=0}^\infty$ and policy iterates $\{\mu_i\}_{i=0}^\infty$ from a single integration matrix $\mathcal{I} \in \mathbb{R}^{l \times \underline{n}}$, *without* having to re-integrate the integral reinforcement signal at each iteration $i$. For this reason, we call $\mathcal{I}$ (33) the IPA matrix. The IPA matrix has further structural significance to the IPA algorithm; indeed, its full rank forms the basis of our theoretical guarantees (cf. Section 3).

## APPENDIX D    PROOF OF THEOREM 3.1

Suppose that the sample count $l \in \mathbb{N}$ and sample times $\{t_k\}_{k=0}^l$ are such that $\mathcal{I}$ (20) has full rank $\underline{n}$. We now proceed by induction on $i$.

Suppose it has been proven for iteration $i \geq 0$ that $A - BK_i$ is Hurwitz and that $K_0, \ldots, K_i$ agree as generated from Kleinman's algorithm and IPA via (13). We claim that $K_0, \ldots, K_{i+1}$ agree.

We first establish that the least-squares matrix $\Theta_i \in \mathbb{R}^{l \times \underline{n}}$ (16) has full column rank $\underline{n}$. For suppose $v(P) \in \mathbb{R}^{\underline{n}}$ is such that $\Theta_i v(P) = 0$. Examining (32), and proceeding through the derivation in Section 2 and Appendix C, after applying the identity (24) we note for *any* symmetric matrix that $\Theta_i v(P) = \mathcal{I}v(N)$, where $N \in \mathbb{R}^{n \times n}$, $N = N^T$ is given by

$$N = (A - BK_i)^T P + P(A - BK_i). \tag{35}$$

However, (35) is an ALE. Since $N = N^T$ and since it's been established that $A - BK_i$ is Hurwitz, (35) has the unique solution $P = \int_0^\infty e^{(A-BK_i)^T t}(-N)e^{(A-BK_i)t}\, dt$ (Rodriguez, 2004). Now, full rank of $\mathcal{I}$ and that $\mathcal{I}v(N) = 0$ imply $v(N) = 0$, whence $N = 0$. That $N = 0$ now implies $v(P) = 0$ by the above. We have shown that $\Theta_i$ has trivial right null space, hence full column rank $\underline{n}$.

Having established that $\Theta_i$ has full rank, we now claim that $P_i \in \mathbb{R}^{n \times n}$, $P_i = P_i^T > 0$ uniquely solves the ALE (21) if and only if $c_i = v(P_i)$ satisfies the regression (16). The forward direction was already proved in the derivation (8)–(32) of Section 2 and Appendix C. Conversely now, suppose that $v(P) \in \mathbb{R}^{\underline{n}}$ minimizes the regression (16). Since it has been established that $\Theta_i$ has full column rank, $v(P) \in \mathbb{R}^{\underline{n}}$ is unique. Next, letting $P_i = P_i^T > 0$ be the unique solution of the ALE (21), the forward direction establishes that $v(P_i) \in \mathbb{R}^{\underline{n}}$ satisfies (16) at equality. Thus, $v(P) = v(P_i)$, and as a result $P = P_i$ (Proposition C.1). The result is proved.

Having established the preceding, the proof now follows by induction on the algorithm iteration $i = 0, 1, \ldots$. ∎

## APPENDIX E    DECENTRALIZABLE ENVIRONMENT FOR FURTHER DATA EFFICIENCY

Consider a decentralized environment $(f, g)$ (1) with $N$ distinct control loops. To illustrate, we present $N = 2$ loops; however, results readily generalize to $N > 2$ loops:

$$\begin{bmatrix} \dot{x}_1 \\ \dot{x}_2 \end{bmatrix} = \begin{bmatrix} f_1(x) \\ f_2(x) \end{bmatrix} + \begin{bmatrix} g_{11}(x) & g_{12}(x) \\ g_{21}(x) & g_{22}(x) \end{bmatrix} \begin{bmatrix} u_1 \\ u_2 \end{bmatrix}. \tag{36}$$

No assumptions are made on dynamic coupling between the loops; i.e., the loops may be coupled. Here, we define $x_j \in \mathbb{R}^{n_j}$, $u_j \in \mathbb{R}^{m_j}$ $(j = 1, \ldots, N)$ with $\sum_{j=1}^N n_j = n$ and $\sum_{j=1}^N m_j = m$.

Such partitions appear in a variety of real-world applications, in particular, for the aircraft control problem studied (Stengel, 2022). In fact, decentralization is a standard practice in control of HSVs (Dickeson et al., 2009a;b), so we refer the reader to these references for further details. Furthermore, such partitions can be found in robotics (Craig, 2005; Dhaouadi & Abu Hatab, 2013), helicopters (Enns & Si, 2002; 2003b;a), and UAVs (Wang et al., 2016). In this case, the IPA learning (16) occurs in a decentralized fashion in each of the loops, thus reducing problem dimensionality. Note that the IPA method holds for general affine nonlinear systems (1) without decentralization. If the physics of the environment permit decentralization, it only helps further improve solution efficiency.

## APPENDIX F    REFERENCE COMMAND INPUT AND IPA ROBUSTNESS

### F.1    REFERENCE COMMAND INPUT

Many SOTA ADP CT-RL algorithms require the persistence of excitation (PE) condition in proofs of algorithm properties (Wallace & Si, 2024). PE is an analytical condition, but it is not constructive, nor is there a verification procedure for PE. In essence, PE is in alignment with the notion of exploration to enable learning and convergence. To achieve PE, it is standard practice to apply a

probing noise $d$ to the system (1) in a feedback control of the form $u = \mu(x) + d$. According to classical feedback control principles, a good feedback control attenuates plant input disturbances; thus, plant-input probing noise excitation is an inherently problematic practice (Rodriguez, 2004). Its consequence and insights on why such excitation signals may cause learning issues are thoroughly analyzed in (Wallace & Si, 2024). We instead propose a reference command input solution to excite the closed-loop system at the favorable reference command $r$. Critically, reference command input is compatible with current RL formulations and is of the required form $u = \mu(x) + d$. Thus, reference command input can improve learning of existing CT-RL algorithms. We have included a block diagram illustrating reference command input in a standard negative feedback structure (Rodriguez, 2004) in Figure 6. Standard in ADP CT-RL is to apply the control $u = \mu(x) + d$ as shown in Figure 6 (i.e., with no reference command $r(t) \equiv 0$ and output $y = x$). Instead, we form an error signal $e = r - y$ from a user-designed reference command input $r$ in order to improve excitation properties.

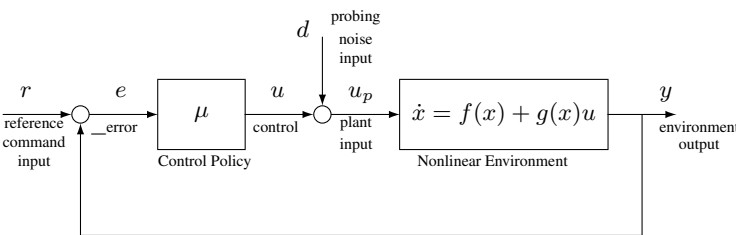

Figure 6: Reference command input in standard negative feedback structure.

To illustrate the issue with inserting probing noise $d$ alone, we first define a few closed-loop maps.

**Definition F.1 (Closed-Loop Maps (Rodriguez, 2004))** Examining Figure 6, we define:

- $S_u \triangleq T_{d \to u_p}$ (sensitivity at the controls $u$): The closed-loop map from plant input disturbance $d$ to control $u$.

- $T_u \triangleq T_{d \to u}$ (complementary sensitivity at the controls $u$): The closed-loop map from plant input disturbance $d$ to plant input $u_p$.

- $S_e \triangleq T_{r \to e}$ (sensitivity at the error $e$): The closed-loop map from reference command $r$ to error $e$.

- $T_e \triangleq T_{r \to y}$ (complementary sensitivity at the error $e$): The closed-loop map from reference command $r$ to plant output $y$.

- $PS_u \triangleq T_{d \to y}$ (P-sensitivity): The closed-loop map from plant input disturbance $d$ to plant output $y$.

To illustrate typical input/output behavior, Figure 7 shows these two closed-loop frequency responses for the HSV. Let us examine the frequency response in loop $j = 2$ of the HSV (associated with flightpath angle $y_2 = \gamma$, yellow dashed curve). Since probing noise is inserted at the *plant input*, the effective closed-loop map from probing noise $d$ to output $y$ is the $P$-sensitivity $T_{d \to y}$. Thus, one glance at the yellow dashed SISO $T_{d \to y}$ response in Figure 7a reveals issues: Any probing noise will be attenuated by at least $-25$ dB (a factor of about 20) at best-case. Furthermore, any probing noise frequency content below $10^{-1}$ rad/s and above 2.5 rad/s will be attenuated by more than $-40$ dB, or a factor of 100. In light of this simple analysis, it is clear that achieving sufficient system excitation via probing noise injection alone proves to be a significant issue for learning the nonlinear flightpath angle dynamics.

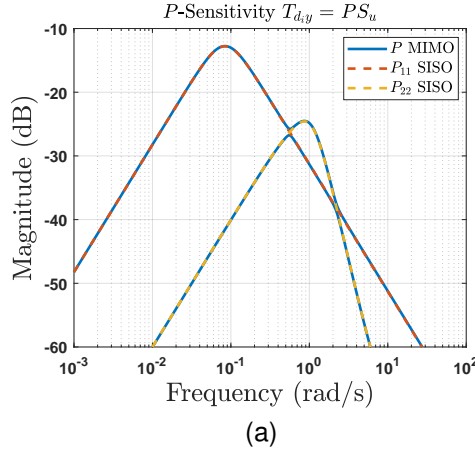 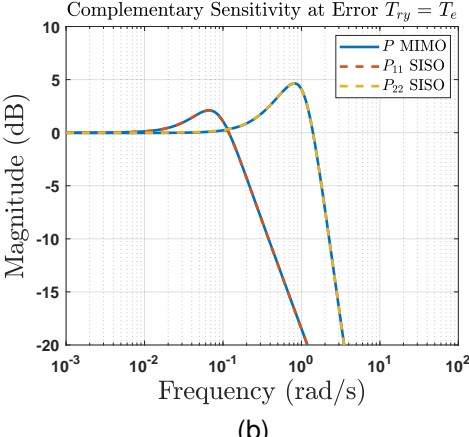

Figure 7: Closed-loop frequency responses: The probing noise injection issue visualized. (a): $P$-sensitivity $T_{d\to y}$. (b): Complementary sensitivity $T_{r\to y}$.

### F.2 IPA Robustness

Theorem 3.1 shows that IPA inherits the guaranteed stability robustness margins of Kleinman's algorithm. These are, repeated here:

- $\|S_u\|_{\mathcal{H}^\infty} \leq 0$ dB,
- $\|T_u\|_{\mathcal{H}^\infty} \leq 2$ dB,

where $\|\cdot\|_{\mathcal{H}^\infty}$ denotes the $\mathcal{H}^\infty$ norm, and where $S_u$, $T_u$ denote the sensitivity and complementary sensitivity closed-loop maps at the controls $u$, respectively (see Definition F.1). These stability robustness margins are substantial, as they offer theoretical guarantees of stability to disturbances inserted in the feedback loop in Figure 6.

For example, let $P$ denote the plant map $u_p \to y$ in Figure 6, $P_0$ be the nominal model of the actual plant $P$, and consider a model uncertainty $\Delta$ of the nominal model $P_0$ satisfying $\|\Delta\|_{\mathcal{H}^\infty} \leq M$ for some $M > 0$, but which is otherwise arbitrary. Then given each of the following uncertainty models, no such uncertainties $\Delta$ can destabilize the closed-loop system under the IPA policy provided the following stability robustness test holds:

| Uncertainty Type | Uncertainty Model | Stability Robustness Test |
|---|---|---|
| Multiplicative | $P = P_0[I + \Delta]$ | $\|T_u\Delta\|_{\mathcal{H}^\infty} < \frac{1}{M}$ |
| Divisive | $P = P_0[I + \Delta]^{-1}$ | $\|S_u\Delta\|_{\mathcal{H}^\infty} < \frac{1}{M}$ |

owing to the small gain theorem (Rodriguez, 2004).

### APPENDIX G PERFORMANCE MEASURES

In these studies, we provide comprehensive evaluations of standard performance measures in: average cost, learning success rate, and generalization with respect to system ICs and modeling error. In addition, we analyze performance with respect to the following learning control measures:

- Cost Performance: The infinite-horizon cost $J(x_0)$ obtained via the integral (2) delivered by the policy in the nonlinear optimal control task. As a note, in controls conventions the cost $J(x_0) > 0$ is a positive number to be minimized (lower is better).
- Relative Cost Performance: The difference $J_{\star FVI}(x) - J_{IPA}(x)$ between the cost performance of the respective FVI algorithm (i.e., $\star = $ c or r) and IPA (Note: $> 0$ means IPA is better; i.e., lower cost).
- Estimation Error: The difference $J(x) - \hat{V}(x)$ between the cost $J(x)$ (2) and the value function approximation $\hat{V}(x)$ (11) at $x \in \mathbb{R}^n$.

**Learning Trial Success.** In this work, we study 9 different closed-loop evaluation tasks on the three environments tested. A policy achieves a "success" in each of these tasks if its closed-loop time response satisfies the respective criteria defined in Table 5.

Table 5: Definitions of closed-loop evaluation tasks

| Task | Description | Definition |
|---|---|---|
| SOS Task 1 | State $x_1$ regulation | $x_1(t)$ within $\pm$ 0.1 at $t = 5$ s |
| Pendulum Task 1 | Pendulum angle $\theta$ regulation | $\theta(t)$ within $\pm$ 5 deg at $t = 5$ s |
| HSV Task 1 | Lax velocity $V$ regulation | $V(t)$ within $\pm$ 10 ft/s at $t = 50$ s |
| HSV Task 2 | Lax FPA $\gamma$ regulation | $\gamma(t)$ within $\pm$ 0.1 deg at $t = 50$ s |
| HSV Task 3 | Strict velocity $V$ regulation | $V(t)$ within $\pm$ 1 ft/s at $t = 25$ s |
| HSV Task 4 | Strict FPA $\gamma$ regulation | $\gamma(t)$ within $\pm$ 0.01 deg at $t = 25$ s |
| HSV Task 5 | Lax $V$ and $\gamma$ regulation | HSV Tasks 1 and 2 satisfied |
| HSV Task 6 | Strict $V$ and $\gamma$ regulation | HSV Tasks 3 and 4 satisfied |
| HSV Task 7 | Lax and strict $V$ and $\gamma$ regulation | HSV Tasks 1, 2, 3, and 4 satisfied |

## APPENDIX H    ADDITIONAL IMPLEMENTATION DETAILS

**Hardware.** We use PyTorch 1.13.1 for FVI implementations, and MATLAB R2022b for IPA implementations. All results are obtained on an NVIDIA RTX 2060, Intel i7 (9th Gen) processor.

**Software: All Code/Data Available.** All IPA and FVI code and datasets developed for this work is available in Supplemental and at (Wallace & Si, 2025). FVI results (Lutter et al., 2021; 2023b) were generated from the open-source repository developed by the authors (Lutter et al., 2023a) implemented on the three environments studied.

### H.1    ADDITIONAL TRAINING/EVALUATION PROCEDURE DETAILS

This section provides additional implementation details for the training/evaluation procedures discussed in Section 4.

**Training Procedure.** An episode is initialized by resetting the environment and terminated at time $T$ of the training horizon for collecting the state-action trajectory data $(x, u)$. A trial is a complete training process that contains a series of consecutive episodes.

IPA learning requires state-action trajectory data from a single episode which usually has on the order of $l = 100$ total samples for the three evaluated environments. This low data complexity allows IPA to learn online from the actual physical process. Deep RL FVIs require training data from over 1 million episodes (cf. Table 1), for details see (Lutter et al., 2021; 2023b). As a result, the only practical means of training FVI is in simulation. Since the modeling error $\nu$ for a given system is not known *a priori*, this means that FVI must train on the nominal model (modulo adversary perturbations in the rFVI case (Lutter et al., 2023b)).

**Random Seeds.** Training and evaluation for each of the methods are based on 20 seeds for random number generation (RNG): 0–19 for training, and 100–119 for evaluation. In the case of the FVIs, the seeds are used in the environment, Numpy, and PyTorch for number generation. In the case of IPA, we have set MATLAB's master RNG seed for number generation.

**Network Weight Initialization.** For FVIs' deep networks, we use the identical network initialization procedure as in the original works (Lutter et al., 2021; 2023b); namely, Xavier normal distribution. The initialization gains of the layers used on each system can be found in Table 8 of Appendix H.2, the same as in previous studies (Lutter et al., 2021; 2023b).

For IPA's simpler quadratic network structure, we need only initialize the critic weights $c_0$ in (11). We initialize each element of the critic weight vector $c_0 \in \mathbb{R}^l$ in the following uniform distributions

for the SOS, pendulum, and HSV, respectively:

$$c_0 \sim \mathcal{U}(-15, 15), \tag{37}$$

$$c_0 \sim \mathcal{U}(-25, 25), \tag{38}$$

$$c_0 \sim \begin{cases} \mathcal{U}(-30, 30), & \text{for weights in translational loop } j = 1 \\ \mathcal{U}(-10, 10), & \text{for weights in rotational loop } j = 2 \end{cases}. \tag{39}$$

**Initializing the Environments.** System initial conditions (ICs) for training and evaluation are generated using uniform distributions $\mathcal{U}$, where the ranges for the SOS, pendulum, and HSV cover the dynamics broadly, well beyond their linear regimes.

**System Initial Condition Generation – Training.** System ICs for training and evaluation are generated using uniform distributions $\mathcal{U}$, where the ranges for the SOS, pendulum, and HSV cover the dynamics broadly, well beyond their linear regimes. We use the following uniform distributions $\mathcal{U}$ for the SOS, pendulum, and HSV, respectively:

$$x_0 \sim \mathcal{U}(\pm 1, \pm 1), \tag{40}$$

$$x_0 \sim \mathcal{U}(\pm \pi/2 \text{ rad}, \pm \pi \text{ rad/s}), \tag{41}$$

$$x_0 \sim \mathcal{U}(\pm 150 \text{ ft/s}, \pm 1.5 \text{ deg}, \pm 5 \text{ deg}, \pm 5 \text{ deg/s}, \pm 0.01 \text{ ft}). \tag{42}$$

where these distributions are centered about the respective system equilibrium point $x_e$ (cf. Appendices J–L for discussion of equilibria of each system).

**System Initial Condition Generation – Evaluation.** For the learning curves plotted in Figure 2, at each algorithm iteration the return of the trained policies is evaluated over 100 episodes of the environment. For evaluation, system ICs for training are generated via the following uniform distributions $\mathcal{U}$ for the SOS, pendulum, and HSV, respectively:

$$x_0 \sim \mathcal{U}(\pm 1, \pm 1), \tag{43}$$

$$x_0 \sim \mathcal{U}(\pm \pi \text{ rad}, \pm \pi/4 \text{ rad/s}), \tag{44}$$

$$x_0 \sim \mathcal{U}(\pm 100 \text{ ft/s}, \pm 1 \text{ deg}, \pm 0.01 \text{ deg}, \pm 0.01 \text{ deg/s}, \pm 0.01 \text{ ft}). \tag{45}$$

For display purposes of generating the surface plots in Figures 4 and 5, we evaluate the final polices of a single trial for each method over the following evaluation grids $x \in G_x$ for the SOS, pendulum, and HSV, respectively:

$$G_x = \texttt{linspace}(-1, 1, 150) \times \texttt{linspace}(-1, 1, 150), \tag{46}$$

$$G_x = \texttt{linspace}(-\pi, \pi, 150) \text{ rad} \times \texttt{linspace}(-\pi/4, \pi/4, 150) \text{ rad/s}, \tag{47}$$

$$G_x = \texttt{linspace}(-100, 100, 150) \text{ ft/s} \times \texttt{linspace}(-1, 1, 150) \text{ deg/s}, \tag{48}$$

where all other ICs on the higher-order HSV environment are set to their equilibrium values $x_e$. Note that we choose the bounds of these grids the same as the bounds of the respective evaluation IC distributions $\mathcal{U}$ in (43)–(45). This is done for illustration purposes to display the policy performance over the evaluation regime. It is these grids which are used to generate the surface plots of Figures 4 and 5 of the manuscript.

**Modeling Error Generation.** In the modeling error generalization studies of Section 5, modeling error $\nu$ is tested at three severity levels: $\nu = 0\%$ (corresponding to the nominal model), $\nu = 10\%$, and $\nu = 25\%$ modeling error. Please see Appendices J–L for an in-depth discussion of the modeling error parameters $\nu$ studied for each system. These are summarized in Table 6.

Table 6: Modeling error parameters studied

| Environment | Parameter | Equation Reference |
|---|---|---|
| SOS | $\dot{x}_2$ unstable nonlinearity $\psi(x)$ | (57) |
| Pendulum | Pendulum length $L$ | (59) |
| HSV | Lift coefficient $C_L$ | (65) |

## H.2 HYPERPARAMETER SELECTIONS

### H.2.1 SHARED HYPERPARAMETERS

For the SOS, we use identical penalty selections to those in the original ADP study (Vamvoudakis & Lewis, 2010); namely,

$$Q_1 = I_2, \qquad\qquad R_1 = 1. \qquad\qquad (49)$$

For the pendulum, we use identical penalty selections to those in the original FVI studies (Lutter et al., 2021; 2023b); namely,

$$Q_1 = \texttt{diag}(1, 0.1), \qquad\qquad R_1 = 0.5. \qquad\qquad (50)$$

For the HSV, consider the decentralized design framework described in Appendix L.2. We choose the following cost structure

$$
\begin{aligned}
Q_1 &= \texttt{diag}(2, 2), & R_1 &= 2.5, \\
Q_2 &= \texttt{diag}(2.5, 5, 0.05, 0), & R_2 &= 1.
\end{aligned}
\qquad (51)
$$

### H.2.2 IPA

For sake of illustration, on IPA we use simple LQ initial policies $\mu_0$. We make the note that, of course, IPA accommodates nonlinear initial policies $\mu_0$. However, to illustrate the practicality of our implementations for real-world design problems, we choose LQ initial policies as a natural designer first-choice. For the SOS, we use the initial stabilizing policy

$$\mu_0(x) = -K_0 x, \qquad K_0 = [\; 3.1952 \quad 6.7099 \;], \qquad\qquad (52)$$

which we obtained from cost structure selections $Q_1 = \texttt{diag}(0.5, 0.25)$, and $R_1 = 0.01$. For the pendulum, we use the initial stabilizing policy

$$\mu_0(x) = -K_0 x, \qquad K_0 = [\; 13.5108 \quad 5.8316 \;], \qquad\qquad (53)$$

which we obtained from cost structure selections $Q_1 = \texttt{diag}(0.5, 0.25)$, and $R_1 = 0.01$. For the HSV in loop $j$ ($j = 1, 2$), we use the initial stabilizing policies

$$
\begin{aligned}
\mu_{0,1}(x_1) &= -K_{0,1} x_1, & K_{0,1} &= [\; 0.8944 \quad 2.7117 \;], & (54) \\
\mu_{0,2}(x_2) &= -K_{0,2} x_2, & K_{0,2} &= [\; 1.5811 \quad 6.7007 \quad 0.8837 \quad 0.7135 \;], & (55)
\end{aligned}
$$

which we obtained from a decentralized design with cost structure selections (51). The remainder of the IPA hyperparameter selections can be found in Table 7.

Table 7: IPA hyperparameter selections

| Hyperparameter | SOS | Pendulum | HSV | |
|---|---|---|---|---|
| | Loop $j = 1$ | Loop $j = 1$ | Loop $j = 1$ | Loop $j = 2$ |
| Sample Period $T_{s,j}$ (s) | 0.1 | 1 | 6 | 2 |
| Number of Samples $l_j$ | 10 | 15 | 15 | 15 |
| Final Iteration $i_j^*$ | 5 | 5 | 10 | 10 |
| Ref Cmd $r_j$ | $\sin(5t)$ | $10\sin(\frac{2\pi}{10}t)$ | $50\sin(\frac{2\pi}{100}t)$ | $0.25\sin(\frac{2\pi}{100}t)$ |
| ([-] \| deg \| ft/s, deg) | | $+5\sin(\frac{2\pi}{5}t)$ | $+5\sin(\frac{2\pi}{25}t)$ | $+0.5\sin(\frac{2\pi}{15}t)$ |
| | | | $+5\cos(\frac{2\pi}{10}t)$ | $+0.03\cos(\frac{2\pi}{6}t)$ |
| Initial Policy $\mu_{0,j}$ | (52) | (53) | (54) | (55) |

### H.2.3 CFVI, RFVI

Hyperparameter selections for cFVI and rFVI can be found in Table 8. These parameter selections are overall quite standard and have indeed demonstrated great learning performance successes on second-order, unstable systems in previous studies (Lutter et al., 2021; 2023b).

**Remark H.1 (FVI Implementation Challenges on HSV System)** The FVIs hyperparameter selections for the HSV in Table 8 are a result of a search over the hyperparameter space in order to achieve the best policy performance possible. This process took approximately two weeks of trial and error. We find that FVIs struggle to converge on the HSV environment, likely due to its higher order, complex nonlinear structure, and its unstable, nonminimum-phase dynamics. In order to achieve FVI convergence, we found it necessary to initialize the weights to a policy trained on 40 iterations with the same hyperparameters given in Table 8, except at a lower learning rate of 1e-7. In order to make IPA's learning procedure more analogous to FVI's on the HSV, we thus center its uniform initial weight distribution (39) about the weights of a nominal LQR policy performed on the nominal linearized dynamics $(A, B)$.

As with our selections of the pendulum model structure and parameters (cf. Appendix K), for our pendulum studies we have selected hyperparameters identical to those of the original cFVI/rFVI evaluations (Lutter et al., 2021; 2023b), with two exceptions. In (Lutter et al., 2021; 2023b), the authors use a logcos control penalty function scaled so that its curvature at the origin $u = 0$ is $2R$; i.e., so that its curvature agrees with that of a quadratic penalty $u^T R u$. In order to make comparisons consistent across the methods studied, and in order to produce a more widely-applicable performance benchmark for real-world designers, we have decided to apply the standard quadratic control penalty $u^T R u$ for all methods. Likewise, the authors in (Lutter et al., 2021; 2023b) wrap the penalty function of the pendulum angle state to be periodic in $[0, 2\pi)$, a practice which we have dropped for consistency of comparison and generalizability of benchmarking. Finally, due to these changes we observed that more iterations were necessary for rFVI to converge in training the pendulum system (cf. Figure 2), so we increased its iteration count from 100 previously (Lutter et al., 2021; 2023b) to 150 here (cf. Table 8).

Table 8: cFVI, rFVI hyperparameter selections

| Hyperparameter | SOS | | Pendulum | | HSV | |
|---|---|---|---|---|---|---|
| | cFVI | rFVI | cFVI | rFVI | cFVI | rFVI |
| Time Step (s) | 0.008 | 0.008 | 0.008 | 0.008 | 0.04 | 0.04 |
| Time Horizon (s) | 5 | 5 | 5 | 5 | 20 | 20 |
| Discounting $\gamma$ | 0.99 | 0.99 | 0.99 | 0.99 | 0.99 | 0.99 |
| Network Dimension | $[3 \times 96]$ | $[3 \times 96]$ | $[3 \times 96]$ | $[3 \times 96]$ | $[3 \times 96]$ | $[3 \times 96]$ |
| # Ensemble | 4 | 4 | 4 | 4 | 4 | 4 |
| Activation | Tanh | Tanh | Tanh | Tanh | Tanh | Tanh |
| Learning Rate | 1e-6 | 1e-6 | 1e-5 | 1e-5 | 1e-4 | 1e-4 |
| Weight Decay | 1e-6 | 1e-6 | 1e-6 | 1e-6 | 1e-6 | 1e-6 |
| Hidden Layer Gain | 1.41 | 1.41 | 1.41 | 1.41 | 1.41 | 1.41 |
| Output Layer Gain | 1.00 | 1.00 | 1.00 | 1.00 | 1.00 | 1.00 |
| Output Layer Bias | -0.1 | -0.1 | -0.1 | -0.1 | -0.1 | -0.1 |
| Diagonal Softplus Gain $\beta_L$ | 1.0 | 1.0 | 1.0 | 1.0 | 7.5 | 7.5 |
| Batch Size | 128 | 128 | 256 | 128 | 256 | 256 |
| # Batches | 200 | 200 | 200 | 200 | 200 | 200 |
| Eligibility Trace | 0.85 | 0.85 | 0.85 | 0.85 | 0.95 | 0.95 |
| $n$-step Trace Weight | 1e-4 | 1e-4 | 1e-4 | 1e-4 | 1e-3 | 1e-3 |
| # Iterations | 25 | 25 | 25 | 25 | 50 | 50 |
| # Epochs/Iteration | 20 | 20 | 20 | 20 | 20 | 20 |
| State Adversary $\|\xi_x\|_{\max}$ | 0.0 | 0.025 | 0.0 | 0.025 | 0.0 | 0.001 |
| Action Adversary $\|\xi_u\|_{\max}$ | 0.0 | 0.1 | 0.0 | 0.1 | 0.0 | 0.05 |
| Model Adversary $\|\xi_\theta\|_{\max}$ | 0.0 | 0.15 | 0.0 | 0.15 | 0.0 | 0.125 |
| Obs Adversary $\|\xi_o\|_{\max}$ | 0.0 | 0.025 | 0.0 | 0.025 | 0.0 | 0.01 |

## APPENDIX I    EVALUATIONS: QUANTITATIVE COMPARISONS BETWEEN IPA & FVIS

In this section, we examine the effects of modeling error $\nu$ on 1) policy cost performance $J$, 2) critic network approximation error $J - \hat{V}$, and 3) closed-loop performance. All hyperparameter selections can be found in Appendix H.

### I.1    AVERAGE COST AND EVALUATION SUCCESS RATE GENERALIZATION

IPA, cFVI, and rFVI are trained with trajectory data generated by the same training uniform IC distribution for each of the respective environments, and the average return is then evaluated over the respective evaluation uniform IC distribution. These distributions are given in Appendix H. The learning curves are plotted in Figure 2 over 20 training seeds. The corresponding average return and variance is tabulated in Table 9, and the corresponding evaluation success rate in Table 10.

Table 9: Average cost at varying modeling error $\nu = 0\%, 10\%, 25\%$

| $\nu$ | Cost $[\mu \pm 2\sigma]$ | SOS | Pendulum | HSV |
|---|---|---|---|---|
| | IPA | 0.50±0.65 | 12.12±17.96 | 24.69±36.14 |
| 0% | cFVI | 0.50±0.66 | 8.99±10.04 | 29.79±31.51 |
| | rFVI | 0.50±0.67 | 16.26±17.61 | 33.37±36.47 |
| | IPA | 0.53±0.69 | 8.48±11.85 | 27.19±40.63 |
| 10% | cFVI | 0.53±0.70 | 7.73±8.70 | 36.42±36.49 |
| | rFVI | 0.53±0.70 | 16.85±17.79 | 41.54±42.29 |
| | IPA | 0.58±0.75 | 5.77±6.31 | 32.45±50.38 |
| 25% | cFVI | 0.58±0.78 | 7.01±7.85 | 63.80±44.73 |
| | rFVI | 0.56±0.76 | 23.15±18.07 | 74.15±48.64 |

Table 10: Success rates for different environments and tasks at varying modeling error $\nu = 0\%$, 10%, 25% (cf. Table 5 for task criteria)

| $\nu$ | Success [%] | SOS Task 1 | Pendulum Task 1 | HSV Task 1 | HSV Task 2 | HSV Task 3 | HSV Task 4 | HSV Task 5 | HSV Task 6 | HSV Task 7 |
|---|---|---|---|---|---|---|---|---|---|---|
| | IPA | 100 | 100 | 100 | 100 | 100 | 100 | 100 | 100 | 100 |
| 0% | cFVI | 100 | 100 | 100 | 100 | 31.95 | 15.30 | 100 | 3.90 | 3.90 |
| | rFVI | 100 | 100 | 100 | 100 | 19.20 | 8.75 | 100 | 1.25 | 1.25 |
| | IPA | 100 | 100 | 100 | 100 | 100 | 100 | 100 | 100 | 100 |
| 10% | cFVI | 100 | 100 | 100 | 99.50 | 32.70 | 9.95 | 99.50 | 2.35 | 2.35 |
| | rFVI | 100 | 100 | 100 | 100 | 15.50 | 7.70 | 100 | 1.45 | 1.45 |
| | IPA | 100 | 100 | 100 | 100 | 100 | 100 | 100 | 100 | 100 |
| 25% | cFVI | 100 | 100 | 100 | 99.20 | 48.00 | 8.30 | 99.20 | 4.20 | 4.20 |
| | rFVI | 100 | 100 | 100 | 100 | 16.45 | 5.70 | 100 | 1.95 | 1.95 |

**Average Cost Performance.** The learning curves of the tested methods trained to 20 seeds are plotted in Figure 8. The average cost of the tested methods is evaluated on three CT-RL environments in Table 9 at three levels of modeling error $\nu = 0\%, 10\%, 25\%$. We include the evaluation success rates with respect to the closed-loop control tasks for the three environments and three levels of modeling error in Table 10 (cf. Table 5 for definitions of task criteria). Examining Figure 8, we see that IPA exhibits the fastest learning convergence of the methods tested, converging in 5 iterations or less for all three environments. By comparison, FVIs converge at a similar rate on the simpler SOS benchmark but require $\approx 10$ iterations to converge on the pendulum and 30 iterations on the HSV. With the exception of cFVI on the pendulum, IPA also exhibits the lowest cost and variance between seeds.

**Average Cost Performance/Success Rate – SOS.** Turning to Table 9, we first see that all three methods perform comparably on the SOS benchmark, exhibiting similar average and standard deviation evaluated cost at all three modeling error levels $\nu$. Similarly, all methods achieve 100%

evaluation success on the SOS regardless of modeling error severity (cf. Table 10). Thus, all methods successfully generalize with respect to modeling error for this system.

**Average Cost Performance/Success Rate – Pendulum.** Examining Figure 8, we see that cFVI outperforms IPA in terms of average cost and variance on the nominal model $\nu = 0\%$. However, we would like to highlight this result in the context of the closed-loop performance observed of these two methods. We examine the swing-up performance of the pendulum beginning from full pendulum displacement $\theta_0 = 180°$ in Figure 16. As can be seen, the IPA response is the fastest of the three methods and is well-behaved with no overshoot. This hence points to a trade-off between lower cost (achieved by cFVI) and faster swing-up performance (achieved by IPA). Furthermore, IPA exhibits the best modeling error generalization on the pendulum (cf. Table 9), eventually outperforming cFVI in terms of average cost and variance when modeling error is increased. By comparison, rFVI exhibits the highest cost on the nominal pendulum model and struggles to generalize (cf. Table 9). However, all methods achieve a 100% success rate in the pendulum angle regulation task regardless of modeling error severity (cf. Table 10).

**Average Cost Performance/Success Rate – HSV.** On the flagship HSV environment, IPA exhibits a clear cost and evaluation success advantage over both FVI methods as well as the best modeling error generalization. Examining Table 9 the nominal model, IPA has the lowest average policy cost (24.69), followed by cFVI (29.79) and rFVI (33.37). The performance discrepancy increases with modeling error. At $\nu = 25\%$, IPA cost (32.45, a $+31\%$ degradation from nominal) far surpasses that of cFVI (63.80, a $+114\%$ degradation) and rFVI (74.15, a $+122\%$ degradation).

Turning to the evaluation success rates for the HSV in Table 10, we see that IPA achieves a 100% success rate on all tasks regardless of modeling error severity. FVIs exhibit a 100% success rate on the lax velocity regulation Task 1, and (with the exception of $> 99\%$ of cFVI at $\nu = 10\%$ and 25% modeling error) 100% success on the lax FPA regulation Task 2. However, FVIs struggle with the stringent velocity regulation Task 3 and FPA regulation Task 4. As will be examined shortly (cf. Figure 17 and surrounding discussion), FVIs struggle with evaluation success rate due to slow-decaying transients in their closed-loop time-domain responses, which prevent them from falling within the regulation thresholds.

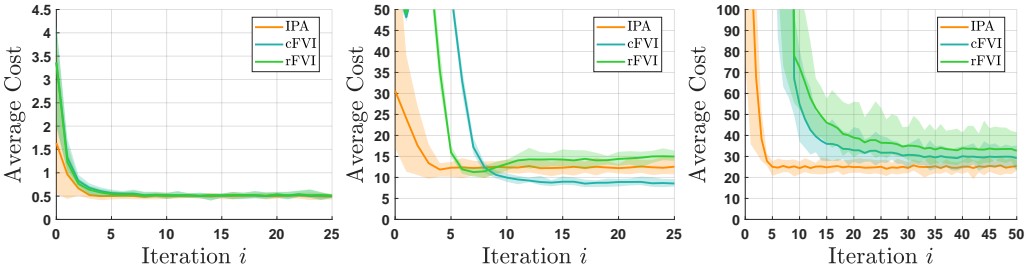

Figure 8: Learning curves of IPA and FVIs obtained over 20 seeds for the nominal models $\nu = 0\%$ of the SOS (left), pendulum (middle) and HSV (right). The shaded area displays the min/max range among seeds, as in the original works (Lutter et al., 2021; 2023b).

## I.2 Modeling Error Generalization: Cost Performance

Figure 9 shows the cost difference $J_{cFVI} - J_{IPA}$ between cFVI and IPA (first row), and the difference $J_{rFVI} - J_{IPA}$ between rFVI and IPA (second row) for the nominal SOS model $\nu = 0\%$ (left column), a 10% modeling error $\nu = 10\%$ (middle column), and a 25% modeling error $\nu = 25\%$ (right column). Note that wherever this difference is positive, IPA delivers *better* performance than the respective FVI algorithm. Table 11 presents the corresponding min, max, average, and standard deviation data. Figure 10 and Table 12 are laid out analogously for the pendulum, and Figure 11 and Table 13 are laid out analogously for the HSV.

**Cost Performance – SOS.** Overall, the three methods tested perform comparably with respect to cost on the SOS benchmark. However, IPA exhibits a slight performance edge, achieving the lowest

cost pointwise regardless of modeling error level. The most pronounced performance difference occurs with cFVI at $\nu = 25\%$ modeling error (cf. top right plot of Figure 9). Here, one can see two distinct regions corresponding to large initial displacements in the state $x_2$ for which cFVI has higher cost than IPA. As a note, this result is intuitive given that the modeling error parameter $\nu$ (57) selected primarily affects the unstable $x_2$ dynamics (cf. Appendix J for discussion). Meanwhile, we can see that rFVI manages to successfully generalize to the modeling error, exhibiting similar performance to IPA even at the severe 25% modeling error.

**Cost Performance – Pendulum.** Examining the performance on the nominal model in Figure 10, we see that cFVI performs quite well on the pendulum benchmark. IPA outperforms cFVI in a large region around the origin; however, cFVI outperforms IPA on the fringes of the state domain near full pendulum displacement $\theta = \pm 180°$. Once modeling error is introduced, the performance advantage of cFVI erodes; in the mean, cFVI and IPA perform almost comparably at 10% modeling error, while IPA outperforms cFVI at 25% modeling error (cf. Table 12). Thus, we conclude that IPA generalizes better with respect to modeling error than cFVI on this benchmark.

Meanwhile, rFVI performs worse than cFVI on the pendulum overall and struggles to generalize. On the nominal model (bottom left plot of Figure 10), IPA outperforms rFVI in the same regions where it outperforms cFVI to a larger degree, and the regions in which rFVI outperforms IPA are limited to the very corners of the state domain. IPA outperforms rFVI in the mean for all modeling error levels $\nu$, and the performance discrepancy increases by a large degree from 4.99 nominally $\nu = 0\%$ to 19.02 at $\nu = 25\%$ modeling error (cf. Table 12). This degradation is corroborated visually in Figure 10 (cf. bottom row, moving left to right).

**Cost Performance – HSV.** On the flagship HSV model, IPA exhibits the lowest mean cost regardless of the modeling error, and a distinct generalization advantage as the modeling error severity increases. On the nominal model (left column of Figure 11), FVIs and IPA perform comparably near the origin, but IPA begins to outperform FVIs by $\approx 10 - 20$ in the majority of the state domain. There are two fringes of the domain (corresponding to large initial FPA displacements $\gamma = \pm 1$ deg) in which FVIs slightly outperform IPA; however, in the mean IPA outperforms cFVI by 13.14 and rFVI by 16.45 on the nominal model $\nu = 0\%$.

Cost performance of both FVI algorithms degrades significantly when modeling error is introduced, in particular at the severe 25% modeling error (right column of Figure 11). Here, IPA outperforms FVIs pointwise, in the mean outperforming cFVI by 34.10 and rFVI by 35.90.

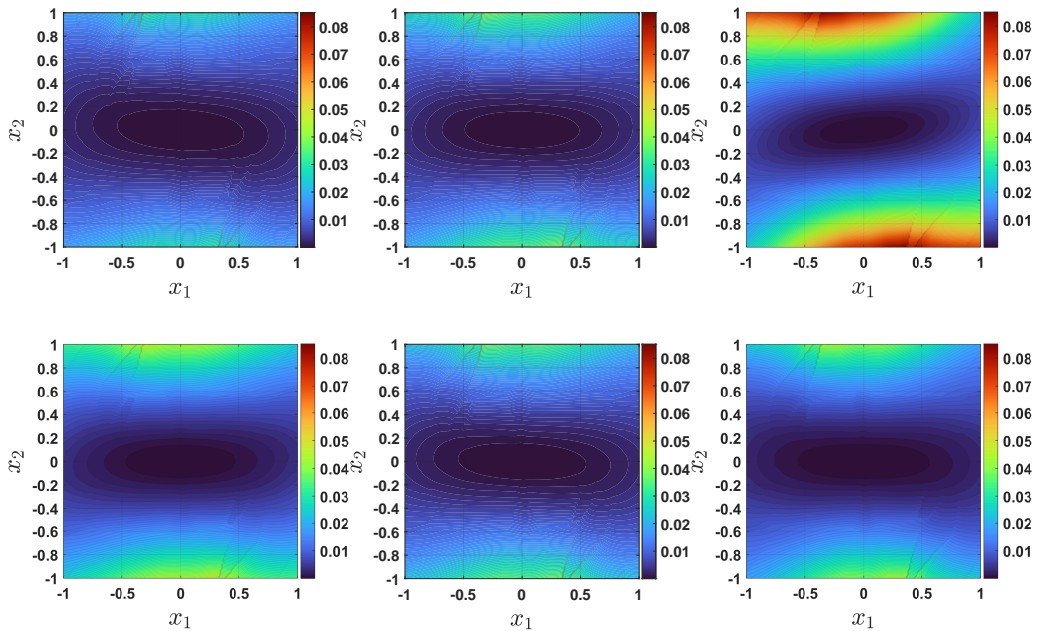

Figure 9: Cost performance results of SOS model. First row: cFVI cost difference $J_{cFVI} - J_{IPA}$ (2). Second row: rFVI cost difference $J_{rFVI} - J_{IPA}$ (2). Left: Nominal model $\nu = 0\%$. Middle: 10% modeling error $\nu = 10\%$. Right: 25% modeling error $\nu = 25\%$.

Table 11: SOS training cost/approximation data

| Function | Data | $\nu$ | | |
|---|---|---|---|---|
| | | 0% | 10% | 25% |
| | min | 1.34e-06 | 1.80e-06 | 3.24e-06 |
| $J_{cFVI} - J_{IPA}$ | max | 0.03 | 0.04 | 0.09 |
| | avg | 0.01 | 0.01 | 0.02 |
| | std | 0.01 | 0.01 | 0.02 |
| | min | 2.29e-06 | 1.71e-06 | 1.99e-06 |
| $J_{rFVI} - J_{IPA}$ | max | 0.05 | 0.04 | 0.04 |
| | avg | 0.01 | 0.01 | 0.01 |
| | std | 0.01 | 0.01 | 0.01 |
| | min | -0.01 | -0.01 | -0.01 |
| $J - \hat{V}$    IPA | max | -9.16e-08 | 0.03 | 0.10 |
| | avg | -6.28e-04 | 0.01 | 0.03 |
| | std | -6.83e-04 | 0.01 | 0.02 |
| | min | 2.51e-06 | 6.53e-06 | 1.28e-05 |
| $J - \hat{V}$    cFVI | max | 0.06 | 0.14 | 0.30 |
| | avg | 0.02 | 0.05 | 0.10 |
| | std | 0.01 | 0.03 | 0.08 |
| | min | -0.11 | -0.06 | -8.88e-03 |
| $J - \hat{V}$    rFVI | max | -3.24e-06 | -3.67e-07 | 0.12 |
| | avg | -0.03 | -0.01 | 0.03 |
| | std | 0.03 | 0.01 | 0.03 |

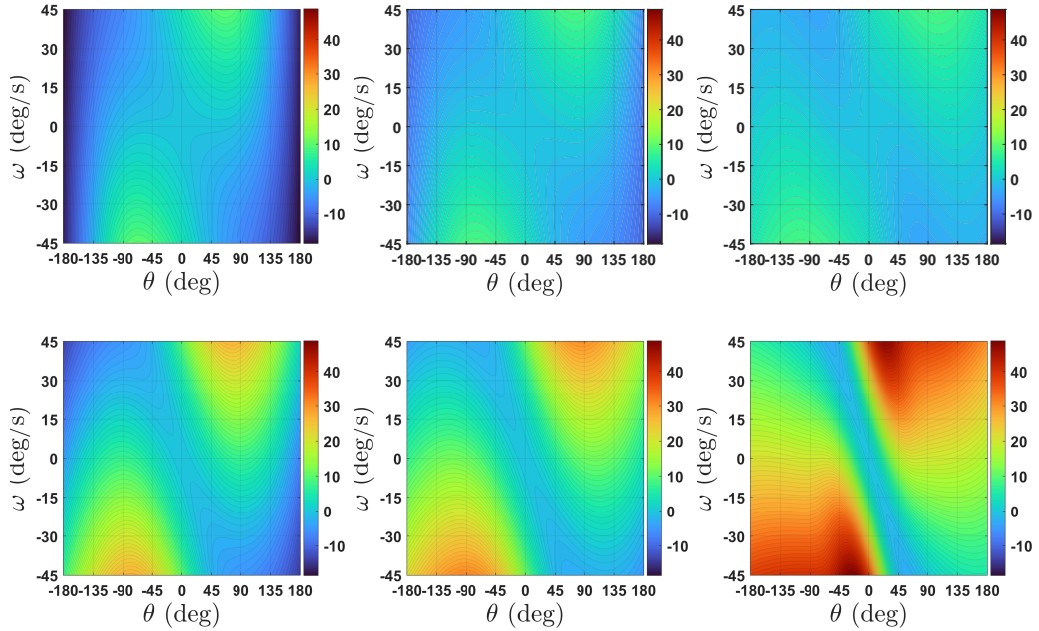

Figure 10: Cost performance results of pendulum model. First row: cFVI cost difference $J_{cFVI} - J_{IPA}$ (2). Second row: rFVI cost difference $J_{rFVI} - J_{IPA}$ (2). Left: Nominal model $\nu = 0\%$. Middle: 10% modeling error $\nu = 10\%$. Right: 25% modeling error $\nu = 25\%$.

Table 12: Pendulum training cost/approximation data

| Function | Data | $\nu$ | | |
|---|---|---|---|---|
| | | 0% | 10% | 25% |
| $J_{cFVI} - J_{IPA}$ | min | -18.53 | -10.21 | -3.33 |
| | max | 10.17 | 9.05 | 8.91 |
| | avg | -3.04 | -0.52 | 1.48 |
| | std | 6.43 | 3.91 | 2.87 |
| $J_{rFVI} - J_{IPA}$ | min | -14.25 | -4.94 | -2.68 |
| | max | 27.36 | 31.35 | 48.74 |
| | avg | 4.99 | 9.39 | 19.02 |
| | std | 8.54 | 8.87 | 12.40 |
| $J - \hat{V}$ IPA | min | -40.61 | -35.84 | -26.20 |
| | max | 4.78e-05 | 0.04 | 1.44 |
| | avg | -9.88 | -8.79 | -5.59 |
| | std | 11.13 | 9.91 | 7.63 |
| $J - \hat{V}$ cFVI | min | -8.90 | -9.17 | -9.15 |
| | max | 10.71 | 6.85 | 5.20 |
| | avg | -0.02 | -1.31 | -2.06 |
| | std | 4.02 | 3.40 | 3.17 |
| $J - \hat{V}$ rFVI | min | -27.49 | -26.56 | -23.22 |
| | max | 13.71 | 15.33 | 43.98 |
| | avg | -7.07 | -6.48 | 0.40 |
| | std | 9.11 | 9.19 | 13.87 |

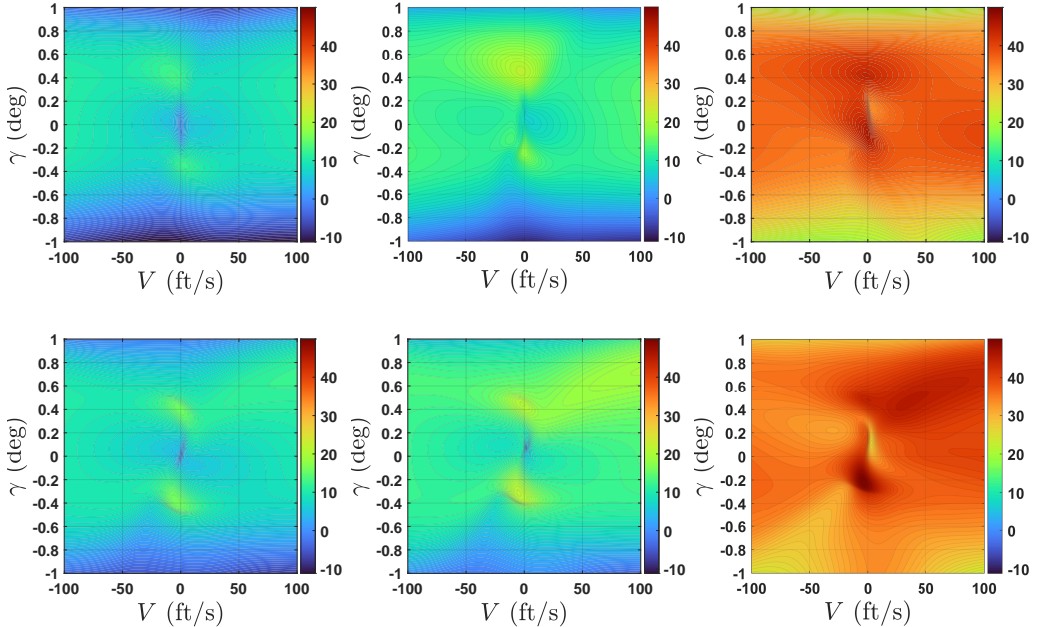

Figure 11: Cost performance results of HSV model. First row: cFVI cost difference $J_{cFVI} - J_{IPA}$ (2). Second row: rFVI cost difference $J_{rFVI} - J_{IPA}$ (2). Left: Nominal model $\nu = 0\%$. Middle: 10% modeling error $\nu = 10\%$. Right: 25% modeling error $\nu = 25\%$.

Table 13: HSV training cost/approximation data

| Function | Data | $\nu$ | | |
|---|---|---|---|---|
| | | 0% | 10% | 25% |
| $J_{cFVI} - J_{IPA}$ | min | -11.16 | -8.51 | 19.18 |
| | max | 13.14 | 21.77 | 45.83 |
| | avg | 5.63 | 9.76 | 34.10 |
| | std | 5.23 | 5.60 | 5.56 |
| $J_{rFVI} - J_{IPA}$ | min | -6.92 | -2.90 | 23.61 |
| | max | 16.45 | 21.75 | 49.97 |
| | avg | 7.24 | 11.37 | 35.90 |
| | std | 4.13 | 4.60 | 4.54 |
| $J - \hat{V}$    IPA | min | -46.62 | -50.55 | -56.11 |
| | max | -1.83e-03 | 0.02 | 0.20 |
| | avg | -13.04 | -14.76 | -17.66 |
| | std | 9.61 | 10.87 | 13.10 |
| $J - \hat{V}$    cFVI | min | -49.48 | -37.75 | -0.63 |
| | max | 28.33 | 31.21 | 56.68 |
| | avg | -7.77 | -0.96 | 29.12 |
| | std | 17.15 | 14.22 | 10.53 |
| $J - \hat{V}$    rFVI | min | -92.74 | -79.03 | -41.03 |
| | max | 27.66 | 30.60 | 57.32 |
| | avg | -25.27 | -18.46 | 11.81 |
| | std | 29.52 | 26.41 | 21.36 |

I.3   MODELING ERROR GENERALIZATION: APPROXIMATION PERFORMANCE

Figures 12, 13, and 14 show the critic network error $J - \hat{V}$ for IPA (first row), cFVI (second row), and rFVI (third row) for the SOS, pendulum, and HSV systems, respectively. The corresponding data is tabulated in Tables 11, 12, and 13, respectively. In general, it is desirable for the critic approximation error $J - \hat{V}$ to be as small in magnitude as possible (so the critic is accurate).

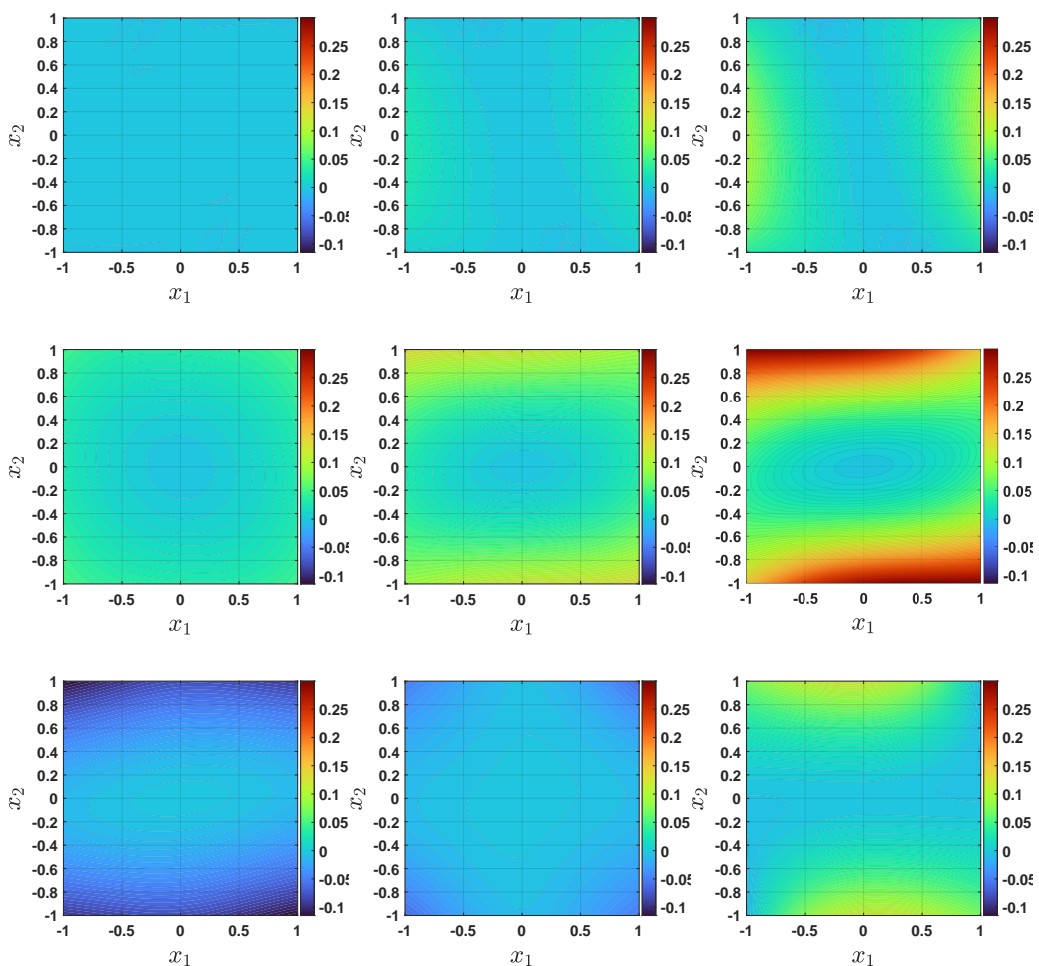

Figure 12: Critic NN approximation error $J - \hat{V}$ of SOS model. Left: Nominal model $\nu = 0\%$. Middle: 10% modeling error $\nu = 10\%$. Right: 25% modeling error $\nu = 25\%$. First row: IPA. Second row: cFVI (Lutter et al., 2021). Third row: rFVI (Lutter et al., 2023b).

**Approximation Performance – SOS.** Examining Figure 12, IPA exhibits the lowest critic network approximation error regardless of modeling error severity and the best generalization overall. As is the case with cost performance, cFVI struggles the most to generalize on the SOS. Meanwhile, rFVI tends to underestimate policy performance on the nominal model and exhibits slightly higher approximation error than IPA when modeling error increases.

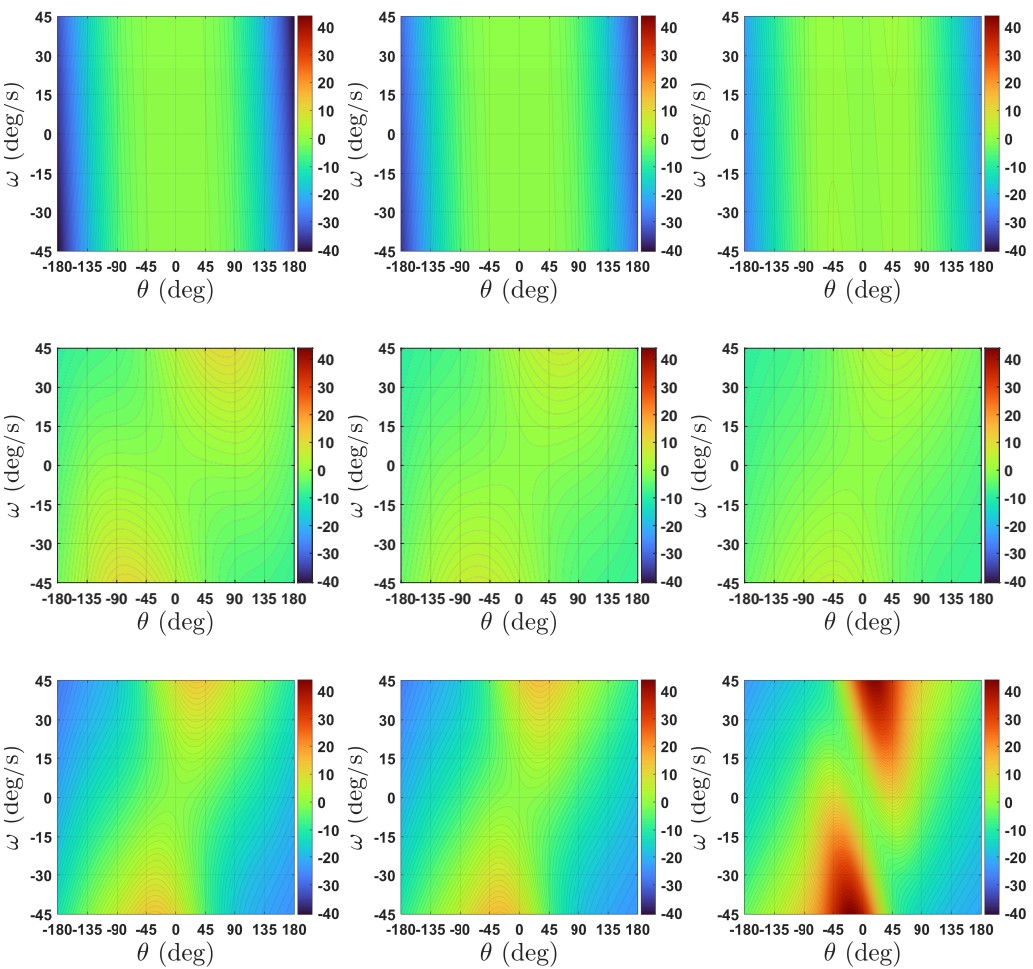

Figure 13: Critic NN approximation error $J - \hat{V}$ of pendulum model. Left: Nominal model $\nu = 0\%$. Middle: 10% modeling error $\nu = 10\%$. Right: 25% modeling error $\nu = 25\%$. First row: IPA. Second row: cFVI (Lutter et al., 2021). Third row: rFVI (Lutter et al., 2023b).

**Approximation Performance – Pendulum.** Examining Figure 13, IPA approximation error increases in the fringes near full pendulum displacement $\theta = \pm180°$, where IPA cost performance also struggles (cf. Figure 10). However, IPA exhibits low approximation error in a wide band around the origin encompassing the majority of the state domain, meeting or outperforming cFVI in this region. Furthermore, IPA estimation error improves as modeling error is introduced, demonstrating good generalization. Nevertheless, cFVI performs the best overall in terms of estimation error on the pendulum benchmark, exhibiting both excellent performance on the nominal model and generalization to modeling error. Meanwhile, rFVI performs worse on the nominal model and struggles to generalize in comparison to IPA or cFVI.

**Approximation Performance – HSV.** As is the case with cost performance, IPA exhibits a pronounced approximation performance advantage on the nominal HSV model and the best generalization to modeling error. On the nominal HSV model $\nu = 0\%$ (left column of Figure 14), one can see that both cFVI and rFVI struggle with overestimation in the band $\gamma \in [-0.5°, 0.5°]$, in particular for large velocity displacements $V = \pm100$ ft/s. IPA approximation error remains lower in this region. Furthermore, rFVI begins to underestimate policy performance for large FPA displacements $\gamma = \pm1°$, where cFVI performs comparably better.

Approximation performance significantly degrades for FVIs, in particular at the 25% modeling error (right column of Figure 14). Here, both FVI algorithms begin to overestimate policy performance to a large degree. Where rFVI begins underestimates policy performance on the nominal model near $\gamma = \pm1°$, it performs comparably better than cFVI at 25% modeling error. This suggests that rFVI adversarial input successful improves generalization properties of this robust variant somewhat. By comparison, IPA's approximation error remains at comparable levels and in a similar radial pattern to its performance on the nominal model.

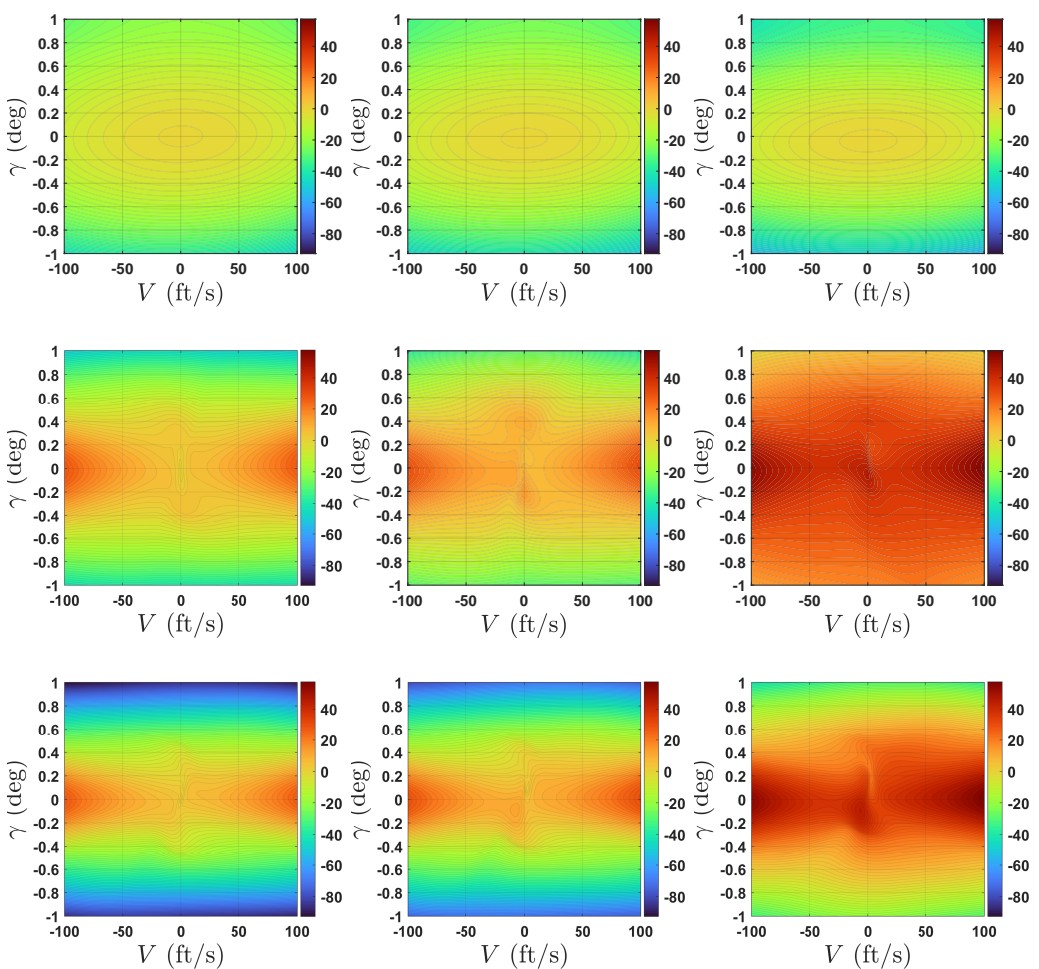

Figure 14: Critic NN approximation error $J - \hat{V}$ of HSV model. Left: Nominal model $\nu = 0\%$. Middle: 10% modeling error $\nu = 10\%$. Right: 25% modeling error $\nu = 25\%$. First row: IPA. Second row: cFVI (Lutter et al., 2021). Third row: rFVI (Lutter et al., 2023b).

### I.4 MODELING ERROR GENERALIZATION: CLOSED-LOOP PERFORMANCE

We examine the closed-loop time-domain responses of all three environments at 0%, 10%, and 25% modeling error $\nu$. The associated plots for the SOS, pendulum, and HSV can be found in Figures 15, 16, and 17, respectively.

**Closed-Loop Performance – SOS.** In Figure 15, we plot the closed-loop regulation responses of the tested methods to the initial condition $x_0 = [1, 1]^T$. As can be seen, regardless of the modeling error severity IPA and the FVIs perform very similarly. This corroborates the cost and approximation results in Figures 9, 12, and Table 11, which establish that these methods successfully learn the same optimal policy.

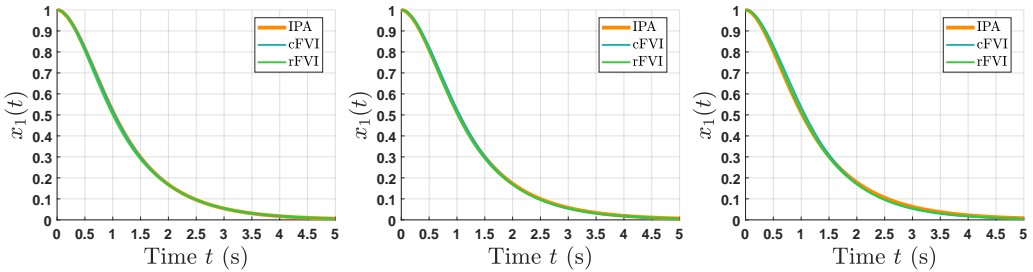

Figure 15: Closed-loop response $x_1(t)$ of SOS model to initial condition $x_0 = [1, 1]^T$. Left: Nominal model $\nu = 0\%$. Middle: 10% modeling error $\nu = 10\%$. Right: 25% modeling error $\nu = 25\%$.

**Closed-Loop Performance – Pendulum.** We examine the swing-up performance of the pendulum beginning from full pendulum displacement $\theta_0 = 180°$ in Figure 16. As can be seen, the IPA response is the fastest of all three methods regardless of the modeling error tested. We would like to highlight this result in the context of the cost and approximation performance of IPA illustrated in Figures 10 and 13, respectively. In these two figures, it can be seen that IPA struggles for large initial pendulum displacements $\theta = \pm180°$; however, Figure 16 shows that IPA's swing-up performance at full pendulum displacement $\theta_0 = 180°$ is the most responsive of the three methods. This hence points to a trade-off between lower cost/approximation error (achieved by FVIs) and faster swing-up performance (achieved by IPA). This trade-off is corroborated between the FVIs themselves. Comparing FVIs in 16, we see that rFVI exhibits a faster response than cFVI; meanwhile, examining the state $x = [\theta, \dot{\theta}]^T = [180°, 0]^T$ in Figures 10 and 13, we see that cFVI achieves lower cost and approximation error than rFVI at this same initial condition.

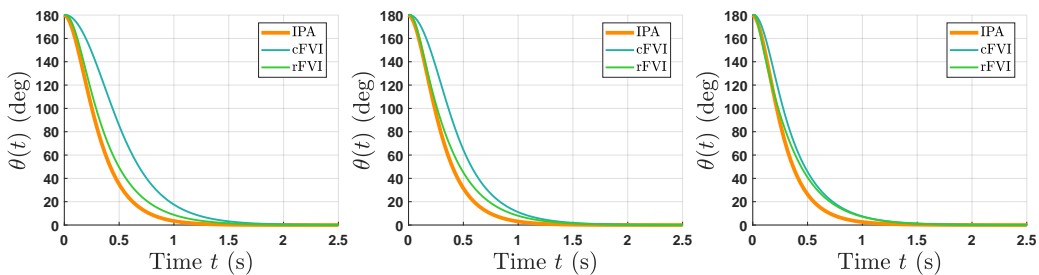

Figure 16: Swing-up closed-loop response of pendulum model $\theta(t)$ from hanging pendulum position $\theta_0 = 180°$. Left: Nominal model $\nu = 0\%$. Middle: 10% modeling error $\nu = 10\%$. Right: 25% modeling error $\nu = 25\%$.

**Closed-Loop Performance – HSV.** We examine the closed-loop performance on the HSV in Figure 17, issuing a 100 ft/s step velocity command (top row) and a 1° step FPA command (bottom row). Beginning on the velocity response (top row of Figure 17), we can see that both FVI algorithms have large overshoot compared to IPA. Furthermore, the FVI responses exhibit pronounced and slow-decaying transients about the 100 ft/s reference command setpoint. This time-domain behavior is a clear visual indicator of why FVIs struggle with evaluation success rate in velocity regulation (cf. Table 3, HSV Tasks 1 and 3). By comparison, IPA exhibits less overshoot, faster settling time, and a well-behaved system response without oscillatory transients.

Examining the FPA response (bottom row of Figure 17), we see that IPA and FVIs yield similar FPA overshoot. However, FVIs exhibit similar oscillatory transients as in the velocity response, giving IPA faster settling time. It is because of these oscillations that FVIs struggle with evaluation success rate in FPA regulation (cf. Table 3, HSV Tasks 2 and 4). Furthermore, such oscillations in FPA (i.e., in deflection the vehicle velocity vector) are highly undesirable in flight control settings, as they cause wear on the fuselage and may excite the HSV flexible modes (Bolender & Doman, 2006a).

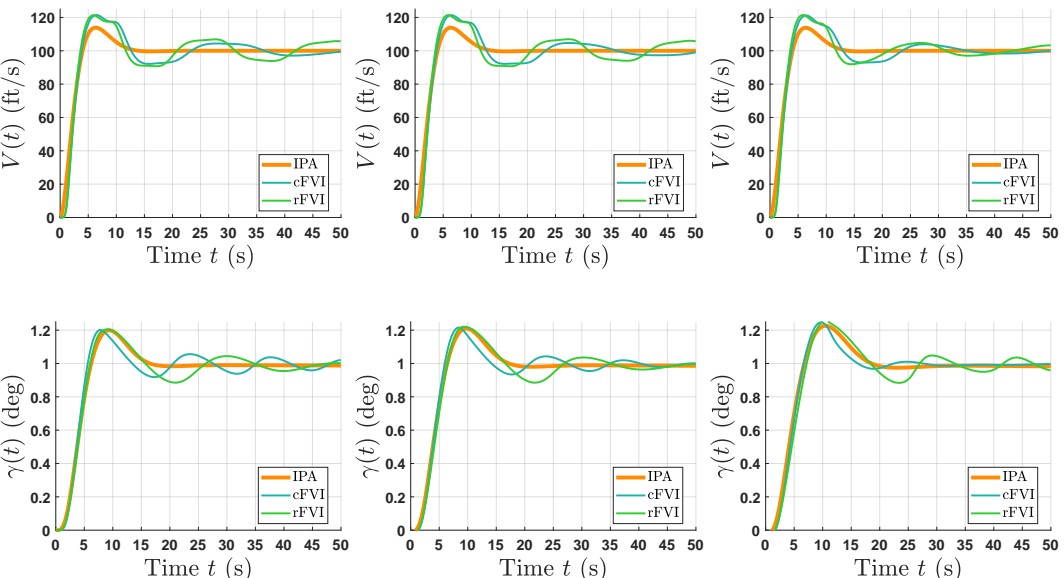

Figure 17: Closed-loop responses of HSV model. Top row: 100 ft/s step velocity $V$ command. Bottom row: 1° step FPA $\gamma$ command. Left: Nominal model $\nu = 0\%$. Middle: 10% modeling error $\nu = 10\%$. Right: 25% modeling error $\nu = 25\%$.

**Time/Data Efficiency.** Table 14 lists key algorithm complexity parameters for IPA and FVI. In many cases, IPA exhibits orders of magnitude less time/data complexity than the SOTA FVI works. To illustrate, we examine the ratio of IPA/FVI on the HSV model. For number of data samples required: 1/5,750,000, training episodes required: IPA/FVI = 1/5,250,000, training time: 1/1,500, algorithm iterations: 1/5.

Table 14: Comparison of time/data complexity of IPA and FVIs on 3 environments

| Parameter | SOS | | Pendulum | | HSV | |
|---|---|---|---|---|---|---|
| | IPA | cFVI/rFVI | IPA | cFVI/rFVI | IPA | cFVI/rFVI |
| # Trajectory Data Samples | 10 | 2.88e+07 | 15 | 8.63e+07 /2.88e+07 | 30 | 1.73e+08 |
| # Episodes (Data Collection) | 1 | 6.40e+05 | 1 | 2.63e+06 /6.40e+05 | 1 | 5.25e+06 |
| Avg Training Time (s) | 0.20 | 1.42e+03 /1.41e+03 | 0.22 | 1.76e+03 /1.46e+03 | 2.75 | 4.60e+03 4.79/e+03 |
| # Algorithm Iterations $i$ | 5 | 25 | 5 | 25 | 10 | 50 |

# APPENDIX J    SECOND ORDER ACADEMIC EXAMPLE (VAMVOUDAKIS & LEWIS, 2010)

## J.1    SECOND ORDER SYSTEM (SOS) MODEL

We consider the identical second order system (SOS) model used in the seminal ADP work (Vamvoudakis & Lewis, 2010), which has the following dynamics

$$\begin{bmatrix} \dot{x}_1 \\ \dot{x}_2 \end{bmatrix} = \begin{bmatrix} -x_1 + x_2 \\ -0.5x_1 - 0.5\psi(x) \end{bmatrix} + \begin{bmatrix} 0 \\ \cos(2x_1) + 2 \end{bmatrix} u, \tag{56}$$

$$\psi(x) \triangleq \nu\,\psi_0(x), \qquad \psi_0(x) \triangleq x_2\big(1 - (\cos(2x_1) + 2)^2\big). \tag{57}$$

The function $\psi$ (57) contributes to the nonlinearity in the $\dot{x}_2$ drift dynamics $f(x)$ for this system. It also determines the stability properties of the origin equilibrium $x_e = 0$. This system is open-loop unstable, and as the modeling error parameter $\nu$ (57) increases above its nominal value $\nu = 1$, the system nonlinearity grows stronger and the SOS becomes more unstable. We have included the eigenvalues of the open-loop linearization of the SOS in Table 15. Thus, in this study we examine the more challenging perturbation direction $\nu > 1$.

Table 15: SOS instability versus modeling error parameter $\nu$ (57)

| $\nu$ (57) | Unstable Pole Location |
|---|---|
| 1 (nom, $= 0\%$) | 3.8979 |
| 1.1 ($= 10\%$) | 4.3058 |
| 1.25 ($= 25\%$) | 4.9155 |

# APPENDIX K    PENDULUM MODEL & DESIGN FRAMEWORK

## K.1    PENDULUM MODEL

We consider the identical pendulum model used in the cFVI evaluations (Lutter et al., 2021; 2023b) for this work, which has the following dynamics

$$\dot{\theta} = \omega,$$
$$\dot{\omega} = \frac{mgL}{2I}\sin\theta + \frac{1}{I}\tau, \tag{58}$$

where $\theta$ is the pendulum angle (measured zero pointing upward, positive counterclockwise), $\omega$ is the pendulum angular velocity, and $\tau$ is the torque applied to the pendulum base. The numerical values of all model constants are chosen identical to the cFVI evaluations (Lutter et al., 2021; 2023b) and are available in Table 16. We examine the upright pendulum equilibrium $x_e = [\theta_e, \omega_e]^T = [0\text{ rad}, 0\text{ rad/s}]^T$. This equilibrium is naturally unstable (upright pendulum instability).

Table 16: Pendulum model parameters

| Definition | Symbol | Value |
|---|---|---|
| Pendulum length | $L$ | $L_0 = 1$ m (nominal) |
| Pendulum mass | $m$ | 1 kg |
| Gravitational field constant | $g$ | 9.81 m/s$^2$ |
| Pendulum moment of inertia | $I$ | $\frac{1}{3}mL^2$ |

The pendulum length $L$ is a central physical parameter in the dynamics (58), as it determines the severity of the upright pendulum instability. In the studies conducted in this work, we will focus on how modeling errors in the pendulum length $L$ affect the pendulum dynamics and learning performance. Specifically, we study modeling errors of the form

$$L = \nu\, L_0, \tag{59}$$

where $L_0 \in \mathbb{R}$ is a nominal value of the pendulum length, and $\nu \in \mathbb{R}$ is the modeling error parameter (nominally 1). As $\nu < 1$ decreases, $L < L_0$ decreases. Table 17 shows the inverted pendulum instability as a function of the modeling error $\nu$ (59). As can be seen, the pendulum becomes more unstable with decreasing pendulum length $L$. Thus, the modeling error perturbation direction $\nu < 1$ (59) is studied in this work.

Table 17: Pendulum instability and control effectiveness versus modeling error parameter $\nu$ (59)

| $\nu$ (59) | Unstable Pole Location |
|---|---|
| 1 (nom, $= 0\%$) | 3.8360 |
| 1.1 ($= 10\%$) | 3.6575 |
| 1.25 ($= 25\%$) | 3.4310 |

## Appendix L  HSV Model and Decentralized Design Framework

### L.1  HSV Model

The HSV model used in this study is the standard Wang and Stengel model developed in (Wang & Stengel, 2000; Marrison & Stengel, 1998) based on NASA Langley's winged-cone tabular aero-propulsive data (Shaughnessy et al., 1990). This model has served as a standard testbed for HSV control development and has since been used in seminal classically-based works such as (Xu et al., 2003; 2004), and simplified variants of it have been used in state-of-the-art RL-based control works such as (Zhao et al., 2023; Xu et al., 2013; 2015):

$$\dot{V} = \frac{T\cos\alpha - D}{m} - \frac{\mu\sin\gamma}{r^2},$$
$$\dot{\gamma} = \frac{L + T\sin\alpha}{mV} - \frac{(\mu - V^2 r)\cos\gamma}{V r^2},$$
$$\dot{\theta} = q,$$
$$\dot{q} = \frac{\mathcal{M}}{I_{yy}},$$
$$\dot{h} = V\sin\gamma. \tag{60}$$

The HSV (60) is fifth-order, with states $x = [V,\ \gamma,\ \theta,\ q,\ h]^T$. Here, $V$ is the vehicle airspeed, $\gamma$ the flightpath angle (FPA), $\alpha$ the angle of attack (AOA), $\theta \triangleq \alpha + \gamma$ the pitch attitude, $q$ the pitch rate, and $h$ the vehicle altitude. In addition, $r(h) = h + R_E$ is the total distance from the earth's center to the HSV, $R_E = 20,903,500$ ft is earth's radius, and $\mu \triangleq Gm_E = 1.39 \times 10^{16}$ ft$^3$/s$^2$, where $G$ is Newton's gravitational constant and $m_E$ is the earth's mass. As a note, the notation $\mu$ in (60) is standard in flight control literature and is not to be confused with the learning community's use of $\mu$ to denote a control policy. $L, D, T, \mathcal{M}$ are the lift, drag, thrust, and pitching moment, respectively, and are given by

$$L = \frac{1}{2}\rho V^2 S C_L, \qquad D = \frac{1}{2}\rho V^2 S C_D, \qquad T = \frac{1}{2}\rho V^2 S C_T, \qquad \mathcal{M} = \frac{1}{2}\rho V^2 S \bar{c} C_{\mathcal{M}}, \tag{61}$$

where $\rho$ is the local air density, $S = 3603$ ft$^2$ is the wing planform area, and $\bar{c} = 80$ ft is the mean aerodynamic chord of the wing. Air density $\rho$ and speed of sound $a$ are modeled as functions of altitude $h$ by

$$\rho = 0.00238 e^{-\frac{h}{24,000}}, \tag{62}$$

$$a = 8.99 \times 10^{-9} h^2 - 9.16 \times 10^{-4} h + 996, \tag{63}$$

and Mach number $M \triangleq \frac{V}{a}$. The aerodynamic coefficients $C_L$, $C_D$, $C_{\mathcal{M}}$, and $C_T$ are heavily nonlinear functions of the flight condition as follows

$$C_L = C_{L,\alpha} + C_{L,\delta_E}, \tag{64}$$

$$C_{L,\alpha} = \nu \, C_{L,\alpha 0}, \qquad C_{L,\alpha 0} \triangleq \alpha \left( 0.493 + \frac{1.91}{M} \right), \tag{65}$$

$$C_{L,\delta_E} = \left( -0.2356\alpha^2 - 0.004518\alpha - 0.02913 \right) \delta_E, \tag{66}$$

$$C_D = 0.0082 \left( 171\alpha^2 + 1.15\alpha + 1 \right) \left( 0.0012 M^2 - 0.054 M + 1 \right), \tag{67}$$

$$C_{\mathcal{M}} = C_{\mathcal{M},\alpha} + C_{\mathcal{M},q} + C_{\mathcal{M},\delta_E}, \tag{68}$$

$$C_{\mathcal{M},\alpha} = 10^{-4} \left( 0.06 - e^{-\frac{M}{3}} \right) \left( -6565\alpha^2 + 6875\alpha + 1 \right), \tag{69}$$

$$C_{\mathcal{M},q} = \left( \frac{q\bar{c}}{2V} \right) \left( -0.025 M + 1.37 \right) \left( -6.83\alpha^2 + 0.303\alpha - 0.23 \right), \tag{70}$$

$$C_{\mathcal{M},\delta_E} = 0.0292(\delta_E - \alpha), \tag{71}$$

$$C_T = \begin{cases} 0.0105 \left( 1 + \frac{17}{M} \right) (1 + 0.15)\delta_T, & \delta_T < 1 \\ 0.0105 \left( 1 + \frac{17}{M} \right) (1 + 0.15\delta_T), & \delta_T \geq 1, \end{cases} \tag{72}$$

where $\delta_E$ is the elevator setting, $\delta_T$ is the throttle, and $\nu \in \mathbb{R}$ is the modeling error parameter (nominally 1) in the basic lift increment coefficient $C_{L,\alpha}$ (65). The controls are given by $u = [\delta_T, \delta_E]^T$, and we examine the outputs $y = [V, \gamma]^T$. As in (Wang & Stengel, 2000; Marrison & Stengel, 1998), we examine a level flight condition $q_e = 0, \gamma_e = 0°$, at $M_e = 15$, $h_e = 110,000$ ft, which corresponds to an airspeed $V_e = 15,060$ ft/s. At this flight condition, the vehicle is equilibrated at $\alpha_e = 1.7704°$ by the controls $\delta_{T,e} = 0.1756$ ($T_e = 4.4966 \times 10^4$ lb), $\delta_{E,e} = -0.3947°$.

**HSV Dynamical Challenges.** The HSV is a significant control challenge due to its dynamical features. First, the HSV is open-loop unstable. Linearization of the model about the equilibrium $(x_e, u_e)$ has eigenvalues at $s = -0.8291, 0.7165$ (short-period modes), $s = -0.00001 \pm 0.0276j$ (phugoid modes), and $s = 0.0005$ (altitude mode). The unstable short-period mode at $s = 0.7165$ is associated with the vehicle pitch-up instability (long vehicle forebody, aftward-set center of mass due to propulsion system). As is common with tail-controlled aircraft, the elevator-FPA map is nonminimum phase (Bolender & Doman, 2005). The linearization has transmission zeros at $s = 8.3938, -8.4620$, the right half plane zero at $s = 8.3938$ attributed to the elevator-FPA map (parasitic negative lift increment due to pitch-up elevator deflections).

Reducing the lift coefficient $\nu < 1$ (65) represents degraded lift efficiency and a more difficult vehicle to control dynamically. We have calculated the unstable pole location and right-half-plane zero location as a function of the modeling error parameter $\nu$ (65) in Table 18. As can be seen, the system instability decreases slightly with increasing modeling error, but the nonminimum phase zero gets closer to the origin. Thus, the pole/zero ratio drops from 11.72 nominally ($\nu = 0\%$), to 11.36 ($\nu = 10\%$), to 10.88 ($\nu = 25\%$), presenting a greater control design challenge. Thus, we study the modeling error perturbation direction $\nu < 1$ in this study.

Table 18: HSV instability and nonminimum phase zero versus modeling error parameter $\nu$ (65)

| $\nu$ (65) | Unstable Pole Location | Nonminimum Phase Zero Location |
|---|---|---|
| 1 (nom, $= 0\%$) | 0.7165 | 8.3938 |
| 0.9 ($= 10\%$) | 0.7011 | 7.9619 |
| 0.75 ($= 25\%$) | 0.6681 | 7.2664 |

## L.2 HSV Decentralized Control Framework

This work implements a decentralized design methodology inspired by (Dickeson et al., 2009a) developed for HSVs and extensively tested on HSVs. Here, policies are designed separately for the velocity loop (associated with the airspeed $V$ and throttle control $\delta_T$) and rotational loop (associated with the FPA $\gamma$, attitude $\theta$, $q$, and elevator control $\delta_E$). As in (Dickeson et al., 2009a), for controllability reasons we do not feed back altitude $h$ in the learning control, though altitude is still included in the nonlinear simulation. To reduce overshoot due to step reference commands $r$ and initial condition $x_0$ transients, we include in the design a reference command pre-filter outside the feedback loop (a standard control design practice for HSVs, for further discussion see, e.g., (Dickeson et al., 2009a)).

In order to achieve zero steady-state error to step reference commands, we augment the plant at the output with the integrator bank $z = \int y\, d\tau = [z_V,\, z_\gamma]^T = \left[\int V\, d\tau,\, \int \gamma\, d\tau\right]^T$. For decentralization, the state/control vectors are thus partitioned as $x_1 = [z_V,\, V]^T$, $u_1 = \delta_T$ ($n_1 = 2$, $m_1 = 1$) and $x_2 = [z_\gamma,\, \gamma,\, \theta,\, q]^T$, $u_2 = \delta_E$ ($n_2 = 4$, $m_2 = 1$).

# APPENDIX M  ADP-BASED LEADING CT-RL DESIGN INSIGHTS AND PERFORMANCE LIMITATIONS

ADP approaches have been developed largely within the scope of seminal works such as integral reinforcement learning (IRL) (Vrabie & Lewis, 2009), synchronous policy iteration (SPI) (Vamvoudakis & Lewis, 2010), robust ADP (RADP) (Jiang & Jiang, 2014), and continuous-time value iteration (CT-VI) (Bian & Jiang, 2022). These methods achieve substantial theoretical results. As a result of ADP's theoretical frameworks in adaptive and optimal control, Lyapunov arguments are available to prove qualitative properties including weight convergence and closed-loop stability results. However, the results require restrictive theoretical assumptions. For a complete list of assumptions required by 1) IPA, 2) leading ADP CT-RL works (see above), and 3) the SOTA FVI works (Lutter et al., 2021; 2023b), please see Appendix A. In the case of ADPs, these are difficult to satisfy for even simple academic examples, and as a result these methods exhibit empirical issues (Wallace & Si, 2024). These issues impede algorithm performance long before the methods may be substantively evaluated for generalization to varying system ICs and modeling error, as we do for IPA and FVIs in Section 5. The in-depth numerical studies in (Wallace & Si, 2024) find that these methods fail to synthesize controls for simple second-order systems with known closed-form solutions. A small change in the basis of such small problem led to learning failure. We will illustrate this same phenomenon in the coming evaluation.

## M.1 System and Ablation of Critic Bases

**Second Order System (SOS) Studied (Vamvoudakis & Lewis, 2010).** In order to illustrate these empirical issues, we will study the performance of the four seminal ADP works on the academic second order system (SOS) studied in this work (cf. Appendix J). The SOS is a suitable benchmark environment for these ADP methods; indeed, the model was originaly developed for evaluation in the SPI work (Vamvoudakis & Lewis, 2010). The dynamics of the SOS are given in Equation (56). The SOS is an academic example constructed such that, for the nominal model $\nu = 1$ and with the choice of state and control penalties $Q = I_2$, $R = 1$ given in (49), the optimal value $V^*$ and optimal policy $\mu^*$ are known *a priori* in analytic form as

$$V^*(x) = \frac{1}{2}x_1^2 + x_2^2, \tag{73}$$

$$\mu^*(x) = -(\cos(2x_1) + 2)x_2. \tag{74}$$

As is the case with all classes of learning algorithms, a central hyperparameter of ADPs is the basis functions chosen for the critic neural network. The seminal ADPs all study a linear approximation structure of the following form

$$\hat{V}(x) = \phi^T(x)c, \tag{75}$$

where $\phi(x) = \begin{bmatrix} \phi_1(x) & \phi_2(x) & \dots & \phi_M(x) \end{bmatrix}^T \in \mathbb{R}^M$ is the critic bases consisting of $M \in \mathbb{N}$ basis functions, and $c \in \mathbb{R}^M$ is the critic weight vector.

**Ablation: Critic Bases.** In this study, we will demonstrate ablation sensitivity of the ADPs with respect to the following two natural choice of critic bases:

$$\phi(x) = \left[ \begin{array}{ccc} x_1^2 & x_1 x_2 & x_2^2 \end{array} \right]^T, \tag{76}$$

$$\phi(x) = \left[ \begin{array}{cccccc} x_1^2 & x_1 x_2 & x_2^2 & x_1^4 & x_1^3 x_2 & x_1^2 x_2^2 & x_1 x_2^3 & x_2^4 \end{array} \right]^T, \tag{77}$$

The first choice of bases $\phi$ (76) is identical to the choice of bases in the original SPI study for this system (Vamvoudakis & Lewis, 2010). The second choice of bases $\phi$ (77) contains the first as a subset, and it is chosen for illustration because it is identical to the choice of bases in the original IRL study (Vrabie & Lewis, 2009) on a similar academic example. Thus, these bases are well-motivated choices for study and have been demonstrated previously by the leading ADP works.

Crucially, both choices of bases $\phi$ (76), (77) can achieve *exact* approximation of the optimal value $V^*$, and inspection of (73) shows that the optimal critic weights $c^*$ for the bases (76), (77) are given respectively as

$$c^* = \left[ \begin{array}{ccc} \frac{1}{2} & 0 & 1 \end{array} \right]^T, \tag{78}$$

$$c^* = \left[ \begin{array}{cccccccc} \frac{1}{2} & 0 & 1 & 0 & 0 & 0 & 0 & 0 \end{array} \right]^T, \tag{79}$$

## M.2 Setup and Implementation

### M.2.1 Networks and Initialization

**Initial Stabilizing Policy.** For all methods, we initialize the critic weights to $c_0 = [1, 0, 4]^T$ for (76) and $c_0 = [1, 0, 4, 0, 0, 0, 0, 0]^T$ for (76). These critic weights were chosen to implement the same initial stabilizing policy $\mu_0$ for all works, given by

$$\mu_0(x) = -4(\cos(2x_1) + 2)x_2$$
$$= -\frac{1}{2} R^{-1} g^T(x) \frac{\partial}{\partial x} \{\phi^T(x) c_0\} \tag{80}$$

This policy $\mu_0$ was chosen so that it can be implemented in the single-network control policy structures used by IRL (Vrabie & Lewis, 2009) and SPI (Vamvoudakis & Lewis, 2010).

**Actor Neural Network: RADP and CT-VI.** Of the four seminal ADPs, the RADP (Jiang & Jiang, 2014) and CT-VI (Bian & Jiang, 2022) methods make use of a self-standing actor neural network, whose basis functions we will denote $\psi(x)$. For these networks, we choose the minimal bases required to approximate the optimal policy $\mu^*$ (74) and in order to implement the common initial stabilizing policy $\mu_0$ (80):

$$\psi(x) = \left[ \begin{array}{cccc} x_1 & x_2 & \cos(2x_1)x_1 & \cos(2x_1)x_2 \end{array} \right]^T. \tag{81}$$

For reference, the optimal actor network weights $w^* \in \mathbb{R}^4$ for this problem are $w^* = [0, -2, 0, -1]^T$.

**Hamiltonian Neural Network: CT-VI.** The CT-VI method makes use of a novel network to approximate the optimal Hamiltonian function, for details see the original work (Bian & Jiang, 2022) and subsequent discussions in (Wallace & Si, 2024). We shall denote the basis functions of the Hamiltonian network by $\theta(x)$, and likewise we use minimal bases to approximate the Hamiltonian:

$$\theta(x) = \left[ \begin{array}{cccc} x_2^2 \cos^2(2x_1) & x_2^2 \cos(2x_1) & x_1^2 & x_2^2 \end{array} \right]^T. \tag{82}$$

For reference, the optimal Hamiltonian network weights $v^* \in \mathbb{R}^4$ for this problem are $v^* = [1, 4, -1, 3]^T$.

### M.2.2 Hyperparameter Selections

All ADPs collect state-action data under a feedback control of the form $u = \mu(x) + d$, where $d$ is a probing noise excitation. The policy $\mu$ may be kept constant at the initialization $\mu_0$, or it may be updated online (for specific details, see the respective reference of each method).

**Algorithm Hyperparameters.** We choose hyperparameters for the ADPs with the following considerations: 1) The selections reflect in-depth quantitative hyperparameter ablation studies conducted on these algorithms previously (Wallace & Si, 2024), 2) The selections reflect those of the original ADP studies (Vrabie & Lewis, 2009; Vamvoudakis & Lewis, 2010; Jiang & Jiang, 2014; Bian & Jiang, 2022), 3) The selections are kept constant across methods wherever possible for sake of consistent benchmarking, and 4) The selections represent natural designer first-choices to illustrate key design insights in the algorithm empirical behavior. For descriptions of the algorithm hyperparameters and learning procedure, see the respective development work

All methods except IRL accommodate probing noise. For SPI, RADP, and CT-VI, we use the probing noise $d(t) = 5\cos(t)$. For these three methods, we initialize the state to the origin $x_0 = 0$. Since IRL does not accommodate probing noise, its only means of excitation is through the initial condition, which we set to $x_0 = [10, \ 10]^T$. For IRL, we run $i^* = 5$, collecting $l = 15$ data samples at a sample period $T_s = 0.1$ s. For SPI, which tunes its weights dynamically online, we tune the weights for $t_f = 500$ s, with tuning gains $\alpha_1 = \alpha_2 = 10$, $F_1 = 0$, and $F_2 = 5I_M$. For RADP, we run $i^* = 10$ iterations, collecting $l = 15$ data samples at a sample period $T_s = 1$ s. For CT-VI, which tunes its weights dynamically at an independent time $s$ to the simulation time $t$, we collect data for $t_f = 50$ s and tune for $s_f = 50$ s.

## M.3   ABLATION STUDY

We first run each algorithm using the tuning hyperparameters discussed in Appendix M.2 on the minimal basis $\phi$ (76). The results of this training can be found in the top row of Figure 18, which plots the critic weight responses for the four seminal ADP algorithms. As can be seen, all methods successfully converge to the optimal weights $c^*$ (78) for the small basis $\phi$ (76). For SPI in particular, this is an independent validation of the convergence results established in the original study (Vamvoudakis & Lewis, 2010). Thus, we have established a performance baseline: All methods successfully converge for these hyperparameters and choice of critic bases. The critic weight responses are well-behaved and exhibit fast convergence.

We then run all four algorithms with the same hyperparameters on the slightly expanded basis $\phi$ (77) and plot the resulting critic weight responses in the bottom row of Figure 18. All methods fail to converge to the optimal weights $c^*$ (79) for the expanded basis $\phi$ (77), even though this basis can achieve exact approximation of the known optimal value $V^*$ (73).

We now turn to the bottom row of Figure 18 for an analysis of the failure modes of each algorithm. For IRL (Vrabie & Lewis, 2009), the weights diverge after $i = 2$ iterations. For SPI (Vamvoudakis & Lewis, 2010), one can see that the weights do not diverge. However, the weight values $c(t)$ after $t \approx 6.75$ s for SPI fail to stabilize the closed-loop system. To see this, we have plotted the corresponding state response $x(t)$ of SPI's training in Figure 19. As can be seen, the state diverges due to the destabilizing SPI policy. For RADP (Jiang & Jiang, 2014), the weights successfully converge; however, RADP fails to converge to the optimal weights $c^*$ (79); indeed, elements $c_4^* = \cdots = c_8^* = 0$, yet examination of the RADP response in Figure 18 reveals that RADP zeros none of these weights. Finally, for CT-VI (Bian & Jiang, 2022), the weights diverge. These weight divergence issues have been exhibited by CT-VI in previous evaluations (Wallace & Si, 2024).

**Conclusion.** The elegant results of ADP-based leading CT-RL methods have made significant contributions to CT-RL. However, further algorithm development work is required to enable these algorithms to synthesize for meaningful applications, beyond second order academic examples with *a priori* known optimal solutions.

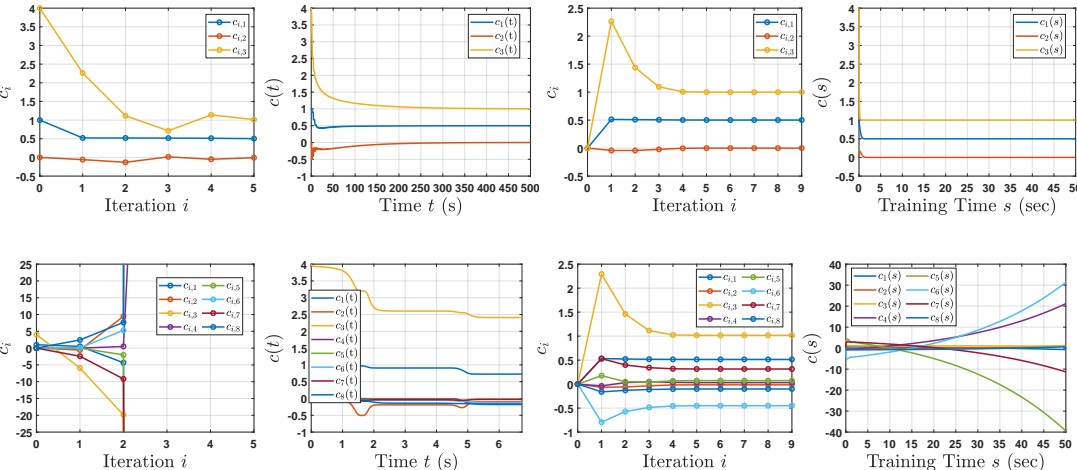

Figure 18: Critic weight responses of ADP methods for minimal basis $\phi$ (76) (top row) and expanded basis $\phi$ (77) (bottom row). First column: IRL, second column: SPI, third column: RADP, fourth column: CT-VI. All methods successfully converge to the optimal weights $c^*$ (78) for the small basis $\phi$ (76), but all fail to converge to the optimal weights $c^*$ (79) for the expanded basis $\phi$ (77).

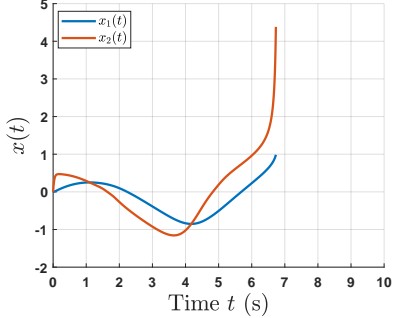

Figure 19: State response $x(t)$ of SPI when training on the expanded basis $\phi$ (77). The state diverges after $t \approx 6.75$ s as the SPI weights fail to stabilize the closed-loop system.

