# OpenReview forum: "Integral Performance Approximation for Continuous-Time Reinforcement Learning Control"
_ICLR.cc/2025/Conference — ICLR 2025 Poster_

### Official Review · Reviewer_K99L · 2024-10-28

**Soundness:** 2
**Presentation:** 3
**Contribution:** 2
**Rating:** 5
**Confidence:** 3

**Summary:**

This paper proposes a continuous-time reinforcement learning control method. The critic network design is novel. However, the theoretical analysis is questionable, which is difficult to follow. Simulation results on three optimal control tasks show that the proposed method outperforms SOTA methods.

**Strengths:**

(1) Extensive simulations are conducted.

(2) The proposed method is robust to model uncertainty empirically.

**Weaknesses:**

(1) The proposed method is model-based, which only considers affine nonlinear systems.

(2) There are some theoretical issues in the paper.

(3) The advantage of the proposed method in improving learning robustness is not analyzed theoretically.

**Questions:**

(1) It is suggested to introduce the value function $V$ (formula (7)) before Section 2.1. In addition, please use different notations to represent the value function (e.g. $V$) and the critic network (e.g. $\hat V$). In this paper, the authors sometimes replace the value function with the critic network in their theoretical analysis.

(2) In (5), the notation $V$ in the function $H$ should be $V^*$.

(3) In Fig. 1, the HJB equation is unrelated to the CT temporal difference equation.

(4) In (13), why do you linearize the controller $\mu (x)$ at $x=0$? For states which are far away from the origin, the linearized function could not approximate $\mu (x)$ accurately.

(5) The reviewers are confused with the content in Appendix C, as formulas (16) and (17) can be obtained directly based on the Bellman equation (8), the critic network (9), and linearized controller (13).

(6) The proof of Theorem 3.1 is hard to follow. It seems the authors overlook that the system considered in this paper is affine nonlinear.

(7) In appendix F, the authors should explain clearly that how the reference command input and the error influence the control policy, which could be an important trick in improving the learning performance of the proposed method.

(8) Since the system is known, the authors are encouraged to further compare their method with some classic control methods which do not consider optimality. In addition, could the authors explain why the SOTA baselines are significantly sample-inefficient?

(9) Is it possible to discuss the learning robustness of the proposed method from a theoretical perspective?

---

> ### Author Response · Authors · 2024-11-19
> **Response to Reviewer (Part 1)**
>
> * We thank this reviewer for all of their time and insightful feedback. Please see our Global Response above, which gives a comprehensive summary of this work and review discussion.
>
>
> ## Weaknesses:
>
> > **1.** The proposed method is model-based, which only considers affine nonlinear systems.
>
> * Thank you for this point. IPA addresses the same affine nonlinear system $(f, g)$ as the leading ADP and SOTA Deep RL FVI methods (Lutter et al., 2021, 2023b).
>
> * Affine nonlinearity actually is very practical, and addresses a wide breadth of realistic physical systems and their controls. This is because that these real-world systems are naturally modeled by the mechanics of Euler-Lagrange, which are coupled second-order ordinary differential equations (ODEs), and which are in the form of control affine nonlinearity. Such mechanisms naturally exist in many real-life systems, such as wearable robots or exoskeletons, robotic manipulators, unmanned aerial or ground vehicles, navigation systems, and many others as long as they rely on manipulation of mechanical apparatus such as an autofocus of cameras and computer hard disks, to name just a few.
>
> * Additionally, control systems in the HSV application and aerospace systems more broadly are most frequently addressed with affine nonlinear dynamics (Dickeson et al., 2009 a-b), (Rodriguez et al., 2008), (Wang \& Stengel, 2000), (Marrison \& Stengel, 2000).
>
> * Affine nonlinear application domains such as the HSV represent exceptionally challenging control problems. Consider balancing a yardstick on your finger, the HSV’s pitch-up instability places hard lower bounds on how “slow” the HSV can be controlled – too slow, and the yardstick topples over. This instability is uniquely combined with nonminimum phase behavior, another profound dynamical challenge which, simply stated, results in the aircraft jerking downward before initiating an upward climb maneuver. This is due to the great parasitic coupling occurring when upward deflections of the tail (which are commanded to pitch the nose of the vehicle upward to initiate a climb) cause a downward force on the vehicle which thereby results in a temporary dip in altitude. Fundamentally, nonminimum phase behavior places hard upper bounds on how “fast” the HSV can be controlled (Bolender and Doman, 2006a). Due to the combination of instability and nonminimum phase behavior, designers are left with a very narrow band of acceptable speeds they can control the aircraft in, resulting in a formidable design problem which still represents an area of open research (Rodriguez et al., 2008). The inherent dynamics of HSVs aside, the extreme speeds involved cause immense difficulty in aerodynamic modeling due to complex aeropropulsive interactions and body flexing under the stresses involved in flight (Wang \& Stengel, 2000), (Marrison \& Stengel, 1998) (Bolender \& Doman, 2005, 2006a-b).
>
> * To be more specific, we have assembled Table R1, which lists the system type and dynamical information used by the SOTA CT-RL methods in DRL and ADP as compared to the proposed IPA method. As can be seen, IPA stands out as least demanding in requiring dynamic information while providing good system responses. Even though some methods in Table R1 (Vrabie \& Lewis, 2009; Jiang \& Jiang, 2014; Bian \& Jiang, 2022) theoretically require less dynamic information than IPA does, they have not been able to demonstrate any meaningful synthesized controllers yet. These ADP methods failed to synthesize for the 2nd order benchmark (see Appendix M for evaluations).
>
> ### Table R1: System dynamical structure and knowledge required by SOTA CT-RL methods
>
> Algorithm|System type|System dynamics required
> |-|-|-|
> IPA|Affine nonlinear|$f, g$
> ||||
> cFVI (Lutter et al., 2021)|Affine nonlinear|$f, g$, $\partial f/\partial x$, $\partial g/\partial x$ (Remark 1)
> rFVI (Lutter et al., 2023b)|Affine nonlinear|$f, g$, $\partial f/\partial x$, $\partial g/\partial x$, $\partial f/\partial \theta$, $\partial g/\partial \theta$ (Remark 1)
> ||||
> IRL (Vrabie \& Lewis, 2009)|Affine nonlinear|$g$
> SPI (Vamvoudakis \& Lewis, 2010)|Affine nonlinear|$f, g$
> RADP (Jiang \& Jiang, 2014)|Affine nonlinear|None (Remark 2)
> CT-VI (Bian \& Jiang, 2022)|Affine nonlinear|None (Remark 2)
>
> **Remark 1:** Note that the deep RL FVIs (Lutter et al., 2021; 2023b) require more dynamics knowledge than IPA, in particular partial derivative knowledge with respect to the state $x$ and model uncertainty parameters $\theta$.
>
> **Remark 2:** IPA and FVIs are edged out by the ADPs (IRL, SPI, RADP, CT-VI) in dynamical information required, but at the cost of not being able to synthesize meaningful controllers (cf. evaluation studies of all four of these methods in Appendix M).

---

> > ### Author Response · Authors · 2024-11-19
> > **Response to Reviewer (Part 2)**
> >
> > * CT-RL for general nonlinearity is still under investigation and is yet to show effectively synthesized control leading to meaningful performance on realistic, challenging nonlinear dynamics. We discussed this very point in our Introduction section (see pp. 2, lines 60-64). More specifically, as discussed in our Introduction, a handful of existing methods exist addressing CT-RL problems under general nonlinear (non-affine) dynamics. While they are important first steps, and some of them may have great potential, this approach requires significant further developments. Additional examples such as CT-VI in Table R1 are methods that were proposed to address general nonlinear dynamics. But as shown in (Wallace \& Si, 2024), these methods fail to synthesize controls for simple 2nd order systems with known closed-loop solutions. A small change in the basis of such small problem led to learning failure. As the latest development, we demonstrate IPA for controlling affine nonlinear dynamics of hypersonic vehicles, much more advanced from any currently available results achieved by RL methods for general nonlinear dynamics.
> >
> > > **2.** There are some theoretical issues in the paper.
> >
> > * We wish this reviewer had specified what theoretical issue they had with the paper. Inference from context, please see our responses below:
> >   * If it is that the results originally submitted did not treat robustness in their theoretical guarantees: We have added robustness theoretical guarantees to the manuscript -- please see our response to your Question 9 below for further details.
> >   * If it is that Theorem 3.1 is difficult to follow: We have addressed this in your Question 6 below. We have also added explanatory material to Theorem 3.1 and throughout the Methods section (see blue additions throughout Section 2) which helps provide the reader further context and better set up the theoretical results.
> >   * If it is the notation-related questions the Reviewer had: We have addressed these in your Questions 1-2 below.
> >   * If it is the link between the HJB equation and the CT temporal difference equation: We have addressed this in your Question 3 below. The two are directly linked.
> >   * If it is that this Reviewer believes the method is linear and its proofs disregard the affine nonlinearity: We use fully-nonlinear policies, and the results of Theorem 3.1 apply to the full affine nonlinear dynamics -- no neglecting affine nonlinearity. Please see our responses to your Questions 4 and 6 below.
> >   * If it is the additional derivations provided in Appendix C: These are necessary to generalize the results to accommodate the probing noise excitation crucial for design. Please see our response to your Question 5 below.
> >
> > > **3.** The advantage of the proposed method in improving learning robustness is not analyzed theoretically.
> >
> > * Thank you for this point, we have addressed it in your Question 9 below. We have added rigorous language on robustness to the revised manuscript.

---

> > > ### Author Response · Authors · 2024-11-19
> > > **Response to Reviewer (Part 3)**
> > >
> > > ## Questions:
> > >
> > > > **1.** It is suggested to introduce the value function $V$ (formula (7)) before Section 2.1. In addition, please use different notations to represent the value function (e.g. $V$) and the critic network (e.g. $\hat{V}$). In this paper, the authors sometimes replace the value function with the critic network in their theoretical analysis.
> > >
> > > * Done, thank you. We have introduced the $\hat{V}$ notation for the critic to differentiate it from the value function $V$ (see blue notation in pp. 3-4). The value function (7) is an equivalent expression of the performance index (2) and it was included at that point in the manuscript as a reproduction of (2) for the reader's convenience -- we have included clarifying language to this effect in the revision (see pp. 3, ln. 146). It directly relates to the CT temporal difference (8) immediately below it, so its location there adds value for the reader.
> > >
> > > > **2.** In (5), the notation $V$ in the function $H$ should be $V^{*}$.
> > >
> > > * Done, thank you. See the revised (5) on pp. 3, ln 133.
> > >
> > > > **3.** In Fig. 1, the HJB equation is unrelated to the CT temporal difference equation.
> > >
> > > * We have fixed the typo in Fig. 1's equation reference to the HJB equation -- it now points to the correct equation number. Thanks for bringing this to our attention.
> > >
> > > * We have also added clarifying language in the manuscript (see pp. 3, ln. 143-159), to show that the two are, indeed, directly conceptually related.
> > >
> > > * In short, a function $V$ satisfies the CT temporal difference equation if and only if it satisfies the Generalized HJB (GHJB) equation (6) (Beard \& McLain, 1998). The GHJB equation, in turn, collapses to the HJB equation at equality when the policy is optimal (i.e., $V = V^{*}$). Thus, the CT temporal difference and the HJB equation are equivalent characterizations of optimality.

---

> > > > ### Author Response · Authors · 2024-11-19
> > > > **Response to Reviewer (Part 4)**
> > > >
> > > > > **4.** In (13), why do you linearize the controller $\mu(x)$ at $x = 0$? For states which are far away from the origin, the linearized function could not approximate $\mu(x)$ accurately.
> > > >
> > > > * Thank you for this point. We would like to emphasize that the IPA policies $\mu_{i}$ (18):
> > > > \begin{align}
> > > > \textstyle
> > > > \mu_{i+1}(x) = - \frac{1}{2} R^{-1} g^{T}(x) \frac{\partial \hat{V}}{\partial x}(x)
> > > > \hspace{1.5in} (18)
> > > > \end{align}
> > > > are **already fully nonlinear policies.**
> > > >
> > > > * The linear matrices $K_{i}$ (13):
> > > > \begin{align}
> > > > \textstyle K_{i}  \triangleq - \left. \frac{\partial}{\partial x} \\{ \mu_{i}(x) \\} \right|_{x = 0}
> > > > \hspace{1.5in} (13)
> > > > \end{align}
> > > > are used for approximating the integral performance via (14) only:
> > > >
> > > > \begin{align}
> > > > \int_{t_{0}}^{t_{1}} x^{T} Q x + \mu_{i}^{T}(x) R \mu_{i}(x) \\, d\tau \approx \int_{t_{0}}^{t_{1}} x^{T} \big( Q + K_{i}^{T} R K_{i} \big) x \\, d\tau
> > > > \hspace{1.5in} (14)
> > > > \end{align}
> > > > **they are not the IPA policies.**
> > > >
> > > > * On changing the equilibrium point: As is standard in optimal control and CT-RL formulations, we assume an equilibrium at $x = 0$. This can readily generalize to arbitrary equilibria $x_{e} \in \mathbb{R}^{n}$ via a change of variables.
> > > >
> > > > * If $\dot{z} = \tilde{f}(z) + \tilde{g}(z) u$ is any affine nonlinear system with equilibrium $z = x_{e}$, then consider the change of variables $x = z - x_{e}$ and translated dynamics $f(x) = \tilde{f}(x + x_{e})$, $g(x) = \tilde{g}(x + x_{e})$. This translated system $(f, g)$ has an equilibrium at $x = 0$, and we note that its dynamics satisfy
> > > >
> > > > \begin{align}
> > > > \dot{x} = \frac{d}{dt} \\{z - x_{e}\\} = \frac{d}{dt} \\{z\\} = \tilde{f}(z) + \tilde{g}(z) u = \tilde{f}(x + x_{e}) + \tilde{g}(x + x_{e}) u = f(x) + g(x) u
> > > > \end{align}
> > > >
> > > > * i.e., $f(x) + g(x) u = \tilde{f}(z) + \tilde{g}(z) u$, so we may apply IPA to nonlinear systems with  nonzero equilibria.
> > > >
> > > > * Performing integral performance approximation at $x = 0$ is by design for using trajectory data in learning with high data efficiency. Strictly speaking, the Taylor expansion about an arbitrary $x_{0} \in \mathbb{R}^{n}$ goes $\mu_{i}(x_{0} + x) = \mu_{i}(x_{0}) + \left. \frac{\partial \mu_{i}}{\partial z} \right|\_{z = x_{0}} (x - x_{0}) + \text{H.O.T.}$. For $x_{0} = 0$, this collapses nicely: $\mu_{i}(0 + x) = \mu_{i}(0) +  \left. \frac{\partial \mu_{i}}{\partial z} \right|\_{z = 0} (x - 0) + \text{H.O.T.} = - K_{i} x + \text{H.O.T.}$. The zero evaluation of the policy at $x_{0} = 0$ is what enables the algebra of the data reuse. Having to store $\mu_{i}(x_{0})$ across a grid of $x_{0}$ would require re-integrating the policy value at each iteration $i$ (precisely what IPA is intended to bypass).
> > > >
> > > >
> > > > > **5.** The reviewers are confused with the content in Appendix C, as formulas (16) and (17) can be obtained directly based on the Bellman equation (8), the critic network (9), and linearized controller (13).
> > > >
> > > > * Thank you for this point. We have included clarifying language in the revision (see pp. 16, ln. 838-844) to reflect the following:
> > > >
> > > > * The crux of Appendix C is to generalize the results presented in the Methods section, which assume no excitation: $u(t) = \mu_{i}(x(t))$, to include excitation signals such as probing noise: $u(t) = \mu_{i}(x(t)) + d(t)$. The Bellman equation (8) only holds if the control $u(t) = \mu_{i}(x(t))$ is applied without excitation signals, yet these excitation signals are necessary for good data/learning quality and thus need to be accommodated by the IPA framework.
> > > >
> > > > > **6.** The proof of Theorem 3.1 is hard to follow. It seems the authors overlook that the system considered in this paper is affine nonlinear.
> > > >
> > > > * Please see our response to your Question 4 above. IPA uses the fully-nonlinear policies $\mu_{i}$ (18), which are updated via the learning matrices $\Theta_{i}$, $\Xi_{i}$ (33-34) constructed from nonlinear state-action trajectory data $(x, u)$ generated by the actual nonlinear environment. As a result, Theorem 3.1 applies directly to the affine nonlinear system. The proof does not ignore any of the affine nonlinearity. The matrices $K_{i}$ (13) are not the IPA policies.
> > > >
> > > > * Furthermore, to help clarify the results of Theorem 3.1, we have:
> > > >
> > > >   1. Added approximation error bounds throughout the text, including Theorem 3.1). Here we have replaced $\approx$ with e.g., $o(||x||)$ for the approximation involving $K_{i}$ following (13), $o(||x||^{2})$ for the approximations involving IPA in (14), (15).
> > > >
> > > >   2. Along this same vein, these results apply to the nonlinear system and are local, to be precise. We have included clarifying language to this effect in Theorem 3.1.
> > > >
> > > > * The terms $A$, $B$, $K_{i}$ appearing in the proof of Theorem 3.1 are present 1) because they are lumped directly into the nonlinear IPA learning matrices $\Theta_{i}$, $\Xi_{i}$ (33-34) used for the learning update, and 2) because they have been introduced via the IPA equation (14).

---

> > > > > ### Author Response · Authors · 2024-11-19
> > > > > **Response to Reviewer (Part 5)**
> > > > >
> > > > > > **7.** In appendix F, the authors should explain clearly that how the reference command input and the error influence the control policy, which could be an important trick in improving the learning performance of the proposed method.
> > > > >
> > > > > * We have included an additional explanation of the effects of the reference command in Appendix F, including new figures to help illustrate the advantages of the reference command input. Thanks for this suggestion.
> > > > >
> > > > > > **8.** Since the system is known, the authors are encouraged to further compare their method with some classic control methods which do not consider optimality. In addition, could the authors explain why the SOTA baselines are significantly sample-inefficient?
> > > > >
> > > > > * On comparing to methods without optimality criteria: This would be an unfair "apples-to-oranges" comparison biased toward our method. IPA is specifically formulated to optimize its associated cost function, so it has an inherent advantage over methods which do not optimize to any cost. Furthermore, such classical methods are already extensively studied in the literature for many decades now, and they hardly represent new, SOTA results for ICLR 2025.
> > > > >
> > > > > * On comparing to other model-based methods: We would like to remind this reviewer that the SOTA FVI methods we compared against are model-based and require full dynamics knowledge just like IPA with extra assumptions (see Table R1 in our response to your Weaknesses point 1 above).
> > > > >
> > > > > * In addition, we have included evaluation of the leading ADP CT-RL works in Appendix M, which are also model-based. We have also included clearer reference to Appendix M in the revision so that the reader can see additional methods compared.
> > > > >
> > > > > * On sample efficiency of FVIs: Thank you for this question, we have included a discussion in the revision (see pp. 1, ln 36-39) to reflect the following.
> > > > >   * We would first like to emphasize that the FVI works (Lutter et al., 2021, 2023b) stand as the SOTA CT-RL results to-date, achieving significant performance improvements over the leading ADP CT-RL works studied in Appendix M. This approach has demonstrated substantial empirical results, both in the original FVI works and in the present work.
> > > > >   * These methods utilize 1) a nominal model of the system (much like IPA) + 2) large amounts of closed-loop trajectories to learn the optimal value function from data.
> > > > >   * However, this comes at the usual deep RL cost of needing to train a large network ($\approx 80,000$ weights), which requires a large amount of data.
> > > > >   * This is an algorithm design trade rather than an inherent limitation of the FVI methods.
> > > > >
> > > > > > **9.** Is it possible to discuss the learning robustness of the proposed method from a theoretical perspective?
> > > > >
> > > > > * Yes, this is a great suggestion! In fact, IPA inherits substantial local stability robustness properties owing to its structural parallels to Kleinman's algorithm. We have added the following result verbage to Theorem 3.1:
> > > > >
> > > > > > ... As a result, IPA inherits the guaranteed local stability robustness margins of Kleinman's algorithm:
> > > > > >* $||S_{u}||\_{\mathcal{H}^\infty} \leq 0$ dB
> > > > > >* $||T_{u}||\_{\mathcal{H}^\infty} \leq 2$ dB
> > > > > >
> > > > > > where $||\cdot||\_{\mathcal{H}^\infty}$ denotes the $\mathcal{H}^\infty$ norm, and where $S_{u}$, $T_{u}$ denote the sensitivity and complementary sensitivity closed-loop maps, respectively, at the control loop breaking point $u$ (Rodriguez, 2004). We have also added a definition of terms and discussion of these results to Appendix F of the revision (see in particular the newly-added Appendix F.2 on IPA robustness).

---

> ### Comment · Reviewer_K99L · 2024-11-23
>
> Thanks for providing details responses which solve some of my questions. Sorry for the late reply. We have tried our best to understand the contributions of this work. However, based on our research experiences on ADP and optimal control, we believe this work is not outstanding, and the key contributions are the design of the quadratic critic network, and the update of the critic parameters based on the linearized control policy, which results in excellent empirical results. Some additional comments are as follows:
>
> (1) It is a common sense that the value function is differnet from the objective function. The value function is a function of any state $s$, while the state in the objective function is an initial state (may be sampled from some initial distributions). As a result, we usually use different notations.
>
> (2) We know that the nonlinear control policy is used to sample trajectories, while the linearized policy is employed to approximate the integral performance (which is related to the critic update). However, our problem remains because the critic cannot be accurately estimated due to linearization errors when the state is far from the origin.
>
> In addition, the theoretical contribution of this work is questionable, as it primarily relies on Theorem B.1.If you believe that your work makes sufficient theoretical contributions to optimal control, you may consider submitting it to TAC or Automatica for more specialized feedback.
>
> Our decision on this work is final.

---

> > ### Author Response · Authors · 2024-11-24
> > **We thank this Reviewer. We have carefully reviewed each and every comment from the Reviewer, and we address them thoroughly and one by one below. But we cannot help noticing that some critical comments from the Reviewers are opinions. Instead, we wish the reviewer had supplied us with concrete and actionable arguments pertaining to the actual technical content of our work. In any event, we have responded to your arguments with direct evidence and technically-backed refutation. (1 of 3)**
> >
> > > Thanks for providing details responses which solve some of my questions. Sorry for the late reply. We have tried our best to understand the contributions of this work. However, based on our research experiences on ADP and optimal control, we believe this work is not outstanding ...
> >
> > * Given this Reviewer's background research experiences on ADP, we would expect that they are well-aware of the substantial performance limitations facing the existing leading ADP CT-RL methods. We are perplexed by, on one hand, that the reviewer sees:
> >
> >   * The existing leading ADP CT-RL works, which fail on second-order toy problems,
> >
> >   and, on the other, sees:
> >
> >   * The proposed IPA method, which offers a substantial suite of evaluations with demonstrated control performance on perhaps the most complex hypersonic vehicle environment studied in CT-RL to-date
> >
> >   and deems the proposed IPA work "not outstanding" -- This is the Reviewer's opinion, which we respect. However, we would like to point out that our results demonstrate, beyond opinion, concrete evidence of substantial new results in CT-RL control design with theoretical insights.
> >
> >
> > * We would like to mention again that ADP performance limitations have been  brought to the discussion by, for example, the below independent sources:
> >
> >   1. The work of (Wallace \& Si, 2024) which was the original in-depth performance evaluation work studing the leading CT-RL methods. This work showed that these methods fail to synthesize controls for simple second-order systems with known closed-form solutions. A small change in the basis of such small problem led to learning failure.
> >   2. The reference [1] (appeared in the same ICRL venue) brought to the discussion by Reviewer 5RXP, which independently verifies the performance issues raised in (Wallace \& Si, 2024).
> >   3. The Appendix M of this work, which includes yet another performance evaluation also on a second-order academic benchmark. Similarly, all the leading ADP methods are shown to fail for small changes to the parameters.
> >
> > * The reviewer may have their gauge for what is important for ICLR. For communication purposes, we would like to share our perspective in the meantime. ICLR appreciates results which don't just sound good theoretically on paper, but these also must stand up to the scrutiny of in-depth evaluation and reproducibility by other researchers. We provide both.
> >
> > > ... and the key contributions are the design of the quadratic critic network, and the update of the critic parameters based on the linearized control policy, which results in excellent empirical results.
> >
> > * Firstly -- as we clearly presented in the paper, we constructively formulated our bases to leverage the quadratic cost structure of the control problem. We are not the only one using a quadratic objective performance measure, but we used it **creatively**, together other design innovations. To provide a context, consider, for example the seminal IRL ADP method (Vrabie \& Lewis, 2009), which leverages the same cost. If this quadratic cost is our *only* contribution, how then is the proposed IPA method so data efficient? How then do the ADPs struggle to synthesize for simple examples, while IPA demonstrates significant performance on a variety of applications including the substantial hypersonic vehicle?
> >
> > * The quadratic bases are by no means the only feature that has enabled IPA. On data efficiency for example, to further compare with (Vrabie \& Lewis, 2009), the algebra derived for the IPA update matrix $\Xi_{i}$ (17), enabled by Kleinman's structure, allows IPA to reuse the same trajectory data collected under the initial stabilizing policy $\mu_{0}$ for the generation of the entire learning sequence $\\{\mu_{i}\\}\_{i=1}^{\infty}$. This is in contrast to (Vrabie \& Lewis, 2009), which for iteration $i$ requires state-action data to be collected under the stabilizing policy $\mu_{i}$ before updating to $\mu_{i+1}$. Data from previous iterations is discarded. This data reuse enables IPA to be significantly data efficient compared to previous methods.
> >
> > * Consider also the relaxed PE condition IPA features: Full column rank $\underline{n} \triangleq \frac{n (n+1)}{2}$ of the IPA matrix $\mathcal{I} \in \mathbb{R}^{l \times \underline{n}}$ (20). This assumption is easy to satisfy, as the algorithm can continue to collect $l > \underline{n}$ samples until it is met. It is also virtually instantaneous to verify (matrix rank calculation). We observe no issue satisfying this assumption on all the environments evaluated, including the complex HSV. Such a constructive requirement is novel and enables high IPA data efficiency and control performance.
> >
> > * "Excellent empirical results" are a testimony of how, after hundreds of ADP CT-RL publications, IPA actually delivers substantial and demonstrated performance with theoretical insight. This is a significant result for CT-RL.

---

> > > ### Author Response · Authors · 2024-11-24
> > > **Response to Reviewer (2 of 3)**
> > >
> > > > Some additional comments are as follows:
> > > >
> > > > (1) It is a common sense that the value function is differnet from the objective function. The value function is a function of any state, while the state in the objective function is an initial state (may be sampled from some initial distributions). As a result, we usually use different notations.
> > >
> > > * We are already on the same page here. Please refer to our response to your original Question 1.
> > >
> > > > (2) We know that the nonlinear control policy is used to sample trajectories, while the linearized policy is employed to approximate the integral performance (which is related to the critic update). However, our problem remains because the critic cannot be accurately estimated due to linearization errors when the state is far from the origin.
> > >
> > > On: "..the critic cannot be accurately estimated due to linearization errors when the state is far from the origin":
> > >
> > > 1. We clearly acknowledged that critic estimation introduces linearization errors as we specifically said in Theorem 3.1. We even included  bounds on the approximation error: first-order $o(||x||)$ for the nonlinear policies $\mu_{i}$ and stronger second-order $o(||x||^{2})$ for the critic $\hat{V}$. Our theoretical results precisely reflected that, and we were careful not to overstate this theoretical result.
> > >
> > > 2. The Reviewer's deep concern of "the state is far from the origin" is addressed by our nonlinear policy of the form (4), which is the policy that interacts with the environment. Or in other words, learning takes place to address nonlinearity in the environment among other factors such as linearization error and unmodeled dynamics.
> > >
> > > * We already specifically considered examination of approximation performance. We explicitly studied this for all environments, including the highly complex HSV. Please refer to  Figure 5 in the manuscript, and please note that IPA exhibits the BEST approximation performance of the three methods studied.
> > >
> > > * To be even more specific, we want to be explicit here: The reviewer's concern of **states far from the origin is already refuted** by extensive and systematic evaluations we have conducted -- in short, we focused our studies on states $||x|| >> 0$ far from the origin.
> > >
> > >   1. For the complex HSV environment, evaluations of the initial flight path angles of $\pm 1^{\circ}$ and airspeeds $\pm 100$ ft/s -- **are well away from the trim equilibrium and in the nonlinear regime.**
> > >
> > >   2. In addition, we focus evaluations of the pendulum environment (Figure 3, Table 2) on the swing-up task at the **full pendulum displacement $\theta = 180^{\circ}$ -- the furthest from the upright equilibrium $\theta = 0^{\circ}$ this system can get, and the region with the heaviest nonlinearity.** Our evaluation domain used to generate the cost data in Table 2 is generated over the full displacement distribution $\mathcal{U}(-180^{\circ},180^{\circ})$.

---

> > > > ### Author Response · Authors · 2024-11-24
> > > > **Response to Reviewer (3 of 3)**
> > > >
> > > > > In addition, the theoretical contribution of this work is questionable, as it primarily relies on Theorem B.1.If you believe that your work makes sufficient theoretical contributions to optimal control, you may consider submitting it to TAC or Automatica for more specialized feedback.
> > > >
> > > > * We, with great care and technical detail, already responded to the Reviewer's concerns of the theoretical results in their Question 6 of the original Review. Without providing any specifics, nor technical follow-up to our rebuttal, the Reviewer instead resorted to a vague claim that the results are "questionable". We find ourselves having difficulty to be convinced what is "questionable". For example, our newly included stability robustness results have nothing to do with Theorem B.1. We leverage on the Kleinman structure (with its important properties in B.1), but the theoretical result and its proof is specific to the proposed IPA.
> > > >
> > > > * We are confused by this Reviewer's stance on discussion of theoretical results in the ICLR venue, as they seem to have simultaneously:
> > > >
> > > >   1. Specifically requested additional theoretical results on stability robustness be added to this ICLR work. Indeed, this Reviewer made the lack of such results one of their central Weaknesses of the proposed work.
> > > >   2. When we have added the stability robustness results that this Reviewer requested, we received no follow-up discussion of the robustness results added.
> > > >   3. Instead, after we provided the theoretical results the Reviewer requested, the Reviewer now holds that works wishing to present theoretical results should opt-out of ICLR and instead "...consider submitting it to TAC or Automatica for more specialized feedback."
> > > >
> > > > * We are actually troubled by the implications of this assertion -- should including theoretical results be automatic grounds to avoid ICLR, then?
> > > >
> > > > * If on the other hand, theoretical results in venues like TAC/Automatica are the end-all-be-all gold standard for the Reviewer's perspective -- consider the Nobel Prize in physics, perhaps the highest award in science. Even Nobel-Prize theories require experimentally verified before the award is given.
> > > >
> > > > > Our decision on this work is final.
> > > >
> > > > * We sincerely appreciate the Reviewers’ effort in reading our paper. Our perspectives are different so our conclusions differ. But we respect your choice. Thank you.

---

> ### Comment · Reviewer_ZBUg · 2024-11-27
> **Further Comments from Reviewer ZBUg**
>
> Being the other reviewer (Reviewer ZBUg) reading this thread, I would like to offer my comments on this discussion:
>
> 1. I disagree with this reviewer's comment, "if you believe that your work makes sufficient theoretical contributions to optimal control, you may consider submitting it to TAC or Automatica for more specialized feedback". ICLR explicitly calls for contributions to the areas of reinforcement learning, robotics, autonomy, and planning. I do not believe a paper being appropriate for TAC or Automatica is an automatic ground for rejection at ICLR. I wish the ICLR community was more open-minded and accepting of interdisciplinary results.
>
> 2. Although I increased my score to 8, this reviewer does have a valid point that "the critic cannot be accurately estimated due to linearization errors when the state is far from the origin". The authors responded stating they do acknowledge this fact and mentioned they incorporate higher order terms in the Taylor series. However, the higher order terms are still not incorporated explicitly in Eq. (30). The approximation in Eq. (25) was expressed as a limit, which is then carried forward to (30). The higher order terms, if explicitly treated as a perturbation, would show up in the integral. As $t_1$ increases, the integral of this perturbation term would grow and might end up being unbounded. In this case, the integral approximation might no longer hold. The authors are suggested to include a more rigorous treatment for this issue. By that, I mean construct an explicitly defined neighborhood (with a predefined radius) around the equilibrium, and do a more rigorous perturbation analysis. The higher order terms can be treated as a bounded perturbation (due to boundedness of higher-order terms in a bounded set) for this purpose. The effects of this perturbation on the performance can be explicitly computed. I understand the authors may not have the time to make a revision to the manuscript at this point, but posting it as a comment here would be sufficient to make a convincing argument to address this reviewer's comment. In the current state, this part of the result holds only in an infinitesimally small neighborhood of the origin.
>
> 3. I condemn the unprofessional tone in the reviewer-author exchanges on this thread.

---

> > ### Author Response · Authors · 2024-11-28
> > **Our Warm Regards on your Thanksgiving Holiday**
> >
> > We thank these Reviewers greatly for all of their thoughtful feedback. We are currently back home with family for Thanksgiving, we'll resume the conversation after getting back in office. We hope that the Reviewers have a wonderful and safe Thanksgiving with friends and family.
> >
> > Warm regards,
> >
> > The authors

---

> > > ### Author Response · Authors · 2024-11-30
> > > **We thank the Reviewers for continuing the conversation, as it has greatly helped us improve our work. We rigorously address your questions on the IPA integral value to prove that 1) it is indeed bounded, as requested by the Reviewers, and 2) further that the designer has nice properties and upper bounds on the error available to them (1 of 3).**
> > >
> > > > I disagree with this reviewer's comment, "if you believe that your work makes sufficient theoretical contributions to optimal control, you may consider submitting it to TAC or Automatica for more specialized feedback". ICLR explicitly calls for contributions to the areas of reinforcement learning, robotics, autonomy, and planning. I do not believe a paper being appropriate for TAC or Automatica is an automatic ground for rejection at ICLR. I wish the ICLR community was more open-minded and accepting of interdisciplinary results.
> > >
> > > * We thank the reviewer for carefully reviewing our paper and for many thoughtful and inspiring questions and comments. We benefited greatly from the discussions. We would also like to sincerely thank all reviewers for giving important comments and  feedback, based on which we were able to improve our paper.
> > >
> > > > Although I increased my score to 8, this reviewer does have a valid point that "the critic cannot be accurately estimated due to linearization errors when the state is far from the origin". The authors responded stating they do acknowledge this fact and mentioned they incorporate higher order terms in the Taylor series. However, the higher order terms are still not incorporated explicitly in Eq. (30). The approximation in Eq. (25) was expressed as a limit, which is then carried forward to (30).
> > >
> > > * We thank the Reviewer for pointing out that we did not include an approximation error bound in (30). We will make sure to do so and include the relevant materials from this discussion in the final version of the manuscript. Specifically in consideration of Equation (30), actually one can directly express it as follows without having to carry it through from (25):
> > >
> > > \begin{align}
> > > & -2 \int_{t_{0}}^{t_{1}}  ( f(x) + g(x) u - \xi(x))^{T} \, P_{i} x  \\, d\tau + \left[ x^{T}(t_{1}) P_{i} x(t_{1}) - x^{T}(t_{0}) P_{i} x(t_{0}) \right]  \approx - \int_{t_{0}}^{t_{1}} x^{T} Q x + \mu_{i}^{T}(x) R \mu_{i}(x) \\, d\tau \qquad\quad \text{(to \\, $o(||x||)$)}.
> > > \end{align}
> > >
> > > > The higher order terms, if explicitly treated as a perturbation, would show up in the integral. As $t_{1}$ increases, the integral of this perturbation term would grow and might end up being unbounded. In this case, the integral approximation might no longer hold. The authors are suggested to include a more rigorous treatment for this issue. By that, I mean construct an explicitly defined neighborhood (with a predefined radius) around the equilibrium, and do a more rigorous perturbation analysis. The higher order terms can be treated as a bounded perturbation (due to boundedness of higher-order terms in a bounded set) for this purpose. The effects of this perturbation on the performance can be explicitly computed. I understand the authors may not have the time to make a revision to the manuscript at this point, but posting it as a comment here would be sufficient to make a convincing argument to address this reviewer's comment. In the current state, this part of the result holds only in an infinitesimally small neighborhood of the origin.
> > >
> > > * We thank the Reviewer for this thoughtful point. Below we will provide an analysis to flesh out the effect of the high order terms, alongside design-related insights.
> > >
> > > * First, increasing the horizon $t_{1}$ will not increase the integral cost indefinitely; in fact, it asymptotically approaches an upper bound as $t_{1} \rightarrow \infty$. To see why this is so, we first note that IPA generates its trajectory data $\\{x(t)\\}\_{t \in [t_{0}, \infty)}$ under the initially feasible policy $\mu_{0}$, this set of data is re-used to generate the sequence of policies $\\{\mu_{i}\\}\_{i=1}^{\infty}$. Given a compact set $\Omega \subset \mathbb{R}^{n}$ of interest tailored to the designer's application-specific needs, the designer can use whichever method they see fit to design the initial policy $\mu_{0}$ to achieve this domain of attraction. They can then select any initial condition $x(t_{0}) \in \Omega$ to collect the data $\\{x(t)\\}\_{t \in [t_{0}, \infty)}$ and be assured that 1) the solution is stable, and 2) that the cost $V(x(t_{0})) = \int_{t_{0}}^{t_{1}} x^{T} Q x + \mu_{0}^{T}(x) R \mu_{0}(x) \\, d\tau$ is finite, namely, $\mu_{0}$ is feasible.

---

> > > > ### Author Response · Authors · 2024-11-30
> > > > **Response to Reviewer (2 of 3).**
> > > >
> > > > * We now claim that, for the same compact set $\Omega \subset \mathbb{R}^{n}$ of interest, the value integral associated with all subsequent policies $\\{\mu_{i}\\}\_{i=1}^{\infty}$ is also finite over $\Omega$, and further that an upper bound independent of the horizon $t_{1} > t_{0}$ can be derived. First note that, by virtue of the trajectory $\\{x(t)\\}\_{t \in [t_{0}, \infty)}$ being generated under the initial feasible policy $\mu_{0}$ which achieves finite cost, the integral $\int_{t_{0}}^{\infty} ||x||^{2} \\, d \tau$ is finite. Now, IPA's choice of critic structure $\hat{V}$ (9) alongside $g(x)$ being Lipschitz assures that the policies $\mu_{i}$ (18) are Lipschitz. Given also that convergence of the critic weights $\\{c_{i}\\}\_{i=1}^{\infty}$ has already been established by Theorem 3.1 independently of this result (indeed, we showed that $c_{i} = v(P_{i})$, where $\\{P_{i}\\}\_{i=1}^{\infty}$ is as generated by Kleinman's algorithm), we then can find a Lipschitz constant $\alpha > 0$ such that $||\mu_{i}(x)|| \leq \alpha ||x||$ over $x \in \Omega$ for all $i \geq 0$. We thus have given any $i \geq 0$ that
> > > >
> > > > \begin{align}
> > > > 	\int_{t_{0}}^{t_{1}} x^{T} Q x + \mu_{i}^{T}(x) R \mu_{i}(x) \\, d\tau
> > > > 	\leq
> > > > 	(||Q|| + \alpha^{2} ||R||) \int_{t_{0}}^{t_{1}} ||x||^{2} \\, d\tau
> > > > 	\leq
> > > > 	(||Q|| + \alpha^{2} ||R||) \int_{t_{0}}^{\infty} ||x||^{2} \\, d\tau.
> > > > \end{align}
> > > >
> > > > * Note that the leftmost bound in the above is independent of $t_{1} > t_{0}$. Thus, there is an upper bound on the value integral, regardless of the designer's selection of the horizon $t_{1}$.
> > > >
> > > > * The same holds for the IPA approximation of the cost, and a proof for this is as follows. We first expand the Taylor series:
> > > >
> > > > \begin{align}
> > > > 	x^{T} Q x + \mu_{i}^{T}(x) R \mu_{i}(x)
> > > > 	&=
> > > > 	x^{T} Q x + (- K_{i} x + r(x))^{T} R (- K_{i} x + r(x))
> > > > 	\\\\
> > > > 	&=
> > > > 	x^{T} (Q + K_{i}^{T} R K_{i}) x - 2 r^{T}(x) R K_{i} x + r^{T}(x) R r(x)
> > > > \end{align}
> > > >
> > > > * where the remainder $r : \mathbb{R}^{n} \rightarrow \mathbb{R}^{m}$ satisfies $\lim_{x \rightarrow 0} \frac{||r(x)||}{||x||} = 0$. Indeed, for *any* $a > 0$ we can choose $b > 0$ such that $||r(x)|| < a ||x||$ for all $||x|| < b$. Meanwhile, since $\mu_{0}$ is feasible, there exists a $t^{\prime} \geq t_{0}$ such that $||x(t)|| < b$ for all $t \geq t^{\prime}$. Thus, returning to the IPA integral, we have
> > > >
> > > > \begin{align}
> > > > 	\int_{t_{0}}^{t_{1}} x^{T} (Q + K_{i}^{T} R K_{i}) x \\, d\tau
> > > > 	&=
> > > > 	\int_{t_{0}}^{t_{1}} x^{T} Q x + \mu_{i}^{T}(x) R \mu_{i}(x) + 2 r^{T}(x) R K_{i} x - r^{T}(x) R r(x) \\, d\tau
> > > > 	\\\\
> > > > 	&\leq
> > > > 	\int_{t_{0}}^{t_{1}} x^{T} Q x + \mu_{i}^{T}(x) R \mu_{i}(x) \\, d\tau
> > > > 	+ 2 \int_{t_{0}}^{t_{1}} r^{T}(x) R K_{i} x \\, d\tau
> > > > 	\\\\
> > > > 	&=
> > > > 	\int_{t_{0}}^{t_{1}} x^{T} Q x + \mu_{i}^{T}(x) R \mu_{i}(x) \\, d\tau + 2 \int_{t_{0}}^{t^{\prime}} r^{T}(x) R K_{i} x \\, d\tau + 2 \int_{t^{\prime}}^{t_{1}} r^{T}(x) R K_{i} x \\, d\tau
> > > > 	\\\\
> > > > 	&\leq
> > > > 	\int_{t_{0}}^{t_{1}} x^{T} Q x + \mu_{i}^{T}(x) R \mu_{i}(x) \\, d\tau + 2 \int_{t_{0}}^{t^{\prime}} r^{T}(x) R K_{i} x \\, d\tau + 2 a ||R|| ||K_{i}|| \int_{t^{\prime}}^{t_{1}} ||x||^{2} \\, d\tau
> > > > \end{align}
> > > >
> > > > * Note that the preceding holds for *any* $t_{1} > t_{0}$. Indeed, let $t_{1} \rightarrow \infty$ in the above, and we have from the bound on the value integral derived above that
> > > >
> > > > \begin{align}
> > > > 	\int_{t_{0}}^{\infty} x^{T} (Q + K_{i}^{T} R K_{i}) x \\, d\tau
> > > > 	&\leq
> > > > 	\int_{t_{0}}^{\infty} x^{T} Q x + \mu_{i}^{T}(x) R \mu_{i}(x) \\, d\tau + 2 \int_{t_{0}}^{t^{\prime}} r^{T}(x) R K_{i} x \\, d\tau + 2 a ||R|| ||K_{i}|| \int_{t^{\prime}}^{\infty} ||x||^{2} \\, d\tau
> > > > \end{align}
> > > >
> > > > * where we note, as previously, that the integral $\int_{t^{\prime}}^{\infty} ||x||^{2} \\, d\tau$ is finite. **Thus, the IPA approximation framework is guaranteed to be bounded.**
> > > >
> > > > * This is an important guarantee to have from the perspective of a designer, as it leaves them the freedom to choose $t_{1}$ based on their application-specific knowledge of the problem (e.g., system bandwidth).

---

> > > > > ### Author Response · Authors · 2024-11-30
> > > > > **Response to Reviewer (3 of 3).**
> > > > >
> > > > > * **Aside -- Refinement of the Above Result + Insights.** While the above directly addresses the issue raised by the Reviewer, it might be interesting to us all to note that a refinement to the above bound is available given IPA's choice of policy structure $\mu_{i}$ and critic structure $\hat{V}$. Indeed, it can be confirmed that $r(x)$ is available in closed-form as
> > > > >
> > > > > \begin{align}
> > > > > 	r(x) = \mu_{i}(x) - (-K_{i} x) = - R^{-1} \big(g(x) - B \big)^{T} P_{i} x
> > > > > \end{align}
> > > > >
> > > > > * so given an invariant compact set $\Omega \subset \mathbb{R}^{n}$ of interest containing the origin, the designer can immediately calculate $d = \max_{x \in \Omega} ||g(x) - B||$ from the model data. Then we note that the following holds, applying the form of $r(x)$ in the above:
> > > > >
> > > > > \begin{align}
> > > > > 	||r^{T}(x) R K_{i} x|| \leq d ||P_{i}|| ||K_{i}|| ||x||^{2}, \qquad \forall x \in \Omega
> > > > > \end{align}
> > > > >
> > > > > * Thus, applying this to the preceding result, we have
> > > > >
> > > > > \begin{align}
> > > > > 	\int_{t_{0}}^{t_{1}} x^{T} (Q + K_{i}^{T} R K_{i}) x \\, d\tau
> > > > > 	&\leq
> > > > > 	\int_{t_{0}}^{t_{1}} x^{T} Q x + \mu_{i}^{T}(x) R \mu_{i}(x) \\, d\tau + 2 d ||P_{i}|| ||K_{i}|| \int_{t_{0}}^{t_{1}} ||x||^{2} \\, d\tau
> > > > > \end{align}
> > > > >
> > > > > * This is an important practical result to designers, as it no longer requires finding the theoretically-guaranteed neighborhood $b > 0$ such that $||r(x)|| < a ||x||$ for all $||x|| < b$. All of the data is available to the designer to calculate this bound on the IPA error.
> > > > >
> > > > > * **Aside:** Indeed, the only condition on data our method requires to guarantee full convergence of the policies $\\{\mu_{i}\\}\_{i=1}^{\infty}$ is the full column rank $\underline{n} \triangleq \frac{n (n+1)}{2}$ of the IPA matrix $\mathcal{I} \in \mathbb{R}^{l \times \underline{n}}$ (20) -- once this rank condition is met, Theorem 3.1 guarantees local convergence, optimality, and closed-loop stability. This assumption is easy to satisfy, as the algorithm can continue to collect $l > \underline{n}$ samples until it is met. It is also virtually instantaneous to verify (matrix rank calculation).
> > > > >
> > > > > * To empirically corroborate this argument via extensive evaluations as reported in the paper, we observe no issue satisfying the rank assumption on all the environments studied in practice  -- even on the complex HSV, at different model uncertainty levels $\nu$. We neither observe an issue with  system initial condition $x_{0}$, as we extensively evaluated on initial states that are far from the origin. For the HSV, evaluations study initial flight path angles of $\pm 1^{\circ}$ and airspeeds $\pm 100$ ft/s, and for the pendulum, evaluations study the swing-up task at the full pendulum displacement $\theta = 180^{\circ}$. These regions for both systems are heavily nonlinear to exercise any potential numerical issues with the IPA approximation framework.
> > > > >
> > > > > > I condemn the unprofessional tone in the reviewer-author exchanges on this thread.
> > > > >
> > > > > * We greatly appreciate the Reviewer for pointing this out and for promoting a positive, professional review process. We will keep this in mind and make an effort to conduct properly as authors to keep OpenReview a healthy and productive platform for all. Thank you again for the reminder.

---

> > > > > > ### Comment · Reviewer_ZBUg · 2024-12-02
> > > > > >
> > > > > > Thanks for the detailed response. The finiteness of the approximate integral is now clear.
> > > > > >
> > > > > > 1. Can the authors similarly provide an analysis for the finiteness of the original integral
> > > > > >
> > > > > > \begin{align}
> > > > > > & -2 \int_{t_{0}}^{t_{1}}  ( f(x) + g(x) u - \xi(x))^{T} , P_{i} x  \, d\tau + \left[ x^{T}(t_{1}) P_{i} x(t_{1}) - x^{T}(t_{0}) P_{i} x(t_{0}) \right]
> > > > > > \end{align}
> > > > > >
> > > > > > Proving the finiteness of the original integral is also important in my opinion for the approximation to be valid.
> > > > > >
> > > > > > 2. I am not sure if this approximation is accurate:
> > > > > >
> > > > > > \begin{align}
> > > > > > & -2 \int_{t_{0}}^{t_{1}}  ( f(x) + g(x) u - \xi(x))^{T} , P_{i} x  \, d\tau + \left[ x^{T}(t_{1}) P_{i} x(t_{1}) - x^{T}(t_{0}) P_{i} x(t_{0}) \right]  \approx - \int_{t_{0}}^{t_{1}} x^{T} Q x + \mu_{i}^{T}(x) R \mu_{i}(x) \, d\tau \qquad\quad \text{(to \, $o(||x||)$)}.
> > > > > > \end{align}
> > > > > >
> > > > > > The approximation to  $o(||x||)$ makes sense for the integrand, but I am not sure about the integral. The integral approximation seems to be to  $o(||x||(t_1-t_0))$. Maybe I am missing something here, but additional details from the authors would be appreciated. Perhaps the answer to my first question might answer this question as well.

---

> > > > > > > ### Author Response · Authors · 2024-12-02
> > > > > > > **We thank the Reviewer for their continued discussion. Please find our responses below**
> > > > > > >
> > > > > > > > Thanks for the detailed response. The finiteness of the approximate integral is now clear.
> > > > > > >
> > > > > > > * Thank you! We are glad to hold this discussion and have this opportunity to clarify some details.
> > > > > > >
> > > > > > > > 1. Can the authors similarly provide an analysis for the finiteness of the original integral
> > > > > > > >
> > > > > > > > \begin{align}
> > > > > > > > & -2 \int_{t_{0}}^{t_{1}} ( f(x) + g(x) u - \xi(x))^{T} , P_{i} x , d\tau + \left[ x^{T}(t_{1}) P_{i} x(t_{1}) - x^{T}(t_{0}) P_{i} x(t_{0}) \right]
> > > > > > > > \end{align}
> > > > > > > >
> > > > > > > > Proving the finiteness of the original integral is also important in my opinion for the approximation to be valid.
> > > > > > > >
> > > > > > > > 2. I am not sure if this approximation is accurate:
> > > > > > > >
> > > > > > > > \begin{align}
> > > > > > > > & -2 \int_{t_{0}}^{t_{1}} ( f(x) + g(x) u - \xi(x))^{T} , P_{i} x , d\tau + \left[ x^{T}(t_{1}) P_{i} x(t_{1}) - x^{T}(t_{0}) P_{i} x(t_{0}) \right] \approx - \int_{t_{0}}^{t_{1}} x^{T} Q x + \mu_{i}^{T}(x) R \mu_{i}(x) , d\tau \qquad\quad \text{(to , $o(||x||)$)}.
> > > > > > > > \end{align}
> > > > > > > >
> > > > > > > > The approximation to $o(||x||)$ makes sense for the integrand, but I am not sure about the integral. The integral approximation seems to be to $o(||x||(t_1-t_0))$. Maybe I am missing something here, but additional details from the authors would be appreciated. Perhaps the answer to my first question might answer this question as well.
> > > > > > >
> > > > > > > * Thank you for this question. The approximation $o(||x||)$ does hold. To help clarify, we will define $o(||x||)$ here:
> > > > > > >
> > > > > > > * **Definition.** A function $r$ is $o(||x||)$ if the following holds: $\lim_{x \rightarrow 0} \frac{||r(x)||}{||x||} = 0$.
> > > > > > >
> > > > > > > * **To see how the integral is finite:** Consider the full equation here:
> > > > > > >
> > > > > > > \begin{align}
> > > > > > > & -2 \int_{t_{0}}^{t_{1}}  ( f(x) + g(x) u - \xi(x))^{T} \\, P_{i} x  \\, d\tau + \left[ x^{T}(t_{1}) P_{i} x(t_{1}) - x^{T}(t_{0}) P_{i} x(t_{0}) \right]  \approx - \int_{t_{0}}^{t_{1}} x^{T} Q x + \mu_{i}^{T}(x) R \mu_{i}(x) \\, d\tau \qquad\quad \text{(to \\, $o(||x||)$)}.
> > > > > > > \end{align}
> > > > > > >
> > > > > > > * The right-hand-side integral is bounded, from the boundedness proof provided in the previous response.
> > > > > > >
> > > > > > > * The left-hand-side varies from the right-hand-side by only $o(||x||)$, so the left-hand-side is bounded if and only if the right-hand-side is bounded. Thus, both integrals are bounded.

---

> ### Comment · Reviewer_ZBUg · 2024-12-02
>
> Thanks for the response. I am not sure about the correctness of the mathematical argument. It appears correct only on a bounded interval for the integral (finite-time horizon). Boundedness of the input to an integral does not imply the boundedness of the integral, if the interval is unbounded. For that matter, even small, bounded, and converging functions can have unbounded integrals. To be more precise in the discussion about the approximation, let $r(x)$ denote the residual error in approximation resulting from higher-order terms such that
>
> \begin{align}
> & -2 \int_{t_{0}}^{t_{1}}  (( f(x) + g(x) u - \xi(x))^{T} \ P_{i} x  + r(x)) \ d\tau + \left[ x^{T}(t_{1}) P_{i} x(t_{1}) - x^{T}(t_{0}) P_{i} x(t_{0}) \right] = - \int_{t_{0}}^{t_{1}} x^{T} Q x + \mu_{i}^{T}(x) R \mu_{i}(x)  d\tau \qquad\quad.
> \end{align}
>
> I can see why $r(x)$ term is $o(||x||)$, but the integral would also introduce dependencies on $t_1-t_0$. As a counterexample, consider the case where $r=\frac{1}{1+t}$, which is bounded and converging to 0 as $t\to\infty$. In this case, $\int_{t\_{0}}^{t\_{1}}  r d\tau = ln(1+t_1)- ln(1+t_0)$, which grows unbounded as $t_1 \to \infty$, despite the boundedness and covnergence of $r$. Note that nonlinear systems such as $\dot{x}=-x^2$ do have solutions of the reciprocal form $\frac{a}{b+ct}$, so the scenario I am describing is realistic. This is an example of how even small converging perturbations can accumulate and grow large within an integral. I was expecting a more rigorous analysis for showing boundedness of the original integral with explicit error bounds and perturbation analysis, similar to how authors analyzed the approximated integral.

---

> > ### Comment · Reviewer_ZBUg · 2024-12-02
> >
> > Note ADP papers use a Lyapunov-based stability analysis argument to show boundedness of the value function for the nonlinear system. They do so by incorporating the value function into the Lyapunov function. Maybe a similar approach could be useful to show boundedness of this integral using a value function perspective.

---

> > > ### Author Response · Authors · 2024-12-03
> > > **We thank the Reviewer for their continued discussion. We provide the requested Lyapunov boundedness proof and clarify some points below (1 of 2)**
> > >
> > > > Thanks for the response. I am not sure about the correctness of the mathematical argument. It appears correct only on a bounded interval for the integral (finite-time horizon). Boundedness of the input to an integral does not imply the boundedness of the integral, if the interval is unbounded. For that matter, even small, bounded, and converging functions can have unbounded integrals. To be more precise in the discussion about the approximation, let $r(x)$ denote the residual error in approximation resulting from higher-order terms such that
> > > >
> > > > \begin{align}
> > > > & -2 \int_{t_{0}}^{t_{1}} (( f(x) + g(x) u - \xi(x))^{T} \ P_{i} x + r(x)) \ d\tau + \left[ x^{T}(t_{1}) P_{i} x(t_{1}) - x^{T}(t_{0}) P_{i} x(t_{0}) \right] = - \int_{t_{0}}^{t_{1}} x^{T} Q x + \mu_{i}^{T}(x) R \mu_{i}(x) , d\tau \qquad\quad \text{(to , $o(||x||)$)}.
> > > >\end{align}
> > > >
> > > > I can see why $r(x)$ term is $o(||x||)$, but the integral would also introduce dependencies on $t_1-t_0$. As a counterexample, consider the case where $r=\frac{1}{1+t}$, which is bounded and converging to 0 as $t\to\infty$. In this case, $\int_{t_{0}}^{t_{1}} r d\tau = ln(1+t_1)- ln(1+t_0)$, which grows unbounded as $t_1 \to \infty$, despite the boundedness and covnergence of $r$. Note that nonlinear systems such as $\dot{x}=-x^2$ do have solutions of the reciprocal form $\frac{a}{b+ct}$, so the scenario I am describing is realistic. This is an example of how even small converging perturbations can accumulate and grow large within an integral. I was expecting a more rigorous analysis for showing boundedness of the original integral with explicit error bounds and perturbation analysis, similar to how authors analyzed the approximated integral.
> > > >
> > > > Note ADP papers use a Lyapunov-based stability analysis argument to show boundedness of the value function for the nonlinear system. They do so by incorporating the value function into the Lyapunov function. Maybe a similar approach could be useful to show boundedness of this integral using a value function perspective.
> > >
> > > * We thank the Reviewer for this response and for following up with more specific information that has inspired us to respond with more details than the previous response. We made an effort to address the issue now by providing a Lyapunov analysis and by sharing our thoughts in response to the questions. Please refer to the details below.
> > >
> > > **1. On Your Counterexample.** We thank you for introducing this counterexample, as it's an insightful addition to highlight the key insights of the broader points underlying the discussion. In general, this counterexample is correct. However, in the present context it does not satisfy the assumption of IPA. We provide a discussion of this below, as well as a Lyapunov proof of boundedness available when IPA hypotheses are met which rule out counterexamples such as this one.
> > >
> > > * First, we noted that the crux of the diverging integral as the horizon $t_{1} \rightarrow \infty$ in this example is, as the Reviewer has noted, that this is characteristic of a closed-loop system governed by an ODE of form $\dot{x} = - x^{2}$.
> > > * However, this counterexample does not satisfy the assumption of IPA (cf. Theorem 3.1), which requires that the initial stabilizing policy $\mu_{0}$ satisfy that $\left. \frac{\partial}{\partial x}  \\{  f(x) + g(x) \mu_{0}(x) \\} \right|\_{x = 0} \triangleq A - B K_{0}$ is Hurwitz (i.e., all closed-loop eigenvalues have strictly negative real part). This example fails the IPA hypothesis: $\left. \frac{\partial}{\partial x}  \\{  f(x) + g(x) \mu_{0}(x) \\} \right|\_{x = 0} = \left. \frac{\partial}{\partial x}  \\{  - x^{2} \\} \right|\_{x = 0} = 0$.
> > > * We include this assumption for this reason (among others, which include for well-posedness of the underlying Kleinman algorithm solution). Indeed, the Hurwitz condition is integral to the IPA boundedness proof below.

---

> ### Author Response · Authors · 2024-12-03
> **Response to Reviewer (2 of 2)**
>
> **2. A Lyapunov Proof of Boundedness of the Integral.** Let $r(x)$ be the approximation error of the present discussion as defined by the Reviewer.
>
> **Claim.** Suppose that $x(t_{0})$ is in the basin of attraction of the initial stabilizing policy $\mu_{0}$ (i.e., of the autonomous system $\dot{x} = f(x) + g(x) \mu_{0}(x)$), where $\mu_{0}$ satisfies the IPA hypothesis: $\left. \frac{\partial}{\partial x}  \\{  f(x) + g(x) \mu_{0}(x) \\} \right|\_{x = 0} \triangleq A - B K_{0}$ is Hurwitz. Then $\int_{t_{0}}^{\infty} r(x(\tau)) \\, d\tau < \infty$.
>
> **Proof.** Follows from the following Lemma:
>
> **Lemma.**  Consider an autonomous system of the form $\dot{x} = h(x)$ with $h(0) = 0$, let $H \triangleq \left. \frac{\partial h}{\partial x} \right|\_{x = 0}$, and assume $H$ is Hurwitz. Suppose that $x(t_{0})$ is in the basin of attraction of the autonomous system $\dot{x} = h(x)$. Then $\int_{t_{0}}^{\infty} ||x||^{2} \\, d\tau < \infty$.
>
> **Proof of Lemma.** Since $H$ is Hurwitz, given any $Q = Q^{T} > 0$, the solution $P \in \mathbb{R}^{n}$ of the ALE $P H + H^{T} P = Q$ is symmetric positive definite; i.e.,  $P = P^{T} > 0$. Consider Lyapunov function candidate $V(x) = x^{T} P x$. Note:
>
> \begin{align}
> 	\dot{V}(x) &= x^{T} P h(x) + h^{T}(x) P x
> 	\\\\
> 	&= x^{T} P [H x + s(x)] + [H x + s(x)]^{T} P x
> 	\\\\
> 	&= x^{T} [P H + H^{T} P] x + 2 x^{T} P s(x)
> 	\\\\
> 	&= - x^{T} Q x + 2 x^{T} P s(x)
> \end{align}
>
> where $s(x) \triangleq h(x) - H x$ is $o(||x||)$: $\lim_{x \rightarrow 0} \frac{||s(x)||}{||x||} = 0$. Thus, let $r > 0$ be such that $||s(x)|| < \frac{\sigma_{min}(Q)}{2 ||P||} ||x||$ for all $||x|| < r$. Then, whenever $||x|| < r$, we have
>
> \begin{align}
> 	\dot{V}(x) &= - x^{T} Q x + 2 x^{T} P s(x)
> 	\\\\
> 	&\leq
> 	- x^{T} Q x + 2 ||P|| ||x|| ||s(x)||
> 	\\\\
> 	&<
> 	- \sigma_{min}(Q) ||x||^{2} + 2 ||P|| ||x|| \frac{\sigma_{min}(Q)}{2 ||P||} ||x||
> 	\\\\
> 	&=
> 	- \frac{\sigma_{min}(Q)}{2} ||x||^{2}
> \end{align}
>
> Thus, indeed $V$ is a Lyapunov function for the system, with invariant set $\\{||x|| < r\\}$.
>
> Let $t^{\prime} \geq t_{0}$ be any time at which $||x|| < r$ (such a $t^{\prime}$ exists, since $x(t_{0})$ is in the attraction basin by hypothesis). Let now $t_{1} > t^{\prime}$ be given. The following inequality holds:
>
> \begin{align}
> 	0 \leq \frac{\sigma_{min}(Q)}{2} \int_{t^{\prime}}^{t_{1}} ||x||^{2} d\tau
> 	\leq \int_{t^{\prime}}^{t_{1}} - \dot{V} d\tau
> 	\= V(x(t^{\prime})) - V(x(t_{1}))
> 	\leq
> 	V(x(t^{\prime}))
> \end{align}
>
> Thus,
>
> \begin{align}
> 	\int_{t_{0}}^{t_{1}} ||x||^{2} d\tau
> 	\=
> 	\int_{t_{0}}^{t^{\prime}} ||x||^{2} d\tau + \int_{t^{\prime}}^{t_{1}} ||x||^{2} d\tau
> 	\leq
> 	\int_{t_{0}}^{t^{\prime}} ||x||^{2} d\tau + \frac{2}{\sigma_{min}(Q)} V(x(t^{\prime}))
> \end{align}
>
> Note that the right-hand-side of the above inequality is independent of the selection of $t_{1} > t^{\prime}$. Thus, we may send $t_{1} \rightarrow \infty$ in the above, completing the proof: $\int_{t_{0}}^{\infty} ||x||^{2} d\tau \leq \int_{t_{0}}^{t^{\prime}} ||x||^{2} d\tau + \frac{2}{\sigma_{min}(Q)} V(x(t^{\prime})) < \infty$.
>
> **3. More Broadly: From an Analysis/Design Perspective.**
>
> * The preceding discussion is a very important one to explore key insights and understand the theoretical properties of IPA, and we are glad to have held it with the Reviewer.
>
> * However, the preceding discussion notwithstanding, we would like to note that from a design perspective (even with an analytical guarantee in the above), choosing $t_{1} \rightarrow \infty$ arbitrarily large is not the intended use for this IPA application.
>
> * Indeed, one wishes to select the sample period $t_{1} - t_{0}$ at an appropriate rate relative to the natural bandwidth of the dynamics under control (albeit sampling at Nyquist/Shannon frequency), while meeting IPA's full column rank assumption required for the guarantees of Theorem 3.1.
>
> * For example, a sample period $t_{1} - t_{0}$ of 6 sec was used on the HSV design, and 1 sec and 0.1 sec for the pendulum and second-order environments, respectively.

---

> > ### Comment · Reviewer_ZBUg · 2024-12-03
> >
> > Thanks for the meticulous response. The response was satisfactory. Considering this additional discussion would be included in the final paper upload, the paper is theoretically rigorous by my gauge.

---

> ### Author Response · Authors · 2024-12-03
> **Our thanks to this Reviewer, we are glad they deem the work theoretically rigorous and were happy to meticulously address their questions!**
>
> It's been our pleasure to have this insightful discussion, and we have greatly appreciated the Reviewer's time and feedback. We will ensure these points are incorporated into the final release. Thank you!

---

### Official Review · Reviewer_5RXP · 2024-10-31

**Soundness:** 3
**Presentation:** 3
**Contribution:** 2
**Rating:** 8
**Confidence:** 4

**Summary:**

This paper introduces Integral Performance Approximation (IPA), a new method for continuous-time reinforcement learning (CT-RL) control. IPA utilizes an affine nonlinear dynamic model that partially captures the environment's dynamics, alongside state-action trajectory data, to enable highly data-efficient and robust control. The approach incorporates structures from the Kleinman algorithm to ensure theoretical guarantees for learning convergence, solution optimality, and closed-loop stability. The effectiveness of IPA is demonstrated across three CT-RL environments, including hypersonic vehicle (HSV) control, which presents additional challenges due to unstable and non-minimum phase dynamics.

**Strengths:**

The paper introduces a CT-RL method, IPA CT-RL, which leverages an affine nonlinear dynamic model and quadratic cost structure for data-efficient, robust control. It provides theoretical guarantees for convergence, optimality, and stability, validated through extensive evaluations. Finally, it demonstrates IPA CT-RL's successful application to HSV control.

**Weaknesses:**

However, there are still some aspects of this paper that require clarification:

1. The paper mentions several SOTA methods that are not restricted to control-affine systems, such as Yildiz et al. (2021). However, the authors did not include these methods in their simulations for comparison. Could the authors explain the reasoning behind this choice?

2. The authors briefly mention the discretization of continuous-time environments in Yildiz et al. (2021), suggesting that this process could lead to significant numerical issues for real-world systems. However, their proposed method also relies on discrete data points when performing integration, which could introduce discretization errors. A previous study [1] has analyzed this issue in detail. I believe the authors should at least discuss the impact of discretization error in their approach.

3. The authors suggest that HSV represents a SOTA environment, but it still appears to be relatively low-dimensional. It’s unclear why their approach cannot scale to higher dimensions, especially since the method in Yildiz et al. (2021) seems capable of handling more complex systems. The authors should discuss the limitations that might prevent their approach from scaling up.

4. I am unclear about the theoretical advantage of the IPA method. Is this benefit primarily due to a linearization structure or another feature? I recommend that the authors provide a clear explanation of this through a example. Additionally, it’s unclear if IPA applies effectively to all control-affine systems or only to those with high linearity. The systems simulated by the authors do not exhibit a high degree of nonlinearity, so clarification here would be helpful.

If the authors can address these four points clearly in the updated version of the paper, I would be inclined to raise my score.

[1] Cao W, Pan W. Impact of Computation in Integral Reinforcement Learning for Continuous-Time Control[C]. The Twelfth International Conference on Learning Representations.

**Questions:**

All the questions are listed in the weakness part.

---

> ### Author Response · Authors · 2024-11-19
> **Response to Reviewer (Part 1)**
>
> * We thank this reviewer for all of their time and insightful feedback. Please see our Global Response above, which gives a comprehensive summary of this work and review discussion.
>
> ## Weaknesses:
>
> > **1.** The paper mentions several SOTA methods that are not restricted to control-affine systems, such as Yildiz et al. (2021). However, the authors did not include these methods in their simulations for comparison. Could the authors explain the reasoning behind this choice?
>
> * Thank you for this point. First, please refer to our self-devoted Limitations section on the issue of general nonlinear dynamics.
>
> * As discussed in Section 1, fully nonlinear algorithms are at a very early stage. Comprehensive theoretical results and meaningful designs without stringent assumptions are still under development. Some of them may have great potential, yet few methods have synthesized meaningful controllers with guarantees of convergence, optimality, stability, and robustness. Additional examples such as CT-VI (Bian \& Jiang, 2022) are methods that were proposed to address general nonlinear dynamics. But as shown in (Wallace \& Si, 2024), these methods fail to synthesize controls for simple 2nd order systems with known closed-form solutions. A small change in the basis of such small problem led to learning failure.
>
> * We did not include methods addressing general nonlinearity such as that in (Yildiz et al. 2021) in our simulations for comparison for several reasons.
>   1. On the outset and fundamentally, our IPA method is different from those such as (Yildiz et al. 2021).
>   2. Qualitatively, learning/performing system identification of a general nonlinear dynamic model can be a more complex process than that of an affine nonlinear model, especially since many dynamical systems readily render themselves of the affine nonlinear form. In some cases the problem can even become a model parameter estimation one.
>   3. Carrying on from the previous point, how to use the general nonlinear dynamic model and affine dynamic model also makes a difference among different approaches.
>     * In our case, the affine nonlinear model provides a solution construct for us to derive fully nonlinear control policies. Unmodeled dynamics can be considered and accommodated for by learning adaptation using data directly from the environments. Or in another word, we are able to effectively address the unmodeled dynamics, some of which may come from general nonlinearity even though our development requires affine nonlinearity.
> 	* But the same may not be said for those solution approaches relying on a learned nonlinear dynamic model from sampled data. The (Yildiz et al. 2021) approach learns arbitrary time differentials of the governing real-world dynamics, and then forward simulates the surrogate ODE dynamics to learn optimal policy functions using actor-critic approach. As such, the estimated state differential is used in forward-simulated ODE trajectories aiming to reproduce the true ODE trajectories. The unmodeled dynamics or modeling errors in this case may adversely affect the quality of the policy.
>   4. Furthermore, (Yildiz et al. 2021) considers a discounted cost only, while our work is undiscounted. Direct comparison of the two works via the cost performance as we have done in our evaluations thus is not straightforward. We would have to modify the existing cost structure studied by 1) our work, and 2) the tested FVI methods (Lutter et al., 2021, 2023b) with a new discounted variant for (Yildiz et al. 2021), and even then the numerical comparability of the results loses its rigor. In a similar vein, the other general nonlinear work (Sandoval et al., 2023) considers finite-horizon cost only, an entirely different class than the infinite-horizon problem studied.
>
> * Given the above considerations, within the scope of the present work it would be comparing apples to oranges to include (Yildiz et al. 2021) in the studies presented.

---

> > ### Author Response · Authors · 2024-11-19
> > **Response to Reviewer (Part 2)**
> >
> > > **2.** The authors briefly mention the discretization of continuous-time environments in Yildiz et al. (2021), suggesting that this process could lead to significant numerical issues for real-world systems. However, their proposed method also relies on discrete data points when performing integration, which could introduce discretization errors. A previous study [1] has analyzed this issue in detail. I believe the authors should at least discuss the impact of discretization error in their approach.
> > >
> > > [1] Cao W, Pan W. Impact of Computation in Integral Reinforcement Learning for Continuous-Time Control[C]. The Twelfth International Conference on Learning Representations.
> >
> > * Thank you for bringing [1] to our attention, we have included a discussion of this reference in the Introduction (see pp. 1, ln. 52-57). Reference [1] also discusses the aforementioned studies of (Wallace \& Si, 2024). Indeed, the review and formulation in [1] draws heavily from (Wallace \& Si, 2024), and [1] devotes entire Appendices K and L to discussing the results of this work.
> >
> > * We agree that [1] provides nice convergence rate analysis of and with a focus on the seminal Integral Reinforcement Learning (IRL) method (Vrabie \& Lewis, 2009) given upper bounds on IRL's computation error. However, the proposed IPA method is an entirely different algorithm than (Vrabie \& Lewis, 2009), with significant differences in formulation, approximation, and policy structure. Adapting the results developed in [1] for (Vrabie \& Lewis, 2009) to the present work is not likely straightforward.
> >
> > * Indeed, our method requires discrete samples to satisfy the condition of full column rank $\underline{n} \triangleq \frac{n (n+1)}{2}$ of the IPA matrix $\mathcal{I} \in \mathbb{R}^{l \times \underline{n}}$ (20) -- once this rank condition is met, Theorem 3.1 guarantees convergence, optimality, and closed-loop stability. This assumption is easy to satisfy, as the algorithm can continue to collect $l > \underline{n}$ samples until it is met. It is also virtually instantaneous to verify (matrix rank calculation). We observe no issue satisfying this assumption even on the complex HSV in practice.
> >
> > * On discretization error: We would like to emphasize the difference between
> >
> >   **1.** Discretizing the value function and/or dynamical equations (as in Yildiz and comparable works).
> >     * As [1] notes, even if a discretization can be formed, this casts the problem into a discrete-time optimal control problem -- a completely different class. The CT HJB equation (a first-order nonlinear PDE) is not equivalent to the DT Bellman equation (a difference equation).
> > 	* Discretization is a nontrivial design choice, and the $s \mapsto z$ domain mapping introduces inevitable dynamics discretization error (different error trades for different discretization methods).
> > 	* This may work on relatively simple systems, but generalizing it to complex continuous-time phenomena is exceptionally difficult. Take, for example, hypersonic gas dynamics -- there is no clear way to discretize aeroelastic/flowfield phenomena. Time-discretizing temperature and pressure distributions across, e.g., the forward face of a scramjet cowl door represents a challenge in all likelihood more daunting than the original optimal control problem under study.
> >
> >   **2.** Collecting data from the fully continuous-time system at discrete time instants (as in IPA).
> >     * As [1] discusses, This does not introduce discretization error, which is a significant advantage of CT-RL methods directly addressing the CT dynamics.
> > 	* Also in [1]: It also avoids the need to cast the CT optimal control problem into a different class of DT control.
> > 	* Furthermore, as long as the bandwidth of the system dynamics are sampled at the Nyquist-Shannon rate or greater, the frequency content features of the trajectory data can be successfully captured for learning.

---

> > > ### Author Response · Authors · 2024-11-19
> > > **Response to Reviewer (Part 3)**
> > >
> > > > **3.** The authors suggest that HSV represents a SOTA environment, but it still appears to be relatively low-dimensional. It’s unclear why their approach cannot scale to higher dimensions, especially since the method in Yildiz et al. (2021) seems capable of handling more complex systems. The authors should discuss the limitations that might prevent their approach from scaling up.
> > >
> > > * Thank you for this point, we have included discussion in the revision (see pp. 2, ln. 58-64) to reflect the following points. In short, IPA is expected to handle more complex and high-order dynamics, as:
> > >
> > >   * Given a system of order $n$, the dimension of IPA's quadratic critic network is $\underline{n} = \frac{n(n+1)}{2}$, or quadratic in the system order. Due to IPA's quadratic bases, its learning regression reduces to a simple linear regression problem of dimension $\underline{n}$, the numerical solution of which has been extensively optimized and requires little computing resources compared to computation required in training modern deep neural networks.
> > >   * To put this in perspective and be more specific, consider the following. The deep networks used for FVIs on the second-order pendulum $n = 2$ (identical network dimensions chosen as the original pendulum evaluations in (Lutter et al. 2021,2023b)) use on the order of 80,000 network weights. It would take a system of order $n = 400$ for IPA to have a critic network dimension $\underline{n} = 80,000$. This is favorable scaling to the SOTA methods.
> > >
> > > * **The HSV model is 5th order with 2 inputs. Even though it may not appear that high order, the HSV is higher-dimensional than all of the environments studied in (Yildiz et al., 2021) (4th order, single-input), and the environments in the ADP studies (see Table 4 below).**
> > >
> > > * Specifically, (Yildiz et al., 2021) studies: 1) Pendulum: 2nd order, 2) CartPole: 4th order, and 3) Acrobot (2-link pendulum): 4th order. These pendulum-variant systems are lower-dimensional than the HSV in both their state and input spaces.
> > >
> > > * Furthermore, The HSV dynamical model is based on NASA Langley aeropropulsicve data (Shaughnessy et al, 1990) and is **significantly more complex than the pendulum-variant systems.** To illustrate the dynamics challenge involved for the HSV, consider the following:
> > >
> > > * Consider balancing a yardstick on your finger, the HSV’s pitch-up instability places hard lower bounds on how “slow” the HSV can be controlled – too slow, and the yardstick topples over. This instability is uniquely combined with nonminimum phase behavior, another profound dynamical challenge which, simply stated, results in the aircraft jerking downward before initiating an upward climb maneuver. This is due to the great parasitic coupling occurring when upward deflections of the tail (which are commanded to pitch the nose of the vehicle upward to initiate a climb) cause a downward force on the vehicle which thereby results in a temporary dip in altitude. Fundamentally, nonminimum phase behavior places hard upper bounds on how “fast” the HSV can be controlled (Bolender and Doman, 2006a). The inherent dynamics of HSVs aside, the extreme speeds involved cause immense difficulty in aerodynamic modeling due to complex aeropropulsive interactions and body flexing under the stresses involved in flight (Wang \& Stengel, 2000), (Marrison \& Stengel, 1998) (Bolender \& Doman, 2005, 2006a-b).
> > >
> > > * For a direct comparison of how the present work's environments compare to the evaluations conducted by the SOTA works in ADP CT-RL and deep RL FVIs, please see Table 4 of Appendix A, reproduced here. In short, our evaluations sit comfortably in the CT-RL state of the art.The FVIs study pendulum/cart-pendulum variants of order $n \leq 4$ only. The ADP methods tend to study non-physical academic examples, most second-order with optimal solutions known *a priori*.
> > >
> > > ### Table 4: Environments in SOTA CT-RL evaluations (from Appendix A)
> > >
> > > Algorithm|System|Order|#Inputs|Source of Model Parameters
> > > |-|-|-|-|-|
> > > **IPA**|SOS|$\longrightarrow$|$\longrightarrow$|Identical to SPI as benchmark
> > > ``|Pendulum|$\longrightarrow$|$\longrightarrow$|Identical to FVIs as benchmark
> > > ``|HSV|5|2|NASA Langley aeropropulsive data (Shaughnessy et al., 1990). Unstable, nonminimum phase, complex
> > > **FVIs**|Pendulum|2|1|Quanser STEM curriculum resources (Quanser, 2018)
> > > ``|Cart Pendulum |4|1|Quanser STEM curriculum resources (Quanser, 2018)
> > > ``|Furatura Pendulum|4|1|Quanser STEM curriculum resources (Quanser, 2018)
> > > **IRL**|Simple Academic|2|1|Non-physical, optimal known *a priori*
> > > ``|Simple Academic|2|1|Non-physical, optimal known *a priori*
> > > **SPI**|Simple Linear|3|1|Non-physical LQR example
> > > ``|Simple Academic|2|1|Non-physical, optimal known *a priori*
> > > **RADP**|Simplified Engine|2|1|Non-physical for illustration
> > > ``|Simplified Power Bus|2|1|Non-physical for illustration
> > > **CT-VI**|Simple Academic|2|1|Non-physical, optimal known *a priori*
> > > ``|Simplified Robot Arm|4|2|Non-physical for illustration

---

> > > > ### Author Response · Authors · 2024-11-19
> > > > **Response to Reviewer (Part 4)**
> > > >
> > > > > **4.** I am unclear about the theoretical advantage of the IPA method. Is this benefit primarily due to a linearization structure or another feature? I recommend that the authors provide a clear explanation of this through a example. Additionally, it’s unclear if IPA applies effectively to all control-affine systems or only to those with high linearity. The systems simulated by the authors do not exhibit a high degree of nonlinearity, so clarification here would be helpful.
> > > >
> > > > * Thank you for these important questions that remind us to further clarify and highlight the contributions of IPA. Please see our prior discussion of IPA being a fully nonlinear controller. We have included these points of clarification in the revision, and we appreciate the reviewer helping us clarify.
> > > >
> > > > * In short, IPA's novel integral performance approximation scheme allows Kleinman control structures to be combined with state-action data $(x,u)$ from the actual physical environment in learning. As a result, IPA simultaneously offers:
> > > >   1. Significant theoretical guarantees with much less stringent assumptions than the ADP CT-RL methods,
> > > >   2. Significant data efficiency improvements (cf. Section 5). Indeed, IPA's approximation scheme yields a **6 order of magnitude increase in data efficiency** relative to the SOTA FVIs (cf. Table 1), and
> > > >   3. Demonstrated well-behaved closed-loop performance matching/exceeding the SOTA deep RL FVI methods (cf. Section 5).
> > > >
> > > > * We empirically demonstrate with rigor by compelling evaluation results that IPA exhibits learning/control performance meeting/exceeding the leading ADPs and SOTA FVIs on 3 nonlinear environments, including **the realistic HSV, which is a complex environment represented in affine nonlinear form, the hypersonic gas dynamics alone are highly nonlinear**, which is one of many challenging aeropropulsive phenomena captured in this NASA Langley model (Shaughnessy et al, 1990). Please see details in the above discussion of the HSV dynamics, they are highly complex.
> > > >
> > > > * In addition to the above, IPA inherits local stability robustness properties owing to its structural parallels to Kleinman's algorithm. We thank the reviewer for bringing this point to the conversation, and as a result we have added the following verbage to Theorem 3.1:
> > > >
> > > > > ... As a result, IPA inherits the guaranteed local stability robustness margins of Kleinman's algorithm:
> > > > >* $||S_{u}||\_{\mathcal{H}^\infty} \leq 0$ dB
> > > > >* $||T_{u}||\_{\mathcal{H}^\infty} \leq 2$ dB
> > > > >
> > > > > where $||\cdot||\_{\mathcal{H}^\infty}$ denotes the $\mathcal{H}^\infty$ norm, and where $S_{u}$, $T_{u}$ denote the sensitivity and complementary sensitivity closed-loop maps, respectively, at the control loop breaking point $u$ (Rodriguez, 2004). We have also added a definition of terms and discussion of these results to Appendix F of the revision.
> > > >
> > > > > If the authors can address these four points clearly in the updated version of the paper, I would be inclined to raise my score.
> > > >
> > > > * We thank this reviewer for their insightful questions, encouraging review, and stimulating conversation! Please let us know if you have additional questions.
> > > >
> > > > ## Questions:
> > > >
> > > > > All the questions are listed in the weakness part.

---

> > > > > ### Comment · Reviewer_5RXP · 2024-11-19
> > > > >
> > > > > I would like to express my sincere appreciation for the thoughtful responses provided by the authors. The authors have addressed all of my concerns in great detail, particularly regarding the difficulty of the HSV task. I believe that this is a high-quality paper. As a result, I have decided to raise my score to 8.

---

> > > > > > ### Author Response · Authors · 2024-11-19
> > > > > > **We thank this Reviewer for their insights and positive review!**
> > > > > >
> > > > > > It has been our pleasure to have this insightful discussion. We thank this reviewer greatly for their insights and their positive review in recognizing the significance of IPA.

---

### Official Review · Reviewer_ZBUg · 2024-11-02

**Soundness:** 3
**Presentation:** 3
**Contribution:** 3
**Rating:** 8
**Confidence:** 4

**Summary:**

This paper introduces integral performance approximation (IPA) for continuous-time model-based reinforcement learning control of control-affine nonlinear systems.

**Strengths:**

The paper is well-written, the contributions are clear and of significant importance for solving continuous-time optimal control problems. The simulation and ablation studies are extensive and make a compelling case for the developed controller.

**Weaknesses:**

My main concerns are regarding the general mathematical rigor in the theoretical results. Specifically, there are a lot of linearized approximations throughout the paper, which by itself is not necessarily problematic. However, then one would expect the unknown approximation error to be accounted for in the analysis with some appropriate bounds. Instead, there are a lot of $\approx$ relations in the paper (e.g., (14), (15), (30) etc.) without accounting for the errors.  Specifically, in Eq. (13), K_i is defined as $$-\frac{\partial}{\partial x}\mu_i (x)|_{x=0},$$ and then it is said
$$\mu_i \approx -K_i x.$$
This linearization is foundational to the entire paper. At that point one can also linearize the dynamic model itself, i.e.,
$$f(x)+g(x)u \approx Ax+Bu$$ and then apply the entire development accordingly. This makes me wonder if the development is appropriate for the nonlinear system in Eq. (1) like it is claimed.
Instead of writing $\mu_i \approx -K_i x,$ it would be more appropriate to write $\mu_i = -K_i x + O(||x||^2)$. The linearization error would then grow quadratically with the states, but for the region $||x||\leq r$, there exists some constant $c_1$ such that can be bounded by $ O(||x||^2) \leq c_1 r^2$. Similar analysis needs to be performed for all of the other approximations in the paper to achieve a local stability result with robustness to small perturbations. The way the paper stands now, the analysis is valid only in an infinitesimal neighborhood of the origin.

Besides, the literature review surrounding CT-RL and ADP methods is sparse. In Appendix M, the following vague comment is made about ADP methods:

 "As a result of ADP’s theoretical frameworks in adaptive and optimal control, Lyapunov arguments are available to prove qualitative properties including weight convergence and closed-loop stability results. However, the results require restrictive theoretical assumptions which are difficult to satisfy for even simple academic examples, and as a result these methods exhibit empirical issues."

However, it is not specified what theoretical assumptions are restrictive and difficult to satisfy. Besides the literature and ablation study on ADP is mostly focused on the old result by Vamvoudakis and Lewis (2010). Besides, a lot of further development has happened after this classical work. For example, see the following works and references citing them/therein:

Kamalapurkar, R., Walters, P. and Dixon, W.E., 2016. Model-based reinforcement learning for approximate optimal regulation. Automatica, 64, pp.94-104.

Vamvoudakis, K.G., 2017. Q-learning for continuous-time linear systems: A model-free infinite horizon optimal control approach. Systems & Control Letters, 100, pp.14-20.

In Appendix F, the authors mention "Many SOTA ADP CT-RL algorithms require the persistence of excitation (PE) condition in proofs
of algorithm properties". There are many newer results which relax the PE condition with initial/interval/finite-time excitation conditions. See the following references for example:

Jha, S.K., Roy, S.B. and Bhasin, S., 2019. Initial excitation-based iterative algorithm for approximate optimal control of completely unknown LTI systems. IEEE Transactions on Automatic Control, 64(12), pp.5230-5237.

Yang, Y., Pan, Y., Xu, C.Z. and Wunsch, D.C., 2022. Hamiltonian-driven adaptive dynamic programming with efficient experience replay. IEEE Transactions on Neural Networks and Learning Systems.

The authors can discuss these newer results that relax the PE condition and analyze how they relate to or could potentially improve the proposed IPA method.

EDIT: Increased the score to 8 after the revisions.

**Questions:**

1. Does the state-dependent part of the cost need to be quadratic? Specifically, can the $x^T Q x$ term be generalized to some positive semi-definite function Q(x). I can understand the challenges generalizing so for the $u^T R u$, but $x^T Q x$  must be easy if linearizations are being used in the integral approximation anyway.

2. Does the linearization in (13) need to be necessarily around x=0? What would be difficult about constructing a function $K_i(x)=-\frac{\partial}{\partial x}\mu_i (x)$? across a range of values of x, similar to gain scheduling methods?

---

> ### Author Response · Authors · 2024-11-19
> **Response to Reviewer (Part 1)**
>
> * We thank this reviewer for all of their time and insightful feedback. Please see our Global Response above, which gives a comprehensive summary of this work and review discussion.
>
> ## Strengths:
>
> > The paper is well-written, the contributions are clear and of significant importance for solving continuous-time optimal control problems. The simulation and ablation studies are extensive and make a compelling case for the developed controller.
>
> * We thank this Reviewer for their positive feedback! We have taken great care to ensure that your comments have been clearly addressed and implemented in the revised manuscript. Thank you for this valuable feedback.
>
> ## Weaknesses:
>
> > My main concerns are regarding the general mathematical rigor in the theoretical results. Specifically, there are a lot of linearized approximations throughout the paper, which by itself is not necessarily problematic. However, then one would expect the unknown approximation error to be accounted for in the analysis with some appropriate bounds. Instead, there are a lot of $\approx$ relations in the paper (e.g., (14), (15), (30) etc.) without accounting for the errors ...  Instead of writing $\mu_{i} \approx - K_{i} x$ it would be more appropriate to write $\mu_{i} = - K_{i} x + \mathcal{O}(||x||^{2})$.
>
> * **On your concerns if the IPA development is appropriate for addressing the nonlinear system in (1):** We would like to emphasize that the IPA policies $\mu_{i}$ (18):
> \begin{align}
> \textstyle
> \mu_{i+1}(x) = - \frac{1}{2} R^{-1} g^{T}(x) \frac{\partial V}{\partial x}(x)
> \hspace{1.5in} (18)
> \end{align}
> are **fully nonlinear policies.**
>
> * The linear matrices $K_{i}$ (13):
> \begin{align}
> \textstyle K_{i}  \triangleq - \left. \frac{\partial}{\partial x} \\{ \mu_{i}(x) \\} \right|_{x = 0}
> \hspace{1.5in} (13)
> \end{align}
> are used for approximating the integral performance via (14) only:
>
> \begin{align}
> \int_{t_{0}}^{t_{1}} x^{T} Q x + \mu_{i}^{T}(x) R \mu_{i}(x) \\, d\tau \approx \int_{t_{0}}^{t_{1}} x^{T} \big( Q + K_{i}^{T} R K_{i} \big) x \\, d\tau
> \hspace{1.5in} (14)
> \end{align}
> **they are not the IPA policies.**
>
> * This novel integral performance approximation (IPA) scheme allows us to achieve demonstrated well-behaved system responses from the nonlinear policies $\mu_{i}$ (18) as well as high data efficiency. Our IPA approach directly addresses nonlinear Bellman optimality (3) by learning with state-action trajectory data $(x, u)$ generated by the **actual nonlinear environment.**
>
> * On your concerns of the general mathematical rigor in the theoretical results: The convergence, optimality, and closed-loop stability results of Theorem 3.1 apply to the **nonlinear system (1) -- we achieve these nonlinear results via our nonlinear policies $\mu_{i}$ (18), not the approximate linear policies $K_{i}$ (13).**
>
> * In addition, we have added new local stability robustness results to the revision. IPA inherits these properties owing to its structural parallels to Kleinman's algorithm. We have added the following result to Theorem 3.1:
>
> > ... As a result, IPA inherits the guaranteed local stability robustness margins of Kleinman's algorithm:
> >* $||S_{u}||\_{\mathcal{H}^\infty} \leq 0$ dB
> >* $||T_{u}||\_{\mathcal{H}^\infty} \leq 2$ dB
> >
> > where $||\cdot||\_{\mathcal{H}^\infty}$ denotes the $\mathcal{H}^\infty$ norm, and where $S_{u}$, $T_{u}$ denote the sensitivity and complementary sensitivity closed-loop maps, respectively, at the control loop breaking point $u$ (Rodriguez, 2004). We have also added a definition of terms and discussion of these results to Appendix F of the revision.
>
> * This robustness result is quite substantial in comparison to the existing results available for robust CT-RL methods. By comparison, the SOTA robust FVI work (Lutter et al., 2023b) provides no stability robustness guarantees. The Robust ADP work (Jiang, \& Jiang, 2014) provides robustness only to a narrow class of disturbances, and with very restrictive theoretical assumptions required (see RADP theoretical assumptions list in Table R1 included below). Furthermore, the RADP results apply to single-input systems only - they do not address the multi-input HSV.
>
> * As the Reviewer has recognized, we furthermore present substantial and compelling numerical studies on SOTA, highly-complex nonlinear systems such as the HSV, wherein we demonstrate significant performance improvements over the SOTA CT-RL methods.
>
> * Thus, both from a theoretical and empirical perspective, IPA is very well-suited to address nonlinear systems.
>
> * On accounting for the unkown approximation error and rigor: Thank you for this recommendation, we have introduced clarifying notation alongside explicit accounting for the approximation error in the revision. Please see pp. 5, ln. 224, 228, 233, 269, and to Theorem 3.1 on pp. 6, ln. 282, 284. We have also added language reflecting that the results of Theorem 3.1 are local for explicitness.

---

> > ### Author Response · Authors · 2024-11-19
> > **Response to Reviewer (Part 2)**
> >
> > > Similar analysis needs to be performed for all of the other approximations in the paper to achieve a local stability result with robustness to small perturbations. The way the paper stands now, the analysis is valid only in an infinitesimal neighborhood of the origin.
> >
> > * Thank you for this point. The results are local, as the conception of IPA relies on Kleinman structure, but it uses real environment trajectory data to accommodate the nonlinear dynamics.
> >
> > * Global stability results for nonlinear systems require substantial theoretical assumptions. To illustrate, please see the blue material added to Appendix A in the revision which outlines the key theoretical assumptions required by: 1) IPA, 2) leading ADP CT-RL works (Vrabie \& Lewis, 2009; Vamvoudakis \& Lewis, 2010; Jiang \& Jiang, 2014; Bian \& Jiang, 2022), and 3) the SOTA FVI works (Lutter et al., 2021, 2023b). In short, the ADP CT-RL algorithms which offer global results also require very stringent assumptions which hinder learning and closed-loop performance. As a result, these methods fail to synthesize for the simple second-order system benchmark (see Appendix M).
> >
> > * While theoretical results in IPA are local, we would kindly like to point out that these theoretical results are SOTA among the methods with demonstrated performance in CT-RL. Indeed, the leading cFVI and rFVI methods (Lutter et al 2021; 2023b) provide no stability results. Furthermore, IPA demonstrates these stability guarantees to 100\% effect across a wide range of large initial conditions implemented on a dense grid and environment uncertainties implemented by considering the worst-case dynamical perturbations for the respective environments (cf. Appendices J-L) on the three nonlinear environments studied.
> >
> > > Besides, the literature review surrounding CT-RL and ADP methods [in Appendix M] is sparse.
> >
> > * At the beginning of Appendix M, we provide a summary of the key limitations of ADP methods, and we refer the audience to (Wallace \& Si, 2024) where a comprehensive evaluation and ablation study was discussed at great length. To address the Reviewer's question, we took some pertinent info from (Wallace \& Si, 2024), where a comprehensive coverage of the leading ADP literature is conducted, to make the current discussion complete. We have updated Appendix M accordingly.
> >
> > > In Appendix M, the following vague comment is made about ADP methods:
> > > "As a result of ADP’s theoretical frameworks in adaptive and optimal control, Lyapunov arguments are available to prove qualitative properties including weight convergence and closed-loop stability results. However, the results require restrictive theoretical assumptions which are difficult to satisfy for even simple academic examples, and as a result these methods exhibit empirical issues."
> > > However, it is not specified what theoretical assumptions are restrictive and difficult to satisfy.
> >
> > * Thank you for this point. To directly address it, we have included a new Table R1 here which we have added to Appendix A in the revision listing the key theoretical results required by: 1) IPA, 2) leading ADP CT-RL works (Vrabie \& Lewis, 2009; Vamvoudakis \& Lewis, 2010; Jiang \& Jiang, 2014; Bian \& Jiang, 2022), and 3) the SOTA FVI works (Lutter et al., 2021, 2023b). We have also included a reference to this appendix after the above statement is made in the revision, alongside further discussion (see pp. 39, ln. 2077-2086).
> >
> > * In short, this Table R1 supports that the ADP CT-RL assumptions are restrictive compared to IPA.

---

> > > ### Author Response · Authors · 2024-11-19
> > > **Response to Reviewer (Part 3)**
> > >
> > > * As shown below, IPA is among the least restrictive in CT-RL in its theoretical assumptions. As a note, all methods require that be Lipschitz near origin to assure well-posedness of solutions to the system differential equations. Also note that none of the ADP designs resulted in meaningful controllers, please refer to (Wallace \& Si, 2024).
> > >
> > > ### Table R1: Theoretical Assumptions Required by SOTA CT-RL Works
> > >
> > > Algorithm|Assumption
> > > |-|-|
> > > **IPA (present work):**|Linearization of nonlinear system $(A, B)$ stabilizable and $(Q^{1/2}, A)$ detectable (for well-posedness and definiteness of regulation problem)
> > > ``|Full column rank of the IPA matrix $\mathcal{I}$ (20). This assumption is easy to satisfy, as the algorithm can continue to collect $l > \underline{n}$ samples until it is met. It is also virtually instantaneous to verify (matrix rank calculation). We observe no issue satisfying this assumption on the complex HSV in practice
> > > ``|Initial stabilizing nonlinear policy $\mu_{0}$
> > > **FVIs (Lutter et al., 2021, 2023b):**|$f$ and $g$ are smooth in their partial derivatives in the state $x$ and model uncertainty parameters $\theta$, and these partials are all known *a priori*
> > > ``|Undiscounted problem $\gamma = 1$ can be approximated by discounted problem $0 < \gamma < 1$
> > > ``|Discrete-time running cost $r(x,u)$ can be approximated by  continuous-time counterpart: $r(x,u) = \Delta t \\, r_{c}(x,u)$ with sample time $\Delta t$
> > > ``|Strict convexity of action penalty $g_{c}$
> > > ``|Availability of convex conjugate function to action penalty $g_{c}$
> > > ``|Higher-order terms in Taylor series expansion of optimal value $V^{*}$ are negligible
> > > ``|Existence of an *a priori* state grid $x \in \mathcal{D}$ to contain trajectories to for fitting procedure
> > > ``|Trajectories leaving the grid $x \in \mathcal{D}$ can be instantaneously re-initialized to the previous position inside the grid
> > > **IRL (Vrabie \& Lewis, 2009):**|There exists a sequence of sampling instants $t_{0} < t_{1} < \cdots < t_{l}$ such that the IRL regression matrix has full rank. This assumption is qualitatively similar to IPA, but the method does not lead to meaningful controllers in practice, as there is no constructive method to ensure the full rank condition (Wallace \& Si, 2024)
> > > ``|Chosen basis functions approximate optimal value and its gradient uniformly on compact sets
> > > ``|Basis functions for critic network are linearly-independent
> > > ``|Initial stabilizing policy
> > > **SPI (Vamvoudakis \& Lewis, 2010):**|Existence and uniqueness of least-squares solution to approximate HJB equation
> > > ``|PE assumption on various learning signals
> > > ``|Chosen basis functions approximate optimal value and its gradient uniformly on compact sets
> > > ``|Chosen basis functions approximate optimal policy uniformly on compact sets
> > > ``|Basis functions for critic network are linearly-independent
> > > ``|Basis functions for actor network are linearly-independent
> > > ``|Initial stabilizing policy
> > > **RADP (Jiang \& Jiang, 2014):**|Optimal value can be bounded from above and below by *a priori* known class $\mathcal{K}_{\infty}$ functions
> > > ``|Existence of *a priori* known compact set $\Omega_{0} \subset \mathbb{R}^{n}$ for which the closed-loop system under the initial policy is invariant with respect to the probing noise $d$
> > > ``|PE assumption on various learning signals
> > > ``|Chosen basis functions approximate optimal value and its gradient uniformly on compact sets
> > > ``|Chosen basis functions approximate optimal policy uniformly on compact sets
> > > ``|Basis functions for critic network are linearly-independent
> > > ``|Basis functions for actor network are linearly-independent
> > > ``|Initial stabilizing policy
> > > **CT-VI (Bian \& Jiang, 2022):**|Existence and uniqueness of solutions to an uncountable family of finite-horizon HJB equations
> > > ``|Properness of each solution to the finite-horizon HJB equation
> > > ``|Convergence of family of solutions of finite-horizon HJB equation to the infinite-horizon HJB solution
> > > ``|Invariance of closed-loop state/action trajectory to compact set with respect to the probing noise $d$
> > > ``|Initial *globally asymptotically stabilizing* policy
> > > ``|PE assumption on various learning signals
> > > ``|Chosen basis functions approximate optimal value and its gradient uniformly on compact sets
> > > ``|Chosen basis functions approximate optimal policy uniformly on compact sets
> > > ``|Chosen basis functions approximate optimal Hamiltonian uniformly on compact sets
> > > ``| Basis functions for critic network are linearly-independent
> > > ``| Basis functions for actor network are linearly-independent
> > > ``| Basis functions for Hamiltonian network are linearly-independent

---

> > > > ### Author Response · Authors · 2024-11-19
> > > > **Response to Reviewer (Part 4)**
> > > >
> > > > > Besides the literature and ablation study on ADP is mostly focused on the old result by Vamvoudakis and Lewis (2010). Besides, a lot of further development has happened after this classical work. For example, see the following works and references citing them/therein:
> > > > > * Kamalapurkar, R., Walters, P. and Dixon, W.E., 2016. Model-based reinforcement learning for approximate optimal regulation. Automatica, 64, pp.94-104.
> > > > > * Vamvoudakis, K.G., 2017. Q-learning for continuous-time linear systems: A model-free infinite horizon optimal control approach. Systems \& Control Letters, 100, pp.14-20.
> > > >
> > > > * On (Kamalapurkar, Walters, \& Dixon, 2016):
> > > >
> > > >   * We appreciate that this work relaxes a formal PE requirement (for which no systematic/constructive method exists to ensure PE for nonlinear systems) to a rank requirement placed on the trajectories and which is more easily confirmed by the designer.
> > > >   * IRL provides a similar result in this regard. In comparison, we only require full column rank $\underline{n} \triangleq \frac{n (n+1)}{2}$ of the IPA matrix $\mathcal{I} \in \mathbb{R}^{l \times \underline{n}}$ (20). This assumption is easy to satisfy, as the algorithm can continue to collect $l > \underline{n}$ samples until it is met. It is also virtually instantaneous to verify (matrix rank calculation). We observe no issue satisfying this assumption on the complex HSV in practice.
> > > >   * Additionally, the stability results of the 2016 work only prove uniform ultimate boundedness (UUB), a relatively limited stability result which does not suffice for many real-world application problems. The HSV, for instance, needs an asymptotic stability guarantee like IPA's for pitch regulation -- failure to arrest the pitch asymptotically results in severe aeroelastic heating effects which could ultimately result in catastrophic failure of the fuselage (Bolender and Doman, 2005, 2006a-b).
> > > >   * Furthermore, these UUB results require that a number of definiteness-related inequalities hold pointwise along the trajectories. Ensuring that these inequalities are met for trajectories of a higher-order application like the HSV is not likely straightforward. Lastly, this work studies relatively limited academic examples.
> > > >
> > > > * On (Vamvoudakis, 2017):
> > > >
> > > >   * The work (Vamvoudakis \& Lewis, 2010) mentioned by the reviewer was extensively analyzed and included in Appendix M for a number of reasons.
> > > >   * Firstly, this work is seminal and subsequent works draw heavily from its ideas. Indeed, this includes the gradient-descent training method used in the (Vamvoudakis, 2017) reference mentioned by the Reviewer, which is adopted virtually directly from Vamvoudakis's 2010 work and addresses linear systems only.
> > > >   * Secondly, our second-order-system benchmark system with details provided in Appendix M is directly from (Vamvoudakis \& Lewis, 2010). Thus, for the sake of reproducibility it was integral we include this work in the analysis.
> > > >
> > > > * The analysis in our Appendix M also examines leading ADP CT-RL works, e.g. (Bian \& Jiang, 2022), which are more recent. Together, these four seminal and representative works have significant influence on the ADP CT-RL community, amassing over 2,700 citations and 1,500+ citations for (Vamvoudakis \& Lewis, 2010) alone.

---

> > > > > ### Author Response · Authors · 2024-11-19
> > > > > **Response to Reviewer (Part 5)**
> > > > >
> > > > > > In Appendix F, the authors mention "Many SOTA ADP CT-RL algorithms require the persistence of excitation (PE) condition in proofs of algorithm properties". There are many newer results which relax the PE condition with initial/interval/finite-time excitation conditions. See the following references for example:
> > > > > > * Jha, S.K., Roy, S.B. and Bhasin, S., 2019. Initial excitation-based iterative algorithm for approximate optimal control of completely unknown LTI systems. IEEE Transactions on Automatic Control, 64(12), pp.5230-5237.
> > > > > > * Yang, Y., Pan, Y., Xu, C.Z. and Wunsch, D.C., 2022. Hamiltonian-driven adaptive dynamic programming with efficient experience replay. IEEE Transactions on Neural Networks and Learning Systems.
> > > > > > The authors can discuss these newer results that relax the PE condition and analyze how they relate to or could potentially improve the proposed IPA method.
> > > > >
> > > > > * Thank you for bringing these references to the conversation, we have added a discussion of them to our literature review in the Introduction (see pp. 1, ln. 42-59), to reflect the following:
> > > > >
> > > > > * On Jha et al.: This method is promising, but it applies to LTI systems only. Extending these relaxed PE results to affine nonlinear systems is not expected to be straightforward.
> > > > >
> > > > > * On Yang et al.:
> > > > >
> > > > >   * This ADP CT-RL work applies to affine nonlinear systems. However, the stability results only prove uniform ultimate boundedness (UUB), and the assumptions required to prove these results are complex and must hold pointwise along system trajectories. Ensuring these inequalities poses significantly challenging with increased system dimension and nonlinearity.
> > > > >   * UUB is a relatively weak stability result, please see the discussion provided of UUB in our discussion of the (Kamalapurkar, Walters, \& Dixon, 2016) reference above.
> > > > >   * As is characteristic of ADP-based CT-RL works, this work is deficient in its performance evaluations. This work studies a single second-order academic example:
> > > > > \begin{align}
> > > > > \begin{bmatrix}
> > > > >  \dot{x}\_{1} \\\\
> > > > >  \dot{x}\_{2}
> > > > > \end{bmatrix}
> > > > > \=
> > > > > \begin{bmatrix}
> > > > >  x_{2} \\\\
> > > > >  \- x_{1} - \frac{1}{2} (1 - x_{1}^{2}) x_{2}
> > > > > \end{bmatrix}
> > > > > +
> > > > > \begin{bmatrix}
> > > > >  0 \\\\
> > > > >  x_{1}
> > > > > \end{bmatrix}  u
> > > > > \end{align}
> > > > >
> > > > >   * This system admits a known closed-form optimal value $V^{*}(x) = x_{1}^{2} + x_{2}^{2}$, and with this \emph{a priori} knowledge the authors choose the bases such that the optimal value lies in the direct span of the bases chosen: $\phi(x) = [x_{1}^{2} \\;\\; x_{2}^{2}]$.
> > > > >   * Even though this method admits a relaxed PE condition, we still place this work among the same ADP works that do not lead to meaningful controllers.
> > > > >
> > > > >
> > > > > ## Questions:
> > > > >
> > > > > > **1.** Does the state-dependent part of the cost need to be quadratic? Specifically, can the $x^{T} Q x$ term be generalized to some positive semi-definite function $Q(x)$. I can understand the challenges generalizing so for the $u^{T} R u$, but $x^{T} Q x$ must be easy if linearizations are being used in the integral approximation anyway.
> > > > >
> > > > > * Taking advantage of the quadratic cost structure $x^{T} Q x$ is what enables our novel integral performance approximation (IPA) scheme via (14). IPA allows us to achieve demonstrated well-behaved system responses from the nonlinear policies $\mu_{i}$ (18) as well as high data efficiency. It also enables proof of our theoretical guarantees (cf. Theorem 3.1).
> > > > >
> > > > > * The use of QR cost is quite frequent in RL formulations as well as classical ones for controls applications. Indeed, all four of the references suggested by the Reviewer use QR cost $r(x, u) = x^{T} Q x + u^{T} R u$:
> > > > >
> > > > >   * (Kamalapurkar, Walters, \& Dixon, 2016)
> > > > >   * (Vamvoudakis, 2017)
> > > > >   * (Jha et al., 2019)
> > > > >   * (Yang et al., 2024)
> > > > >
> > > > > * As the Reviewer may very well be aware, quadratic QR cost addresses a variety of real-world application domains. For example, control systems in the HSV application and aerospace systems more broadly are most frequently designed with quadratic state penalties $x^{T} Q x$, see e.g. (Dickeson et al., 2009 a-b).

---

> > > > > > ### Author Response · Authors · 2024-11-19
> > > > > > **Response to Reviewer (Part 6)**
> > > > > >
> > > > > > > **2.** Does the linearization in (13) need to be necessarily around $x=0$? What would be difficult about constructing a function $K_{i}(x) = - \frac{\partial \mu_i}{\partial x} (x)$? across a range of values of x, similar to gain scheduling methods?
> > > > > >
> > > > > > * Thank you for this point. Please see our above response to your Weaknesses section emphasizing that the IPA policies $\mu_{i}$ (18) are already fully nonlinear policies. The $K_{i}$ are for integral performance approximation in (14) only -- they are not the IPA policies.
> > > > > >
> > > > > > * On changing the equilibrium point: As is standard in optimal control and CT-RL formulations, we assume an equilibrium at $x = 0$. This can readily generalize to arbitrary equilibria $x_{e} \in \mathbb{R}^{n}$ via a change of variables.
> > > > > >
> > > > > > * If $\dot{z} = \tilde{f}(z) + \tilde{g}(z) u$ is any affine nonlinear system with equilibrium $z = x_{e}$, then consider the change of variables $x = z - x_{e}$ and translated dynamics $f(x) = \tilde{f}(x + x_{e})$, $g(x) = \tilde{g}(x + x_{e})$. This translated system $(f, g)$ has an equilibrium at $x = 0$, and we note that its dynamics satisfy
> > > > > >
> > > > > > \begin{align}
> > > > > > \dot{x} = \frac{d}{dt} \\{z - x_{e}\\} = \frac{d}{dt} \\{z\\} = \tilde{f}(z) + \tilde{g}(z) u = \tilde{f}(x + x_{e}) + \tilde{g}(x + x_{e}) u = f(x) + g(x) u
> > > > > > \end{align}
> > > > > >
> > > > > > * i.e., $f(x) + g(x) u = \tilde{f}(z) + \tilde{g}(z) u$, so we may apply IPA to nonlinear systems with  nonzero equilibria.
> > > > > >
> > > > > > * In regards to gain scheduling, the utility of a scheduled design for purposes of integral performance approximation is somewhat limited. Strictly speaking, the Taylor expansion about an arbitrary $x_{0} \in \mathbb{R}^{n}$ goes $\mu_{i}(x_{0} + x) = \mu_{i}(x_{0}) + \left. \frac{\partial \mu_{i}}{\partial z} \right|\_{z = x_{0}} (x - x_{0}) + \text{H.O.T.}$. For $x_{0} = 0$, this collapses nicely: $\mu_{i}(0 + x) = \mu_{i}(0) +  \left. \frac{\partial \mu_{i}}{\partial z} \right|\_{z = 0} (x - 0) + \text{H.O.T.} = - K_{i} x + \text{H.O.T.}$. The zero evaluation of the policy at $x_{0} = 0$ is what enables the linear algebra of the data reuse. Having to store $\mu_{i}(x_{0})$ across a schedule of $x_{0}$ would require re-integrating the policy value at each iteration $i$ (precisely what IPA is intended to bypass). Thus, the policy evaluating to 0 at the equilibrium $x_{0} = 0$ in combination of IPA's use of trajectory data from the actual nonlinear process leads to its demonstrated performance.

---

> > > > > > > ### Comment · Reviewer_ZBUg · 2024-11-19
> > > > > > >
> > > > > > > I thank the authors for their detailed responses. The authors addressed my comments. I am increasing the score to 8.

---

> > > > > > > > ### Author Response · Authors · 2024-11-19
> > > > > > > > **We thank this Reviewer for their insights and positive review!**
> > > > > > > >
> > > > > > > > It has been our pleasure to have this insightful discussion, we thank this reviewer greatly for their insights and their positive review in recognizing the significance of IPA.

---

### Author Response · Authors · 2024-11-19
**Global Response**

We greatly appreciate the Reviewers and the ACs for their time and feedback in reviewing our work.

We are very pleased that the Reviewers have recognized the significance of IPA for its several important contributions to CT-RL:

>* "... the **contributions are clear and of significant importance** for solving continuous-time optimal control problems." (Reviewer ZBUg)
>* "IPA ... enable[s] **highly data-efficient and robust control.**" (Reviewer 5RXP)
>* "The simulation and ablation studies are **extensive** and make a **compelling case** for the developed controller." (Reviewer ZBUg)
>* "The critic network design is **novel.**" (Reviewer K99L)
>* "[IPA] provides **theoretical guarantees** for convergence, optimality, and stability, **validated through extensive evaluations.**" (Reviewer 5RXP)
>* "Simulation results on three optimal control tasks show that the **proposed method outperforms SOTA methods.**" (Reviewer K99L)
>* "[The work] demonstrates IPA CT-RL's **successful application to HSV control** ...  which presents **additional challenges due to unstable and non-minimum phase dynamics.**" (Reviewer 5RXP)

For the issues raised by the Reviewers, we have taken great care to ensure all of the Reviewers' valuable feedback has been thoroughly addressed in the rebuttals. A detailed explanation is provided in the individual responses below to each and every point raised by the Reviewers, and we have revised the manuscript accordingly (please see attached PDF). In summary, we believe that most of these issues are due to misunderstanding stemming from our lack of presentation of some of the related text. We are grateful to the Reviewers' questions that have helped us further clarify our contributions.

To provide a context of of IPA's contributions to CT-RL algorithms developed based on an affine nonlinear model (results exist for general nonlinear systems, but as discussed in our Introduction and in our detailed responses to Reviewers below, these are at an early stage), in relation to the existing classes of comparable CT-RL methods, the field has:

1. ADP-based CT-RL methods, which provide theoretical results but with very stringent assumptions, and which fail to synthesize meaningful controllers.

2. Deep RL FVI-based CT-RL methods, which provide no theoretical results, yet have demonstrated meaningful control performance. These methods also inherit high data/computational complexity from training deep networks.

3. The proposed IPA method, which provides local theoretical results of convergence, optimality, closed-loop stability, and robustness, and which provides extensive evaluations on perhaps the most realistic and complex CT-RL environments to-date, demonstrating exceptional data efficiency.

This work is dense, and thus we truly appreciate Reviewers’ comments that have helped and reminded us to provide necessary backdrops. Our focus is on innovative designs under meaningful and realistic conditions. To this end, we would like to note that our evaluations of the HSV environment are based on a highly-complex NASA Langley aeropropulsive model (Shaughnessy et al, 1990), not the usual simplified HSV used in most of the academic publications.

We look forward to your positive feedback, and we hope that the reviewers after seeing our responses are just as enthusiastic as we are about this work.

---

### Meta-Review · Area_Chair_LVnR · 2024-12-10

**Metareview:**

This paper studies continuous-time model-based reinforcement learning for control-affine nonlinear systems. The paper is well-written, the contributions (both theoretical and experimental ones) are significant, substantial, and novel. There were some concerns regarding the rigor and clarity of the theoretical results, as well as a fair and detailed comparison with the related works. The concerns have been addressed meticulously by the authors in the rebuttal. I think it is a good paper that should be accepted.

**Additional Comments On Reviewer Discussion:**

There were some concerns regarding the rigor and clarity of certain theoretical results, and have been addressed satisfactorily by the authors' rebuttals.

---

### Decision · Program_Chairs · 2025-01-22

Accept (Poster)